# Targeting ubiquitin signaling vulnerabilities in KEAP1-inactivated lung cancer

Varun Jayeshkumar Shah[1,12], Oliver Hartmann [2,3,4,12], Martin Wegner [1], Cristian Prieto-Garcia [1,5,6], Rubina Kazi[1], Viktoria von Heyl zu Herrnsheim[2,3,4], Amin Wanli[7,8], Igor Mačinković [7,8], Bianka Bohnacker [2,3,4], Koraljka Husnjak [1], Dmitry Namgaladze[8], Mathias Rosenfeldt[9], Manuel Kaulich [1,10], Markus E Diefenbacher [2,3,4]✉ & Ivan Dikic [1,11]✉

## Abstract

Lung cancer cells rely on protein homeostasis regulators, particularly the ubiquitin-proteasome system (UPS), to sustain malignancy. Genetic alterations in UPS components, such as E3 ubiquitin ligases (E3s) and deubiquitinating enzymes (DUBs), are common and create context-dependent therapeutic dependencies. To investigate how these genetic alterations drive tumor formation, we conducted CRISPR screens on metabolically stressed murine lung cancer models and identified specific cancer dependencies, including ubiquitin ligase subunit KEAP1. Although KEAP1 is frequently mutated in aggressive non-small cell lung cancers (NSCLC, ~15%), our findings reveal an unexpected proto-oncogenic role for KEAP1 in a genetically defined subset of NSCLC. Mechanistically, Keap1 deletion activated Nrf2 and upregulated Aldh3a1. This led to elevated reductive stress and suppressed tumor growth. Given the poor prognosis of KEAP1-mutated patients, combinatorial CRISPR dropout screens revealed druggable E3s and DUBs as Keap1-dependent co-vulnerabilities. Notably, depleting these co-dependencies, such as the E3 ligases Herc2, Ubr4 and Huwe1 ablated the in vivo development of Keap1-inactivated tumors. We demonstrate that targeting the UPS represents an underexplored, promising therapeutic approach for patients with KEAP1-inactivated tumors, especially under metabolic stress.

**Keywords** NSCLC; CRISPR/Cas9; Keap1; Reductive Stress; Ubiquitin-Proteasome System
**Subject Categories** Cancer; Post-translational Modifications & Proteolysis

## Introduction

During oncogenic transformation, cells experience elevated levels of proteotoxic stress as a consequence of rapid growth, metabolic stress and genomic imbalances. These alterations impose a strong selective pressure and are a prerequisite to establishing the 'hallmarks of cancer' (Deshaies, 2014; Hanahan, 2022). Furthermore, genetic alterations in proto-oncogenes and tumor suppressors not only drive selective pressure but also directly affect tumor growth and, ultimately, patient performance and survival expectancy during therapy. As clinically relevant examples, patients harboring alterations in the MAPK signaling pathway, such as activating mutations in the oncogenes KRAS along with alterations in tumor suppressor genes TP53 (p53), liver kinase B1/serine/threonine-protein kinase STK11 (LKB1/STK11) and Phosphatase and tensin homolog (PTEN), develop more aggressive non-small cell lung carcinoma (NSCLC) tumors, with a high frequency of metastasis, and limited treatment options (Cancer Genome Atlas Research Network, 2012; Collisson et al, 2014).

Recent advances towards molecularly targeted therapies, such as the KRAS inhibitors (KRASi) Sotorasib and Adagrasib, show significant improvements in the treatment of KRAS-mutated NSCLC patients (Lanman et al, 2020; Fell et al, 2020), yet their efficacy is limited by inevitable resistance, exacerbating poor outcomes and high mortality rates. Furthermore, recent research suggests that KRAS-driven cancers are addicted to proteostasis for growth (Awad et al, 2021; Lv et al, 2023), and complementary therapies that enhance proteostatic stress may be feasible.

In the past, the appreciation that UPS is a contributor to cancer development and disease progression led to the development of targeted therapies, such as the proteasome inhibitors bortezomib (Velcade®), Carfilzomib and Ixazomib, which have positively transformed the treatment of multiple myeloma and other malignancies (Deshaies, 2014). Conversely, bortezomib-based mono and combination therapy have limited clinical indication

[1]Institute of Biochemistry II, Goethe University Frankfurt, Frankfurt am Main, Germany. [2]Institute of Lung Health and Immunity (LHI), Helmholtz Munich, Comprehensive Pneumology Center (CPC-M),Germany, Member of the German Center for Lung Research (DZL), Neuherberg, Germany. [3]Institute of Experimental Pneumology, Ludwig Maximilian University Munich, Munich, Germany. [4]DKTK Munich, Munich, Germany. [5]Instituto de Biomedicina de Sevilla (IBiS, Hospital Universitario Virgen del Rocío/CSIC/Universidad de Sevilla), Seville, Spain. [6]Departamento de Biología Celular, Facultad de Biología, Universidad de Sevilla, Seville, Spain. [7]Department for Immunity of Inflammation, Mannheim Institute for Innate Immunoscience, Medical Faculty Mannheim Heidelberg University, Mannheim, Germany. [8]Institute of Biochemistry I, Faculty of Medicine, Goethe University Frankfurt, Frankfurt am Main, Germany. [9]Institute of Pathology, University of Würzburg, Würzburg, Germany. [10]Frankfurt Cancer Institute, Frankfurt am Main, Germany. [11]Buchmann Institute for Molecular Life Sciences, Goethe University Frankfurt, Frankfurt am Main, Germany. [12]These authors contributed equally: Varun Jayeshkumar Shah, Oliver Hartmann.✉E-mail: markus.diefenbacher@helmholtz-munich.de; dikic@biochem2.uni-frankfurt.de

in the treatment of NSCLC, which limits its usage and supports the need to develop modulators of other regulators of proteostasis, such as E3 ubiquitin ligases and DUBs (Chua et al, 2023). However, only a few studies have investigated the biological consequences of inactivating E3 ubiquitin ligases and DUBs in the context of different genetic subtypes of lung cancer in vivo.

Cellular redox homeostasis is an essential and dynamic process that ensures the balance between reducing and oxidizing reactions within tumor cells (Hayes et al, 2020). While oxidative stress has been extensively studied, the impact of reductive stress on the proteostasis of the malignant cells is a rather emerging concept. An imbalance in the accumulation of reducing factors, such as NADH, NADPH, or GSH, leads to disruptions in de novo lipid, amino acid, and nucleotide biosynthetic pathways (Ge et al, 2024). One key mechanism governing the expression and abundance of these reducing agents is NFE2L2 (nuclear factor-erythroid 2 p45-related factor 2 / nuclear factor-erythroid 2-related factor 2, hereafter NRF2), the master transcriptional regulator of the cellular antioxidant response. While responsible for resolving antioxidant stress, NRF2 is a key player in oncogenic cells, where it is frequently overrepresented and drives metabolic and biochemical rewiring, thereby assisting cells in tolerating redox stress and thriving in otherwise unfavorable conditions (Glorieux et al, 2024). NRF2 is frequently stabilized in many cancers, particularly NSCLC, where mutations in the Kelch-like ECH-associated protein 1 (KEAP1)–NRF2 pathway are found in up to 30% of cases. At the same time, the addiction of tumor cells to NRF2 may provide an exploitable vulnerability (Baird and Yamamoto, 2020; Best et al, 2018; Wu and Papagiannakopoulos, 2020). Recently, NRF2 has been identified as a critical regulator of reductive stress. It was demonstrated that the inactivation of KEAP1, leading to NRF2 hyperactivation, creates a selective vulnerability to complex I inhibition, which impairs NADH oxidation capacity and potentiates reductive stress. This presents a metabolic vulnerability in lung cancer (Weiss-Sadan et al, 2023; Ge et al, 2024). Additionally, KEAP1 and NRF2 have been reported to modulate multiple aspects of mitochondrial function, including quality control, function and structure. Since mitochondria play a crucial role in regulating metabolism and energy production in cancer cells (Sabouny et al, 2017; Dinkova-Kostova and Abramov, 2015), particularly in lung cancer, where mitochondrial respiration is heavily relied upon, lung tumors are sensitive to mitochondrial dysfunction and the inhibition of core metabolic pathways (Han et al, 2023).

Based on the dependency of tumor cells towards rewiring of metabolic redox pathways, we performed a CRISPR screen to specifically target components of the UPS system to identify potential therapeutic vulnerabilities. To this end, we interrogated common dependencies towards the UPS in three genetically tailored, patient-relevant surrogate model subsets of KRAS-mutant NSCLC by co-occurring genetic events in TP53 ($Kras^{G12D}Tp53^{-/-}$; KP), TP53 and LKB1 ($Kras^{G12D}Tp53^{-/-}Lkb1^{-/-}$; KPL), as well as TP53 and PTEN ($Kras^{G12D}Tp53^{-/-}Pten^{-/-}$; KPP). A comparison of vulnerabilities between the genetic subtypes identified KEAP1, a substrate-specific adapter of a multisubunit Cullin 3-based E3 ligase complex, as a common dependency. Furthermore, a combinatorial CRISPR/Cas9 dropout screen, targeting members of the UPS, unraveled novel Keap1 co-dependencies in NSCLC. Overall, we uncovered a prevalent addiction of KEAP1-inactivated or NRF2-hyperactivated tumors

towards reductive stress, which could harbor significant therapeutic potential.

# Results

## CRISPR screening reveals novel E3 ubiquitin ligases as lung cancer vulnerabilities

Lung tumors represent a solid tumor entity with exceptionally high mutational burden. Several metabolic stress-inducing mutations are prevalent in this entity, with common loss-of-function mutations in PTEN (3% adenocarcinoma (ADC) and 20% squamous cell carcinoma (SCC)) or LKB1 (16% ADC and 2% SCC), directly affecting patient survival and leading to therapy failure, such as immune checkpoint inhibitor treatment (Appendix Fig. S1A–H) (Collisson et al, 2014; Cancer Genome Atlas Research Network, 2012; Araghi et al, 2023). We established three murine patient surrogate cell lines with the genetic background of (1) Kras$^{G12D}$:Tp53$^{\Delta}$ (KP mice/cells, named after Kras and P53), (2) Kras$^{G12D}$:Tp53$^{\Delta}$:Pten$^{\Delta}$ (KPP mice/cells), and (3) Kras$^{G12D}$:Tp53$^{\Delta}$:Lkb1$^{\Delta}$ (KPL mice/cells), thereby covering the most common genetic alterations, as well as having the ability to represent metabolic and redox stress adapted NSCLC (Fig. 1A) (Fischer et al, 2022; Prieto-Garcia et al, 2020; Hartmann et al, 2021).

Publicly available patient data show that several E3 ubiquitin ligases and DUB genes are genetically altered in NSCLC, with these alterations occurring in various combinations (Cancer Genome Atlas Research Network, 2012; Collisson et al, 2014; Jin et al, 2021; Ye et al, 2023). The biological consequence and extent towards cancer fitness of these UPS members is poorly understood. To gather first systemic insights into the role of the UPS in NSCLC, we performed bioinformatic analyses of publicly available data regarding recurring genetic alterations of DUBs and E3 ligases in human lung tumors. This led to the identification of recurring genetic alterations in 24 E3 ligases, one E2 ubiquitin-conjugating enzyme and 13 DUBs (Fig. 1B–D, source: cBioPortal.org) (Gao et al, 2013).

Next, we performed a CRISPR dropout viability screen in three murine patient surrogate cell lines (Fig. 1A). The murine-specific and UPS-targeted single sgRNA screening library was generated using the 3Cs (covalently-closed-circular-synthesized) technology, followed by an assessment of proliferation and dropout effects (Diehl et al, 2021; Wegner et al, 2019). For each target gene, we chose the top-ranked four gRNA sequences and additionally included essential gene and non-targeting gRNAs, resulting in a total of 185 gRNAs in the targeted "E3-DUB" single sgRNA pooled library (Fig. EV1A–D). Cells were transduced at an MOI of 0.5 at a 1000x coverage. After lentiviral transduction, cells were collected on day 2 (for reference time point) and selected by puromycin. Upon infection and subsequent selection, cells were cultured for 14 days (about 10 divisions), followed by their harvesting, genomic DNA extraction and subsequent NGS and data analysis (Fig. EV1E). We utilized MAGeCK, an established algorithm for the analysis of CRISPR/Cas screens, on raw gRNA read counts from three replicates to assess the reproducibility of our single screens in terms of gene phenotypes. We then computed genes with positive/negative effects on cell fitness in all three cell lines by analyzing log2

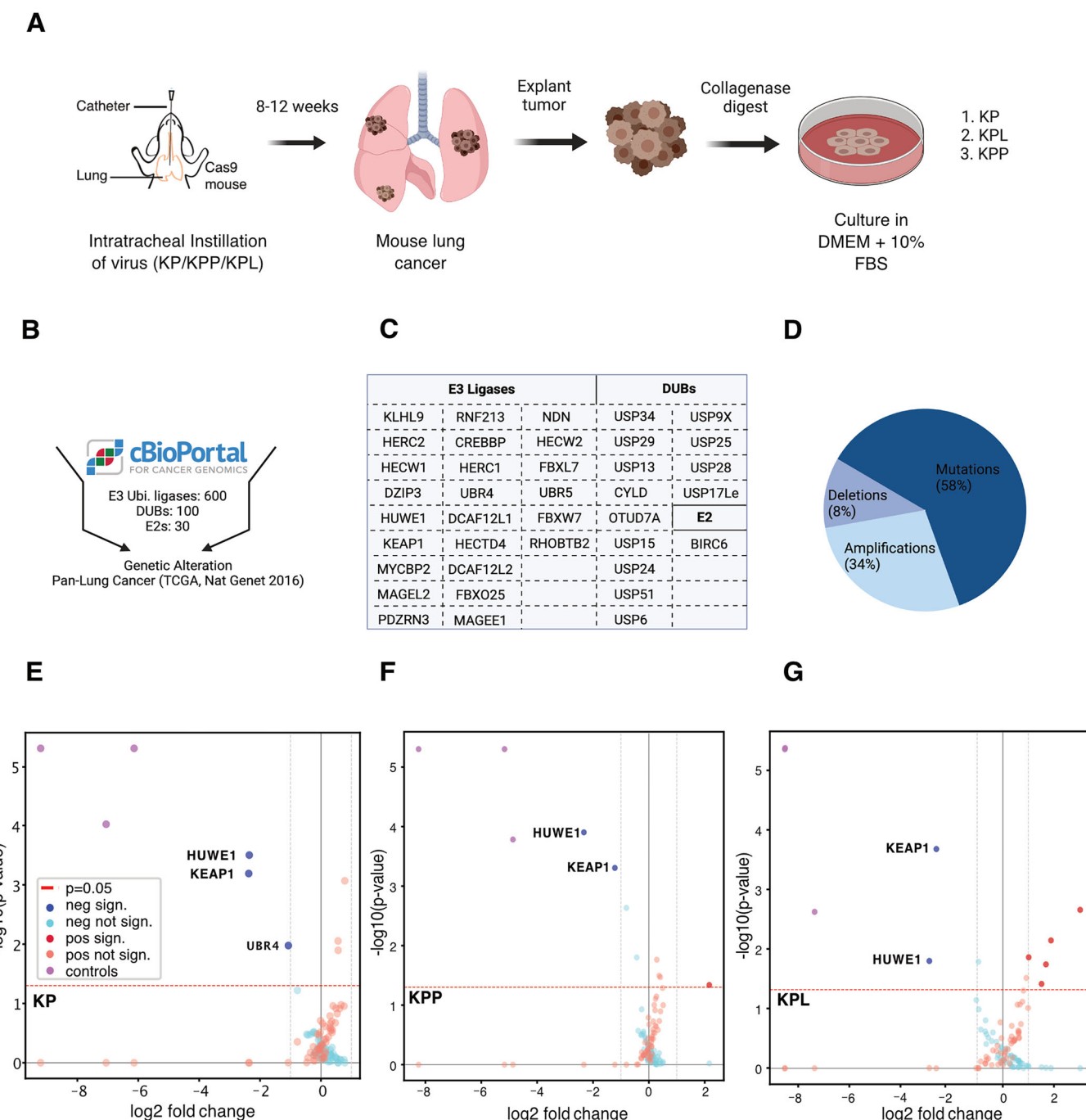

**Figure 1. CRISPR screening reveals novel E3 ubiquitin ligases as lung cancer vulnerabilities.**

(A) Schematic representation of intratracheal tumor induction *via* CRISPR/Cas9 and generation of primary murine lung tumor cells. (B, C) Genetic alterations for the selected list of E3 ligases and DUBs in a pan-lung cancer panel (TCGA, Nat. Genet 2016, source cBioProtal.org). (D) Classification of UPS components identified in C based upon genetic alteration. (E–G) Volcano plots of MAGeCK-derived LFCs and *p* values (FDR <20%) for the dropout viability screens in KP (E), KPP (F), and KPL (G) cell lines. Data were acquired over three replicates. Hit scoring and statistical testing were performed according to MAGeCK's robust rank aggregation (RRA) test for gene-level enrichment/depletion, assuming a negative binomial distribution of single gRNAs. Source data are available online for this figure.

fold changes with associated p-values (Li et al, 2014). By using this approach, we were able to identify several common and unique E3 ligases as positive regulators of cell fitness using our patient surrogate model cell lines (Fig. 1E–G). The identified positive regulators included the HECT E3 ligases Huwe1 (HECT, UBA, and

WWE domain-containing E3 ubiquitin protein ligase 1), Ubr4 (ubiquitin protein ligase E3 component n-recognin 4), and the E3 ligase subunit Keap1. The identification of Huwe1 is well in line with previous reports describing its proto-oncogenic function in NSCLC; HUWE1 regulates the stability or activity of key proteins

in oncogenesis, such as c-MYC, p53 and MCL-1 (Qi et al, 2022). UBR4, an E3 ligase known for its role in the N-end rule pathway, has been reported to degrade aggregation-prone nascent polypeptides, unimported mitochondrial precursors and stress response components (Nakatani et al, 2005; Haakonsen et al, 2024). In contrast, the identification of the KEAP1 was unexpected, given its well-established tumor-suppressive role (Fig. 1E–G) (Baird and Yamamoto, 2020; Wu and Papagiannakopoulos, 2020; Romero et al, 2017). The screen not only confirmed known factors influencing tumor cell fitness but also identified novel players affecting tumor cell growth.

## KEAP1 inactivation drives NRF2–ALDH3A1–dependent NADH-reductive stress

KEAP1 is the major regulator of NRF2 stability and thereby significantly contributes to redox stress signaling and reactive oxygen species (ROS) tolerance. Even though Keap1 has so far been reported to act as a tumor suppressor, our findings, aligned with recent studies, emphasize its proto-oncogenic role in a genetically defined subset of NSCLC (Weiss-Sadan et al, 2023). To dissect the mechanistic consequences of KEAP1 inactivation, we pharmacologically inhibited KEAP1 using KI696, an inhibitor of the KEAP1–NRF2 interaction (Crisman et al, 2023). Transcriptomic profiling of KI696-treated cells revealed extensive NRF2-dependent transcriptional reprogramming, including the induction of transcription factors associated with stress responses, such as Ahr, Jun/AP-1 and Nf-kB (Figs. 2A and EV2A). Consistently, NRF2 activity was markedly upregulated across all three murine cell lines tested (KP, KPL, and KPP) (Figs. 2B and EV2B). Whole-cell proteomics corroborated these findings, demonstrating robust induction of canonical NRF2 targets upon KI696 treatment (Figs. 2C and EV2C,D). Importantly, both transcriptomic and proteomic analyses revealed a significant increase in Nrf2, Aldh3a1 and Ugdh expression, well-established markers associated with reductive stress (Figs. 2B,C and EV2B–D) (Weiss-Sadan et al, 2023).

ALDH3A1 and UGDH, known NRF2 target genes, play a substantial role in regulating the NADH/NAD$^+$ balance in KEAP1-dependent cells. To further examine this link, we investigated the effect of KI696 exposure on the NADH/NAD$^+$ ratio in the patient surrogate cell lines. Interestingly, KI696 treatment led to a significant increase in the NADH/NAD$^+$ ratio across all three cell lines (KP, KPL, and KPP) (Fig. 2D). To determine whether this effect was mediated *via* NRF2 signaling, we conducted rescue experiments by knocking down Nrf2 and Aldh3a1. Depletion of either restored NADH/NAD$^+$ ratios to baseline levels in Keap1-inactivated cells in KPL and KP, indicating that the Keap1–Nrf2–Aldh3a1 axis is a key regulator of NADH/NAD$^+$ homeostasis and reductive stress (Fig. 2E and EV3A–C). In addition, Keap1 inhibition resulted in a modest increase in NADPH/NADP$^+$ levels in KPL and KPP cells, while no significant change was observed in KP cells (Fig. 2F). However, knockdown of Nrf2 or Aldh3a1 had no effect (Fig. EV3D–F), suggesting that this axis primarily regulates NADH/NAD$^+$ rather than NADPH/NADP$^+$ ratio.

Next, we evaluated the impact of Keap1 inhibition on mitochondrial function. Treatment with KI696 or Bardoxolone methyl (BM), an additional Keap1 inhibitor that has entered clinical phase 2/3 trials (NCT03019185), reduced mitochondrial respiratory capacity, as indicated by decreased maximal oxygen consumption rate (OCR) and respiratory reserve capacity, while basal and ATP-linked OCR remained unchanged (Figs. 2G and EV3G). We further assessed whether Keap1 inhibition affected ROS levels. Interestingly, Keap1 inhibition did not lead to any measurable change in cellular ROS levels under basal conditions (Fig. EV3H). Furthermore, we examined the effect of Keap1 depletion using CRISPR/Cas9. Consistent with CRISPR screen data (Fig. 1E–G), genetic inactivation of Keap1 led to a significant reduction in cell viability (Figs. 2H and EV3I). Interestingly, co-depletion of Nrf2 along with Keap1 partially restored cell viability, suggesting that Nrf2 activation contributes to the proliferative defect, but that Nrf2-independent pathways downstream of Keap1 also play a role in regulating cell proliferation (Figs. 2I and EV3J).

We next assessed pharmacological KEAP1 inhibition in human lung cancer cell lines CALU1 (KEAP1 wt) and H1299 (KEAP1 wt) using Bardoxolone methyl (BM) and Omaveloxolone. Treatment with BM and Omaveloxolone significantly altered NRF2 and ALDH3A1 localization in KEAP1 wild-type NSCLC cells (Appendix Fig. S2A,B). Consistent with these molecular changes, BM and Omaveloxolone reduced cellular proliferation and increased cell death (Appendix Fig. S3A,B). Altogether, these findings establish a mechanistic link between KEAP1 inactivation and reductive stress. KEAP1 inhibition activates NRF2, inducing ALDH3A1 expression, elevating NADH/NAD$^+$ ratios, and triggering reductive stress that impairs cell proliferation.

## KEAP1 inactivation restricts oncogenic transformation in NSCLC

KEAP1 is frequently altered in NSCLC ( ~ 15%), and its expression is significantly upregulated in NSCLC tumor samples compared to non-transformed tissue (Fig. EV4A–C). In addition, patient survival data based on Keap1 status (wild-type or mutant, excluding expression data) show no difference in ADC and SCC (Fig. EV4D). We used CRISPR/Cas9-mediated gene targeting as described previously to delineate the function of Keap1 in vivo (Hartmann et al, 2021). To this end, we used a constitutive Rosa26$^{Sor-CAGG-Cas9-IRES-GFP}$ transgenic mouse to induce primary lung lesions in mice after airway infection with recombinant adeno-associated viruses (AAVs). Here, 8-week-old mice were intratracheally intubated with rAAV virions containing sgRNA cassettes targeting sequences that inactivate Tp53 (p53$^\Delta$), Pten (Pten$^\Delta$) and Stk11/Lkb1 (Lkb1$^\Delta$) and introduce the oncogenic mutation G12D, via a repair template, into the Kras locus. We designate these mice as KPP (Kras$^{G12D}$; Tp53$^\Delta$; Pten$^\Delta$) or KPL (Kras$^{G12D}$; Tp53$^\Delta$; Lkb1$^\Delta$) (Fig. 1A). To investigate Keap1's role in tumor induction, we included one sgRNA cassette targeting Keap1 in the experimental cohort of KPP (Keap1$^\Delta$, referred to as KPKP), KPL mice (Keap1$^\Delta$, referred to as KPKL) and KPLP mice (Keap1$^\Delta$, referred to as KPKPL) (Fig. 3A,E). The assessment of Keap1 abundance through IHC revealed that the majority of primary lesions lacked Keap1 among KPKP, KPKL and KPKPL animals (Fig. 3B,E). Furthermore, we could observe that the concurrent targeting of Keap1 at the point of tumor induction substantially suppressed NSCLC formation. In KPKP and KPKL models, the total tumor area and the number of tumor lesions per animal were reduced when compared to KPP and KPL mice, respectively. Thereby, the in vivo experiment

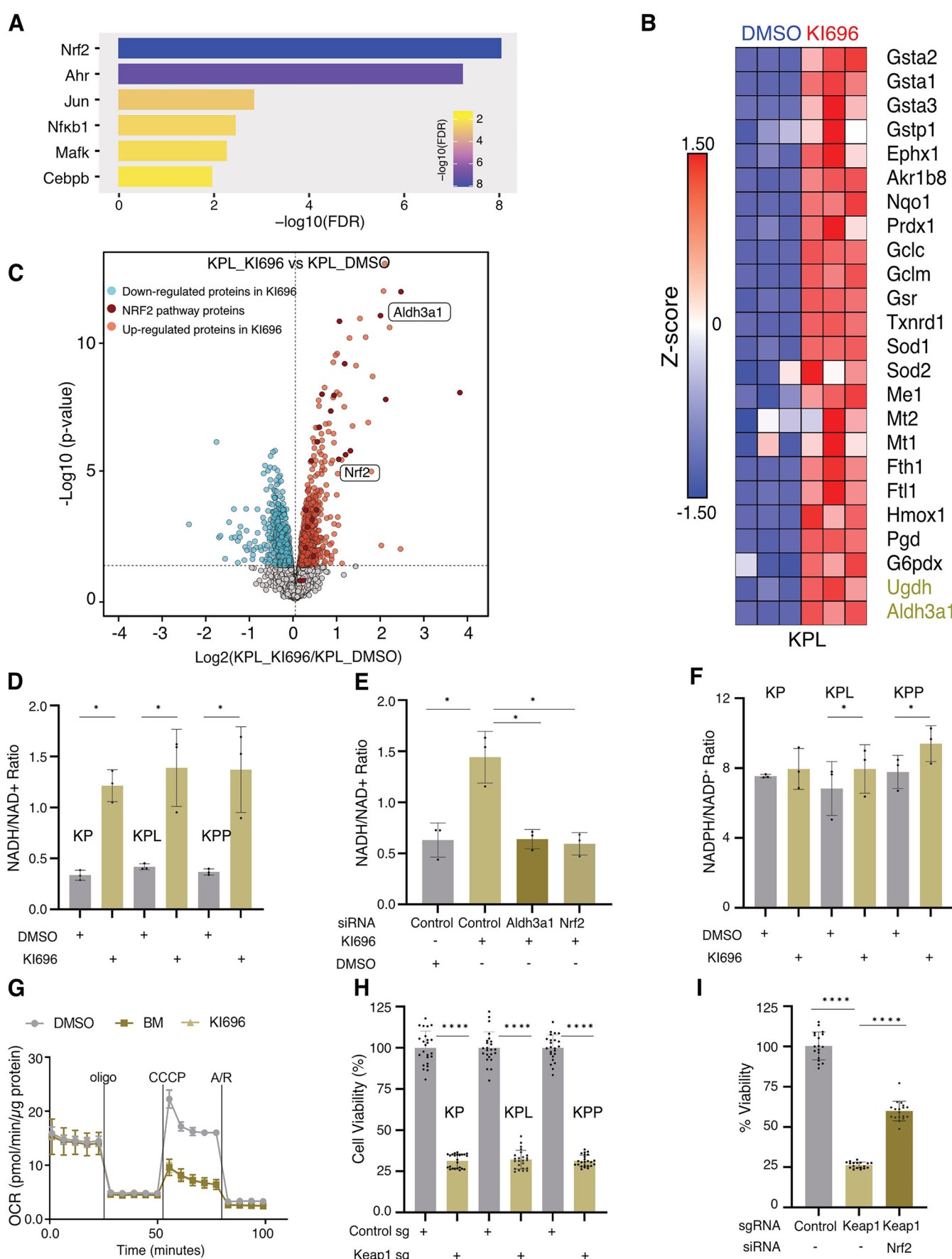

**Figure 2. KEAP1 inactivation drives NRF2–ALDH3A1–dependent NADH-reductive stress.**

(A) KPL cells were treated with KI696 (1 µM) for 24 h, and changes in gene expression were assessed by RNA-seq. Results are presented as enriched transcription factor signatures. (B) Heat map showing gene expression (z-score) of NRF2 target genes in DMSO and KI696 ($n = 3$) treated KPL cells. (C) Whole-cell proteome analysis by mass spectrometry. Volcano plot of proteomic changes in KPL cells treated with KI696 (1 µM) versus DMSO. Significantly decreased (blue) and increased (red) proteins ($p < 0.05$) are shown; dark red highlights significantly increased NRF2 pathway proteins ($n = 3$). Statistical significance was assessed by a two-sided moderated $t$-test as implemented in the limma package via FragPipe Analyst; p-values were adjusted using the Benjamini–Hochberg method. (D) NADH/NAD$^+$ ratio in KP, KPL, and KPP cells treated with DMSO or KI696 for 48 h. Data represent mean ± SD of three biological replicates. ***$P < 0.001$, **$P < 0.01$, *$P < 0.05$, ns not significant $P > 0.05$. KP (DMSO vs KI696) $P = 0.0178$, KPL (DMSO vs KI696) $P = 0.0434$, KPP (DMSO vs KI696) $P = 0.0477$. Statistical significance was calculated using a paired $t$-test. (E) NADH/NAD$^+$ ratio in KPL cells treated with control, Aldh3a1, or NRF2 siRNA, in the presence of DMSO or KI696. Data represent mean ± SD of three biological replicates. *$P < 0.05$. Control vs control + KI696 $P = 0.0113$, Control + KI696 vs Aldh3a1 si + KI696 $P = 0.0315$, Control + KI696 vs NRF2 si + KI696 $P = 0.0233$. Statistical significance was calculated using one-way ANOVA. (F) NADPH/NADP$^+$ ratio in KP, KPL, and KPP cells treated with DMSO or KI696 for 48 h. Data represent mean ± SD of three biological replicates. KPL (DMSO vs KI696) $P = 0.0374$, KPP (DMSO vs KI696) $P = 0.0033$. Statistical significance was calculated using a paired $t$-test. (G) Oxygen consumption rate (OCR) of KPL cells treated with DMSO, KI696, or Bardoxolone methyl (BM). Data represent mean ± SEM of five technical replicates. (H) Cell viability of KP, KPL, and KPP cells following Keap1 depletion by CRISPR/Cas9. Individual dots in the graph represent single wells. Data represent normalized viability from four biological replicates (mean ± SD). ***$P < 0.001$, ns not significant, $P > 0.05$. KP (Control vs Keap1 Sg) $P = 0.0001$, KPL (Control vs Keap1 Sg) $P = 0.0001$, KPP (Control vs Keap1 Sg) $P = 0.0001$. Statistical significance was calculated using a paired $t$-test. (I) Cell viability of KPP cells transduced with control sgRNA or Keap1 sgRNA, with or without NRF2 depletion by siRNA. Individual dots represent single wells. Data represent normalized viability from three biological replicates (mean ± SD). ***$P < 0.001$, ns not significant, $P > 0.05$. Control vs Keap1 Sg, $P = 0.0001$, Keap1 Sg vs Keap1 Sg + NRF2 si, $P = 0.0001$. Statistical significance was calculated using one-way ANOVA. Source data are available online for this figure.

validated our results obtained in the CRISPR screens (Fig. 3F,G). Additionally, we observed that in tumors bearing high genetic complexity, KPLP, depletion of Keap1 (KPKPL) negated tumor onset, highlighting the context-dependent role of Keap1 (Fig. 3E–G).

NSCLC tumors of the KPL and KPP genotypes, harboring STK11/LKB1 or PTEN deletions, respectively, showed an unexpected dependence on KEAP1 for proliferation (Fig. 3A,E) (Romero et al, 2017). In agreement with previous studies, Keap1 depletion in KPP and KPL cohorts resulted in a significant upregulation of Nrf2 abundance (Fig. 3C). Additionally, we observed that NRF2 protein levels negatively correlated with KEAP1, LKB1, and PTEN protein levels in lung human cancer cell lines (Fig. EV4E–G). Recently, it has been shown that distinct Nrf2 signaling thresholds mediate lung tumor initiation and progression (DeBlasi et al, 2023). Moreover, Keap1 depletion in KPP and KPL cohorts resulted in a significant upregulation of Aldh3a1 abundance (Figs. 3D and EV5C). ALDH3A1 and UGDH displayed positive correlation with NRF2, whereas a negative correlation with Keap1 protein levels in human lung cell lines was observed (Fig. EV4H–K). NSCLC tumor samples have significantly higher UGDH and ALDH3A1 transcript levels in comparison to normal tissue (Fig. EV4L). To investigate how NRF2 hyperactivation impairs tumor progression, we measured the levels of reductive stress marker Aldh3a1. In agreement with our cellular and biochemical data, inactivation of Keap1 in KPP, KPL, and KPLP mice resulted in a significant increase in Aldh3a1, highlighting NADH-reductive stress as a critical factor impairing tumor initiation. IHC analysis revealed that Nrf2 and Aldh3a1 levels were low in adjacent lung tissue compared to tumors (Fig. EV5A). Keap1 loss led to marked accumulation of Nrf2 and Aldh3a1 in tumors, while Ugdh was elevated in KP and KPL but not further increased upon Keap1 deletion. Non-transformed tissue showed no elevation of these markers irrespective of genotype, suggesting that Keap1 loss selectively enhances Nrf2–Aldh3a1 signaling in tumors without inducing reductive stress in adjacent tissue (Fig. EV5B–D).

***KEAP1–NRF2 axis stratifies lung cancer patients and predicts survival***
Further, tumor microarray (TMA) analysis of lung cancer patient samples revealed an inverse correlation between KEAP1 expression and NRF2/ALDH3A1 levels: tumors with low KEAP1 expression showed upregulation of NRF2, ALDH3A1, and UGDH, whereas tumors with high KEAP1 expression displayed low levels of NRF2, ALDH3A1, and UGDH (Fig. 4A–C). In agreement with this, and consistent with previous reports (Weiss-Sadan et al, 2023; Arolt et al, 2023; Scalera et al, 2024), comparison of transcriptomic signature in wildtype and KEAP1 (inactivation)/NFE2L2 (activation) mutant lung adenocarcinoma (LUAD) (Fig. 4D) and lung squamous cell carcinoma (LUSC) (Appendix Fig. S4A) patient samples, revealed upregulation of reductive stress markers. Interestingly, reactome pathway analysis from transcriptome reveals activation of NRF2-driven redox reprogramming in KEAP1 (inactivation)/NFE2L2 (activation) mutant LUAD (Appendix Fig. S4B,C) and LUSC (Appendix Fig. S4D,E) patient samples. Kaplan–Meier survival curves showed that the lung cancer patients with high NRF2, UGDH and ALDH3A1 expression had significantly better free progression (FP) and overall (OS) survivals (Fig. 4E,F; Appendix Fig. S4F). Altogether, we demonstrated that inactivation of KEAP1 led to reductive stress, which creates metabolic vulnerability in lung cancer (Fig. 4G).

## Multiplex CRISPR screens reveal combinatorial vulnerabilities of E3 ligases and DUBs

The KP$^{CRISPR}$-based murine model LUAD faithfully mimics human KRAS-driven LUAD, displaying similarities at the molecular and histopathological level following intratracheal administration of viral vectors (Hartmann et al, 2021). We conducted CRISPR-mediated targeting of Trp53, introduced the Kras$^{G12D}$ mutation simultaneously, and additionally co-targeted Keap1 in our murine NSCLC model- KP. Twelve weeks post-infection, animals showed equivalent tumor burden upon loss of Keap1 (Keap1$^\Delta$, referred to as KPK), despite a significant upregulation of Nrf2 abundance, detected by IHC, in Keap1-depleted tumors (Fig. 5A,B; Appendix Fig. S4G,H). This observation is consistent with previous studies showing that, in Kras$^{G12D}$; p53$^{fl/fl}$ adenocarcinoma model, genetic depletion of Keap1 unexpectedly blocked tumor progression (Rogers et al, 2018). Similarly, a homozygous Keap1 R554Q mutation also suppressed tumor progression due to Nrf2 hyperactivation (DeBlasi et al, 2023).

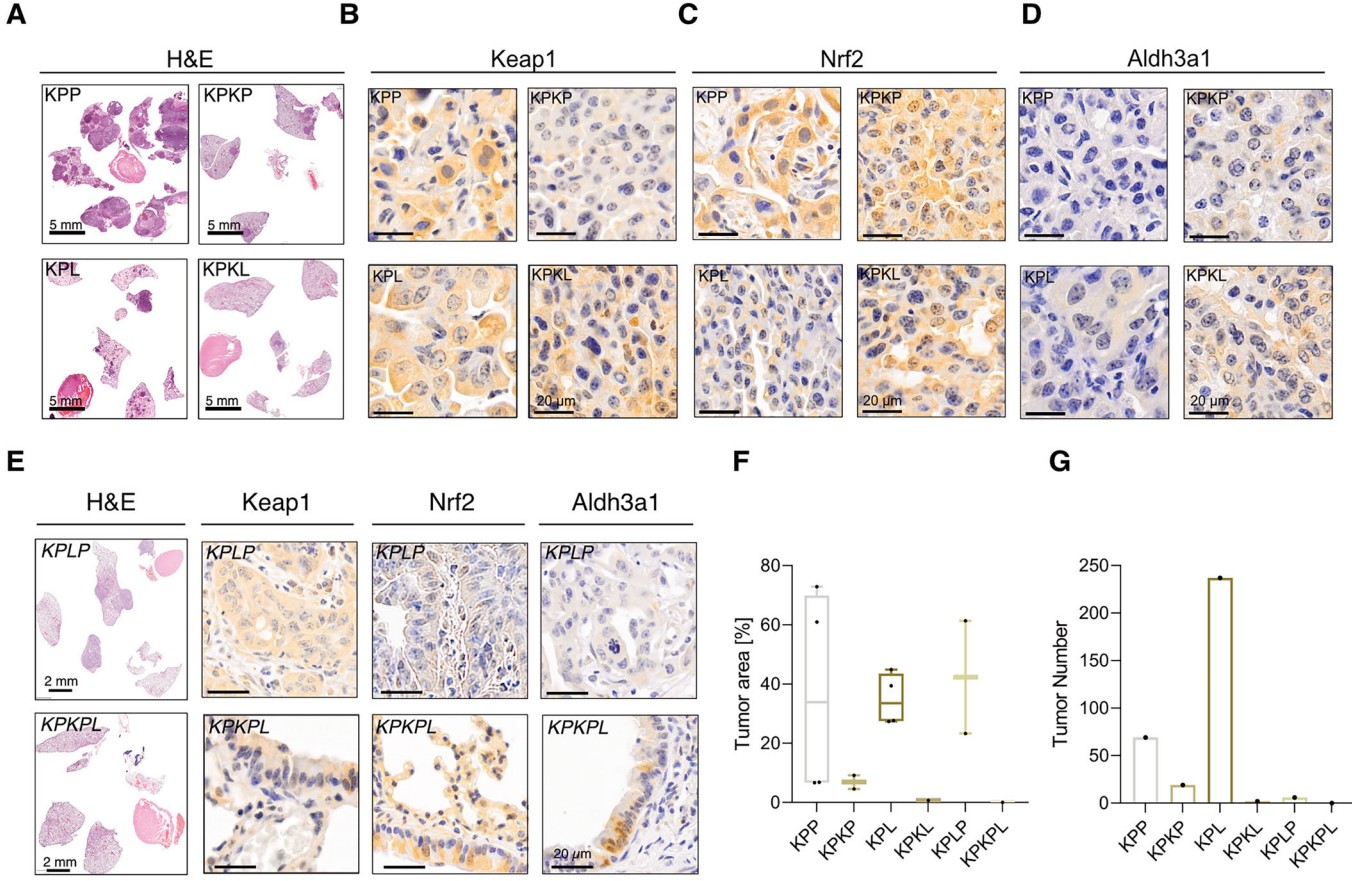

**Figure 3. Keap1 is a context-dependent vulnerability in aggressive NSCLC subsets.**

(A) Representative H&E sections of KPP (Kras$^{G12D}$:Tp53:Pten), KPL (Kras$^{G12D}$:Tp53:Lkb1), KPKP (Kras$^{G12D}$:Tp53:Keap1:Pten), and KPKL (Kras$^{G12D}$:Tp53:Keap1:Lkb1) animals 12 weeks post-intubation. H&E is used to show tumor grade, stage and morphology. Scale bar, 5 mm. (B) Representative IHC against endogenous Keap1 to demonstrate loss of Keap1 in KPKP and KPKL tumors. Scale bar, 20 μm. (C) Representative IHC of Nrf2 in KPP, KPKP, KPL and KPLP tumors. Scale bar, 20 μm. (D) Representative IHC against endogenous Aldh3a1 to demonstrate reductive stress in KPKP and KPKL tumors. Scale bar, 20 μm. (E) Representative H&E sections of KPLP (Kras$^{G12D}$:Tp53:Lkb1: Pten) and KPKPL (Kras$^{G12D}$:Tp53:Keap1:Pten:Lkb1) animals 12 weeks post-intubation. Scale bar, 2 mm/20 μm. (F, G) Quantification of tumor area and tumor burden in KPP, KPKP, KPL, KPKL, KPLP and KPKPL animals. The box plot represents the lower (25%) and upper quartile (75%) with the median (50%) displayed. The whiskers represent the minimum and maximum. ($n$[animals] = 4). Source data are available online for this figure.

Histopathological analysis of the somatic engineered murine models of NSCLC revealed that tumors arising in KPK mice displayed elevated reductive stress, which is evidenced by elevated Aldh3A1 and Ugdh expression levels (Fig. 5B). The emergence of tumors in KPK raised the possibility that additional safeguarding mechanism must be co-targeted to fully ablate tumor onset in a Keap1-dependent fashion (Fig. 5A). In order to further delineate the interplay between E3 ligases and DUBs in tumor growth and progression, and to investigate the possibility of combinatorial targeting of multiple members of the UPS system, we performed a targeted multiplex CRISPR screens. We prepared a combinatorial CRISPR library, incorporating all 185 gRNAs in each cassette (from a single E3-DUB library list). This led to a library of 34,255 distinct targeted E3-DUB combinations of NSCLC-relevant E3-DUB interactions (Appendix Figs. S5A–E and S6A). Next, we employed a multiplex library to proliferation screens in KP, KPP and KPL cells and evaluated a total of 34255 paired gRNAs in biological replicates. Individual samples across biological samples exhibited strong correlations between paired gRNAs and gene-level counts

(Appendix Fig. S5F–H). As expected, essential genes for cell fitness were depleted; hence, we focused on gene pairs that caused proliferative effects (Fig. 5C; Appendix Fig. S6B,C). Our analyses identified 592 significantly depleted and 374 enriched gene pairs in KP, 534 depleted and 441 enriched gene pairs in KPL, and 550 depleted and 205 enriched gene pairs in KPP. Among these, Keap1, Huwe1 and Ubr4 gene pairs exerted the most potent anti-proliferative effects across all three tested cell lines (Fig. 5D; Appendix Fig. S6D–F). In summary, our results reveal previously unanticipated combinatorial roles of E3 ligases and DUBs in NSCLC.

## Identification of E3 ligases as Keap1-dependent genetic co-vulnerabilities

We next transferred the in cellulo observation of Keap1 co-dependencies into an in vivo NSCLC model. In contrast to KPKP and KPKL models (Fig. 3A), the KPK animals exhibit increased reductive stress while maintaining a similar tumor burden to KP mice (Fig. 5A). To functionally dissect the role of Keap1 co-

dependencies, we selected the E3 ligases Herc2 (HECT and RLD domain-containing E3 ubiquitin protein ligase 2), Ubr4, Pdzrn3 (PDZ domain-containing RING finger protein 3), and Huwe1, along with the DUB USP28 (Fig. 5D). USP28 has been previously demonstrated to contribute to metabolic stress, mitochondrial bioenergetics, and is considered a vulnerability factor in lung cancer (Xie et al, 2024; Ruiz et al, 2021; Prieto-Garcia et al, 2020). To in vivo address the potential of dual targeting we employed a CRISPR-dependent murine model of NSCLC; to this end, one adeno-associated virus encoding sgRNA for Kras (and a repair template Kras$^{G12D}$), an sgRNA against Tp53 (KP) and to target Keap1 (KPK), respectively, while the second virus encoded for the identified UPS enzyme an sgRNA targeting the identified co-vulnerability gene, such as Herc2, along with SaCas9 (Appendix Fig. S7A). Remarkably, co-depletion of Keap1 and Herc2 abolished NSCLC formation, implying that Herc2 deficiency enhanced tumor sensitivity to Keap1 depletion (Fig. 6A). The number and the size of lung nodules remained noticeably high in KPK mice, while Herc2 loss entirely ablated tumor onset in our experimental cohort of KPK, as revealed by the relatively healthy appearance of the lung (Fig. 6B,C). KPK tumor samples exhibited increased Herc2 and Pdzrn3 protein levels compared to KP mice, underlining the importance of Herc2 and Pdzrn3 in the maintenance and progression of KPK (Appendix Fig. S7B). Relative to non-transformed tissue, all patient samples from different lung tumor subtypes (LUAD and LUSC) expressed elevated levels of Herc2, Ubr4, and Huwe1 (Appendix Fig. S7C–E). Similar detrimental observations with regard to tumor onset and propagation were made in mice infected with virions encoding for Ubr4, Pdzrn3, Huwe1 or Usp28, respectively (Fig. 6A–C). These observations stress that solid tumors of the respiratory tract rely on intricate protein stability regulatory networks. Notably, deletion of Usp28 reduced lung tumor burden in KPK animals (Fig. 6B,C), as well as in KP animals, as previously reported (Prieto-Garcia et al, 2020). Interestingly, UBR4 has been recently reported as a regulator of mitochondrial homeostasis (Haakonsen et al, 2024). In line with that, it may play a role in the response to reductive stress triggered by Keap1 inactivation. In agreement with mice data, Kaplan–Meier curves showed that lung cancer patients with low UBR4, HERC2, and HUWE1 expression with concurrent low TP53 and KEAP1 levels had significantly better first progression (FP) and overall survival (OS) (Fig. 6D–F; Appendix Fig. S7F–H).

Altogether, we uncovered a prevalent addiction of Keap1-inactivated or Nrf2-hyperactivated tumors towards reductive stress (Fig. 6G). These findings will prompt further development of new therapeutic strategies for patients with KEAP1-inactivated tumors that are refractory to the standard-of-care combination of chemotherapy and immunotherapy.

# Discussion

The genomic complexity of NSCLC involves numerous alterations in E3 ligases and DUBs, making it difficult to understand their functional impact on tumor growth and response to therapy. By employing the unique combinations of single and combinatorial CRISPR screens in a range of representative and genetically defined tumor models, we revealed that targeting the UPS represents a potential therapeutic approach for patients with KEAP1-inactivated

tumors. We demonstrate that Keap1 has a context-dependent vulnerability in a subset of aggressive NSCLC. Furthermore, we show that alterations in the ubiquitin signaling cascade deregulate reductive stress, leading to detrimental hyperactivation of NRF2 for tumor cell growth. Our study provides insights into the complex interplay between Keap1 and frequently altered E3 ubiquitin ligases in KRAS-TP53-driven lung cancer tumorigenesis and thus has significant clinical implications.

Although KEAP1 mutations are frequently discussed as a common feature of NSCLC, our data demonstrate that the functional consequences of KEAP1 loss and the resulting therapeutic vulnerabilities are strongly shaped by genetic context. In murine models, KEAP1 appears to function as a proto-oncogenic dependency specifically in the setting of TP53 loss in combination with KRAS$^{G12D}$, as evidenced by the profound suppression of tumor initiation in KPLK and KPKP mice. In contrast, human LUAD comprises both KPL and KLK subtypes, with KLK tumors occurring nearly as frequently as KPL tumors, and clinical studies indicate that co-mutation of KEAP1 and STK11/LKB1 defines a highly aggressive and therapy-resistant disease subtype (Galan-Cobo et al, 2025; Wohlhieter et al, 2020; Shen et al, 2019). The inability of KEAP1/STK11 mutant murine models to initiate or sustain tumor growth may therefore reflect differences in tumor initiation, mutational timing, selective pressures, or species-specific tumor biology. Notably, TP53 loss may act as a critical determinant that enables tolerance to KEAP1 dysfunction by relaxing genomic or proteostatic constraints, while rarer events such as sequential mutation acquisition or modulation by additional bystander alterations may underlie the emergence of human KPLK tumors that are not captured by the murine systems used here.

The role of KEAP1 loss in lung tumorigenesis appears highly context-dependent. Diaz-Jimenez et al reported that in EML-ALK4-driven NSCLC (TP53-wild-type context), KEAP1 loss can be either tumor burden neutral or might partially suppress tumor formation in a KRAS-independent manner (Diaz-Jimenez et al, 2026). Further, the Tp53$^{fl/fl}$:Tet-On-EGFR$^{L858R}$ mouse model of NSCLC, CRISPR-mediated loss of KEAP1 was tumor burden neutral, whereas CRISPR-mediated Keap1 loss in a genetic KP model reduced tumor burden (Foggetti et al 2021). These findings, together with those of Rogers et al., are consistent with our observations and support a highly context-dependent role of KEAP1 (Rogers et al, 2018; Foggetti et al, 2021). Importantly, the same study also reported that KEAP1 mutations can promote tumor growth when EGFR-driven tumors are treated with RTK inhibitors. Future studies will be important to determine whether the synthetic lethality of E3 ligases with KEAP1 loss extends to tumors driven by other oncogenic drivers and whether these vulnerabilities can be therapeutically exploited.

Notably, NSCLC patients harboring Keap1 mutations are presented with limited therapeutic options, and exhibit high therapy failure, especially towards immune checkpoint inhibitors targeting PD-1/PD-L1 are frequently reported. This results in reduced survival due to the immune-evasive nature of Keap1-mutated tumors (Fox et al, 2023). In light of these clinical challenges, our work adds several new insights; firstly, we uncovered an unprecedented synthetic lethality by co-deletion of discrete members of the ubiquitin signaling pathway and deactivating Keap1. Secondly, our work opens the possibility to

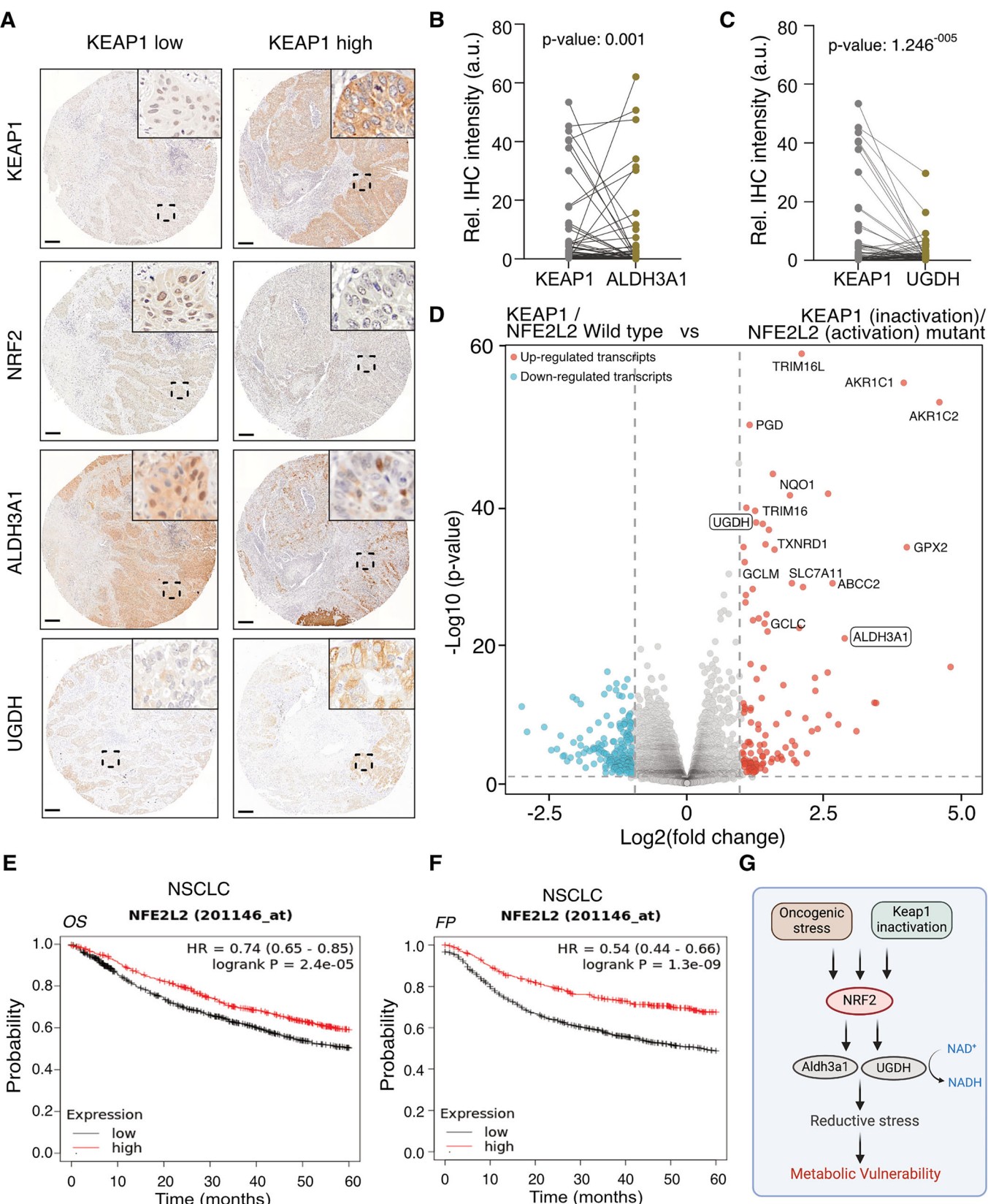

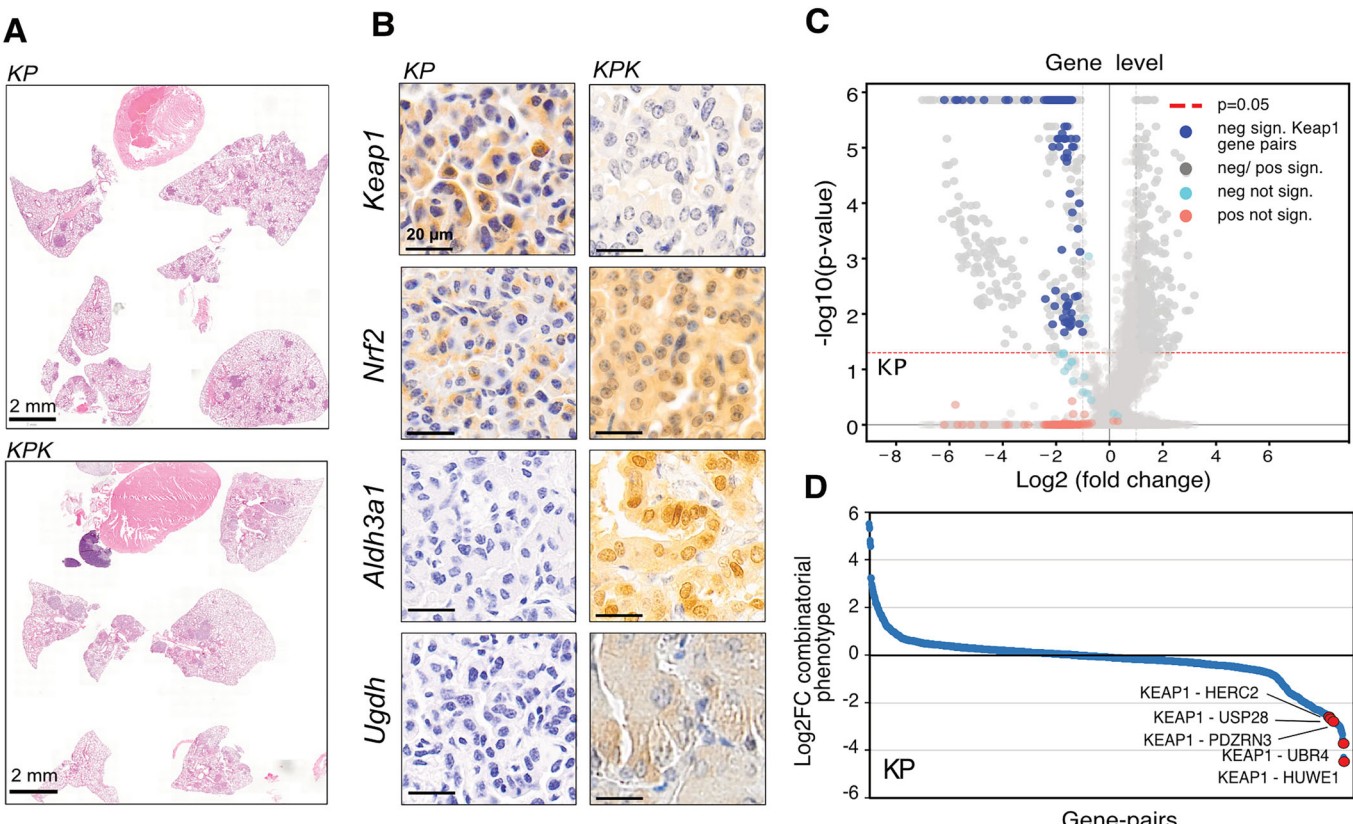

**Figure 4. KEAP1 loss links to NRF2/ALDH3A1 upregulation and survival in NSCLC patient samples.**

(A) Representative images of individual tissue cores of a patient's NSCLC-derived tissue microarray, showing immunohistologic staining of endogenous KEAP1, NFE2L2/NRF2 and ALDH3A1. Scale bar, 200 μm. (B, C) Quantification of staining of endogenous KEAP1, ALDH3A1, and UGDH from (A). Units are defined as arbitrary units. $N = 72$ cores. P value = 0.001 (B) and $1.246^{-005}$ (C). (D) Volcano plot displaying transcriptomic signature in wildtype and KEAP1 (inactivation)/NFE2L2 (activation) mutant LUAD patient samples, ($n = 398$, KEAP1/NFE2L2 WT; $n = 107$, KEAP1/NFE2L2 mutant). Data were obtained from Arolt et al, 2023. (E, F) Kaplan–Meier curves of five-year overall survival (OS) (E) and first progression (FP) (F) rates of NSCLC patients based on NRF2. Data were obtained from kmplot.com. (G) Schematic model—KEAP1 inactivation or oncogenic stress hyperactivates NRF2, inducing Aldh3a1 and Ugdh expression. Aldh3a1 elevates NADH/NAD⁺ ratios, driving reductive stress that creates a metabolic vulnerability and impairs cell proliferation. Source data are available online for this figure.

**Figure 5. Multiplex CRISPR screens reveal combinatorial role of E3 ligases and DUBs.**

(A) Representative H&E sections of mice 12 weeks post intratracheal intubation with various genetic combinations: KP (Kras[G12D]:Tp53); KPK (Kras[G12D]:Tp53:Keap1). Scale bar, 2 mm. (B) Representative IHC of Keap1, Nrf2, Aldh3a1, and Ugdh in KP and KPK tumors. Scale bar, 20 μm. (C) Volcano plots of MAGeCK-derived LFCs and p values for the multiplex dropout viability screen. FDR <20% for positive and negative selections are color-coded gray. Keap1-dependent gene pairs are highlighted in blue. Significant ($p < 0.05$) data points with LFC >1 or LFC <−1 have dashed strokes. Data acquired over two replicates. Hit scoring performed according to MAGeCK's robust rank aggregation (RRA) test for gene-level enrichment/depletion. (D) sgRNA log fold-change data for gene pairs plotted based on combinatorial phenotypes in the KP cell line. Source data are available online for this figure.

revisit KEAP1 as an exploitable target in KEAP1 wild-type patients, as its genetic loss reduces overall tumor burden. In this context, targeted KEAP1 inhibition could disrupt tumor metabolic signaling to a level intolerable for transformed cells. And lastly, it offers insights for stratifying patients likely to respond to UPS inhibitors based on Keap1 mutation or NRF2 hyperactivation in KRAS-TP53 driven lung NSCLC.

Deregulation of the KEAP1/NRF2 axis has been reported to alter metabolic requirements, rendering the KRAS-driven lung tumors more sensitive to glutamine metabolism inhibitors, mitochondrial complex I inhibitors and ATM inhibitors (Weiss-Sadan et al, 2023; Li et al, 2023a; Pillai et al, 2024). Our findings suggest that the

inactivation of selective E3 ligases is sufficient to sensitize lung tumors addicted to Keap1 depletion *in cellulo* and in vivo. It has been demonstrated that HERC2-deficient cells activate transcription of oxidative stress response genes such as NRF2, SOD1, and GPX1, which predisposes cells to an enhanced resistance to oxidative stress (Sala-Gaston et al, 2022). Thus, cells with inactive KEAP1 are selectively vulnerable to Herc2 depletion. Moreover, KRAS and PTEN have also been reported to transcriptionally upregulate NRF2 levels and confer chemoresistance (Rojo et al, 2014; Tao et al, 2014). These results indicate that there is an optimal threshold of NRF2 activity, and that excess NRF2 activation can impair lung cancer phenotypes (DeBlasi et al,

2023). Interestingly, UBR4 has been recently reported as a regulator of mitochondrial homeostasis, and in agreement, it may play a role in the response to reductive stress triggered by Keap1 inactivation (Haakonsen et al, 2024). Given the biological function of Keap1, we anticipate that the depletion of Keap1 and its co-dependency compromises essential safeguarding mechanisms and leads to redox imbalance, mitochondrial dysfunction, and DNA damage, which are detrimental to the growth of NSCLC. These results align with prior research indicating that NRF2 hyperactivation can present synthetic lethal opportunities, and that targeting the ubiquitin signaling pathway represents a therapeutic vulnerability in NSCLC.

Emerging UPS modulators, such as PROTACs (proteolysis-targeting chimeras) and molecular glues, which induce protein degradation via E3 ligase-mediated ubiquitination, show promising potential as a novel pharmacological approach (Tsai et al, 2024). Multiple PROTAC molecules have been developed for validated therapeutic targets in NSCLC, such as EGFR, KRAS, ALK, BRAF, and BCL-XL, showing antitumor efficacy in cell models and preclinical lung tumor models (Li et al, 2023b). However, degraders that have reached clinical evaluation or approval recruit ubiquitously expressed CRBN and VHL E3 ligases, which have the potential to cause on-target toxicity. In addition, recent data suggest that resistance can emerge due to the non-essential nature of the CRBN, potentially limiting its applicability (Tsai et al, 2024). Our results reveal the identification of essential E3 ligases in NSCLC that can be hijacked to design the next generation of modalities. In this way, cancer cells are less likely to develop resistance, since losing an essential E3 ligase would compromise their survival. Nevertheless, exploring such modalities in treating NSCLC patients using Keap1 dependencies warrants future interrogation.

Our work, furthermore, identified several E3 ligases for which their therapeutic potential in NSCLC, and especially in the context of KEAP1 mutation, has not been reported yet. These enzymes, exemplified by PDZRN3, HERC2, UBR4, and HUWE1, warrant further investigation. Given the observed dependency, particularly in the context of Keap1 inactivation, these enzymes could hold the potential for selective targeting of solid tumors such as NSCLC, and expand the repertoire of employable degrader E3 ligases for drug development. While cancer cells exhibit common metabolic reprogramming hallmarks, our results underscore how the influence of the ubiquitin network uniquely shapes the extent and direction of metabolic reprogramming. The new paradigm of incorporating combinatorial CRISPR screening and autochthonous mouse modeling presented here will provide a new platform to unveil new therapeutic targets, empowering clinicians to better identify responsive patient populations based on genetic context.

# Methods

### Reagents and tools table

| Reagent/resource | Reference or source | Identifier or Catalog Number |
| --- | --- | --- |
| **Experimental models** | | |
| B6J.129(Cg)-Gt(ROSA)26Sor<sup>tm1.1(CAG-cas9*,-EGFP)Fezh</sup>/J | Jax Repository | IMSR_JAX:026179 |
| KP cells | This work | |
| KPL cells | This work | |

| Reagent/resource | Reference or source | Identifier or Catalog Number |
| --- | --- | --- |
| KPP cells | This work | |
| NCI-H2170 (H2170) | ATCC | CRL-5928 |
| A549 | ATCC | CCL-185 |
| CALU1 | ATCC | HTB-54 |
| NCI-H1299 (H1299) | ATCC | CRL-5803 |
| **Recombinant DNA** | | |
| pAAV-DJ Vector | Cell Biolabs, INC. | VPK-420-DJ |
| pAdDeltaF6 | Cell Biolabs, INC. | |
| AAV:ITR-U6-sgRNA(Kras)-U6-sgRNA(p53)-pEFS-2A-mCherry-shortPA-KrasG12D-HDRdonor-ITR | https://doi.org/10.15252/emmm.201911101 | N/A |
| AAV:ITR-U6-sgRNA(Kras)-U6-sgRNA(p53)-U6-sgRNA(Pten)-pEFS-2A-mCherry-shortPA-KrasG12D_HDRdonor-ITR | This work | N/A |
| AAV:ITR-U6-sgRNA(Kras)-U6-sgRNA(p53)-U6-sgRNA(Lkb1)- pEFS-2A-mCherry-shortPA-KrasG12D_HDRdonor-ITR | This work | N/A |
| AAV:ITR-U6-sgRNA(Kras)-U6-sgRNA(p53)-U6-sgRNA(Keap1)- pEFS-2A-mCherry-shortPA-KrasG12D_HDRdonor-ITR | This work | N/A |
| AAV:ITR-U6-sgRNA(Kras)-U6-sgRNA(p53)-U6-sgRNA(Lkb1)-U6-sgRNA(Keap1)-pEFS-2A-mCherry-shortPA-KrasG12D_HDRdonor-ITR | This work | N/A |
| AAV:ITR-U6-sgRNA(Kras)-U6-sgRNA(p53)-U6-sgRNA(Lkb1)-U6-sgRNA(PTEN)-pEFS-2A-mCherry-shortPA-KrasG12D_HDRdonor-ITR | This work | N/A |
| AAV:ITR-U6-sgRNA(Kras)-U6-sgRNA(p53)-U6-sgRNA(Keap1)-U6-sgRNA(PTEN)-pEFS-2A-mCherry-shortPA-KrasG12D_HDRdonor-ITR | This work | N/A |
| AAV:ITR-U6-sgRNA(Kras)-U6-sgRNA(p53)-U6-sgRNA(Lkb1)-U6-sgRNA(Keap1)- U6-sgRNA(Pten)-pEFS-2A-mCherry-shortPA-KrasG12D_HDRdonor-ITR | This work | N/A |
| **Antibodies** | | |
| Keap1 | Proteintech Europe / PTGlab | 10503-2-AP, AB_2132625 |
| Lkb1/Stk11 | Proteintech Europe / PTGlab | 10746-1-ap, AB_2271311 |
| Pten | Proteintech Europe / PTGlab | 10047-1-AP, AB_2174343 |
| NFE2l2/Nrf2 | Invitrogen | PA5-27882, AB_2545358 |
| ALDH3A1 | Sigma | HPA051150, AB_2681364 |
| UGDH | Sigma | HPA036656, AB_10671706 |
| HERC2 | MyBioSource | MBS8109709 |
| PDZRN3 | Aviva Systems Biology | OACA09962 |
| **Oligonucleotides and other sequence-based reagents** | | |
| sgRNA murine KEAP1 for | CACCGCGCCCGCTGTGTAGATGAGG | Sigma |
| sgRNA murine KEAP1 rev | AAACCCTCATCTACACAGCGGGCGC | Sigma |
| sgRNA murine Pten 1 for | CACCGTGTGCATATTTATTGCATCG | Sigma |
| sgRNA murine Pten 1 rev | AAACCGATGCAATAAATATGCACAC | Sigma |
| sgRNA murine KRas #1 for | CACCGACTGAGTATAAACTTGTGG | Sigma |

| Reagent/resource | Reference or source | Identifier or Catalog Number |
|---|---|---|
| sgRNA murine KRas #1 rev | AAACCCACAAGTTT ATACTCAGTC | Sigma |
| sgRNA murine Trp53 #1 for | CACCGATGGTGGTATA CTCAGAGC | Sigma |
| sgRNA murine Trp53 #1 rev | AAACGCTCTGAGTATA CCACCATC | Sigma |
| sgRNA murine Stk11/Lkb1 for | CACCGCGAGACCTTAT GCCGCAGGG | Sigma |
| sgRNA murine Stk11/Lkb1 rev | AAACCCCTGCGGCATA AGGTCTCGC | Sigma |
| siRNA murine Nfe2l2 | ACUCAAAUCCCACC UUAAA | Dharmacon -L-040766-00-0005 |
| siRNA murine Nfe2l2 | UGGAGUAAGUCGA GAAGUG | Dharmacon -L-040766-00-0005 |
| siRNA murine Nfe2l2 | CAUGUUACGUGAU GAGGAU | Dharmacon - L-040766-00-0005 |
| siRNA murine Nfe2l2 | GGACAGCAAUUA CCAUUUU | Dharmacon - L-040766-00-0005 |
| siRNA murine Aldh3a1 | GGGAUCAGCCU UCACGAUA | Dharmacon - L-065465-01-0005 |
| siRNA murine Aldh3a1 | GGACGAGCCUG UCGGAAAG | Dharmacon - L-065465-01-0005 |
| siRNA murine Aldh3a1 | GCCUCUAACCUG CGCAAGA | Dharmacon - L-065465-01-0005 |
| siRNA murine Aldh3a1 | GCAGAGACAUCA AGCGGUG | Dharmacon - L-065465-01-0005 |
| **Chemicals, Enzymes and other reagents** | | |
| DMSO | ROTH | 4720.4 |
| KI696 | MedChem Express | HY-101140 |
| Bardoxolone Methyl | MedChem Express | HY-13324 |
| Omaveloxolone | Hycultec | HY-12212 |
| Formaldehyde | ROTH | 7398.1 |
| Propidium idoide | ROTH | CN74.1 |
| Hoechst 33342 | TargetMol | T5840 |
| Lipofectamine™ RNAiMAX | Invitrogen/Thermo Fisher Scientific | 13778150 |
| RNAzol® RT | Sigma-Aldrich | R4533-50ML |
| Dodecyltrimethylammoniumbromid (DTAB) | Sigma | D8638 |
| Trizma® Base | Sigma-Aldrich | T1503 |
| Acetonitrile | Sigma-Aldrich | 34851 |
| BSA | Carl Roth | 8076.2 |
| Chloroform | VWRchemical | 67-66-3 |
| Complete™, Mini, EDTA-free Protease Inhibitor Cocktail | Roche | 4693159001 |
| DTT | Carl Roth | 6908.2 |
| EPPS | Sigma | E9502 |
| Methanol | Roth | AE71.1 |
| SDS | Carl Roth | CN30.3 |
| Sequencing Grade Modified Trypsin | Promega | V511C |
| TCEP | Sigma | 646547 |
| TMTpro 18plex reagent | Thermo Fisher Scientific | A52045 |
| TMT 10plex reagent | Thermo Fisher Scientific | 90111 |
| Hydroxylamine | Sigma | 467804 |
| Tris-(hydroxymethyl)-aminomethane (TRIS) | Carl Roth | 4855.3 |
| Triton X-100 | Carl Roth | 3051.2 |
| Trypsin EDTA solution | Pan Biotech | P10-023100 |

| Reagent/resource | Reference or source | Identifier or Catalog Number |
|---|---|---|
| Tween 20 | Roth | 9127.2 |
| Urea | Applichem | A1049.1000 |
| **Software** | | |
| GraphPad Prism, version 8 for windows | GraphPad Software, La Jolla California, USA | http://www.graphpad.com/ |
| FragPipe v21.1 | FragPipe | https://fragpipe.nesvilab.org/ |
| RStudio, version 2023.09.1 | RStudio: Integrated Development for R. RStudio, Inc. | https://www.rstudio.com/ |
| HCS Navigator ™ version 6.6.4 (build 8616) | Thermo Fisher | |
| QuPath software | v.0.6.0 | https://qupath.github.io/ |
| ImageJ | 2.17.0 | https://imagej.net/ij/download.html |
| **Other** | | |
| CellTiter-Glo® Luminescent Cell Viability Assay | Promega | G7570 |
| ROS-Glo™ H₂O₂ Assay | Promega | G8820 |
| NADP/NADPH-Glo™ Assay | Promega | G9081 |
| NAD/NADH-Glo™ Assay | Promega | G9071 |
| Seahorse XFe96/XF Pro FluxPak Mini. | Agilent | 103793-100 |

## Lentiviral CRISPR-Cas9 library

The UPS system CRISPR-KO library was generated using the covalently-closed-circular-synthesized (3Cs) technology, as previously described (Wegner et al, 2019; Diehl et al, 2021). The single library contained 185 gRNAs cloned under the U6 promoter, and the multiplex library contained 34,255 gene pairs in a modified pLentiCRISPRv2-puromycin vector containing a modified gRNA scaffold sequence starting with GTTTG. Each gene was represented by 4 gRNAs selected with the Azimuth 2.0 (GPP sgRNA Designer, PMID: 26780180). Besides 152 gRNAs targeting 38 UPS genes, the library also included 16 gRNAs targeting four essential genes and 17 non-targeting sequences as controls.

## Cells

The murine lung carcinoma cell lines KP, KPP and KPL were derived as described in Fig. 1A and cultured in high-glucose Dulbecco's Modified Eagle's Medium (DMEM) supplemented with 10% FBS and 1% penicillin/streptomycin.

## CRISPR screen

KP, KPP, and KPL cells were transduced with the UPS system lentiviral CRISPR-Cas9 library at an MOI of 0.5 and a coverage of 1000x (single library) or 60x (multiplex library). After lentiviral transduction, cells were collected on day 2 (for reference time point) and selected with puromycin (1 µg/ml). Upon infection and subsequent selection, cells were cultured for 14 days (about ten divisions), followed by cell harvesting, genomic DNA extraction and subsequent NGS sequencing and data analysis.

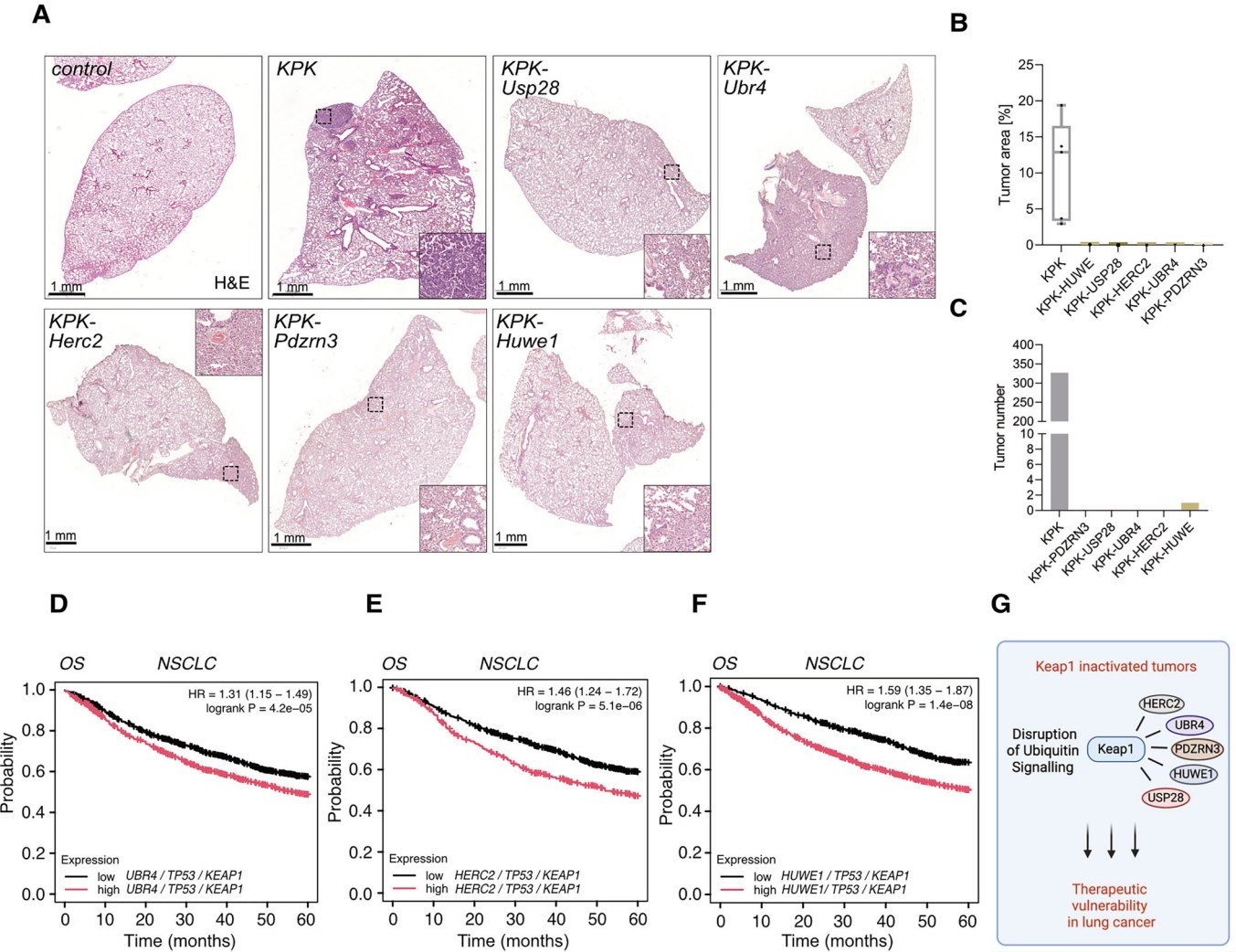

**Figure 6.    Identification of E3 ubiquitin ligases and DUBs as Keap1-dependent genetic co-vulnerabilities.**

(A) Representative H&E images of tumor-bearing animals 12 weeks post intratracheal infection with specified genotypes. Scale bar, 1 mm. (B, C) Quantification of tumor area and tumor burden in KPK, KPK-Usp28, KPK-Ubr4, KPK-Herc2, KPK-Pdzrn3, and KPK-Huwe1. Quantification of tumor area and tumor burden in KPK, KPK-Usp28, KPK-Ubr4, KPK-Herc2, KPK-Pdzrn3, and KPK-Huwe1. (n [animals] = 3). The box plot represents the lower (25%) and upper quartile (75%) with the median (50%) displayed. The whiskers represent the minimum and maximum. (D–F) Kaplan–Meier curves of 5-year overall survival (OS) rates of NSCLC patients based on TP53, KEAP1, with UBR4 (D), HERC2 (E), and HUWE1 (F) gene expression levels. Kmplot.com. (G) Schematic model—In KEAP1-inactivated tumors, targeting E3 ubiquitin ligase co-dependencies induces redox imbalance and mitochondrial dysfunction, impairing NSCLC growth. This highlights the ubiquitin signaling pathway as a therapeutic vulnerability. Source data are available online for this figure.

## Sample preparation for NGS

Cells were trypsinized, washed three times with 1xPBS, followed by genomic DNA isolation. For NGS PCR, 75 ml Next High-Fidelity 2x PCR Master Mix (New England Biolabs) was mixed with 7.5 ml of each 10 mM PCR primer and 3 mg genomic DNA. The reverse primers contained Illumina adapters and barcodes for multiplexed sequencing. PCR conditions were as follows: initial denaturation at 98 °C for 3 min, followed by 30 cycles of denaturation at 98 °C for 55 s, annealing at 64 °C for 55 s and extension at 72 °C for 110 s, with final extension at 72 °C for 7 min. PCR products were purified from a 3% TAE/agarose gel using GeneJET Gel Extraction Kit (Thermo Fisher Scientific) and additionally concentrated by using

DNA Clean & Concentrator-5 kit (Zymo Research) according to the manufacturer's protocol. PCR product concentration was determined with a Qubit 3.0 fluorometer (Thermo Fisher Scientific).

## NGS

Purified concentrated PCR products were diluted and denatured according to the Illumina MiSeq kit instructions, to a final concentration of 2.6 pM in a final volume of 2.2 ml and 15% PhiX control. Sequencing was performed with paired-end reads (150 cycles, eight cycles of index reading) on a MiSeq (Illumina) machine.

## Sequencing data quality control and read count table generation

Raw sequencing data were demultiplexed with bcl2fastq v2.20.0.422 (Illumina) to generate raw fastq files. To determine the abundance of individual gRNAs per sample, the fastq files were trimmed using cutadapt 2.8 to retain only the putative gRNA sequences. These sequences were then aligned to the original gRNA library with Bowtie 2.3.0, and only perfect matches were counted.

## Enrichment and depletion analyses

All MAGeCK depletion and enrichment analyses were performed with median or total normalization of read counts. gRNAs with zero counts in the control samples were removed (Li et al, 2014). Genes with an LFC >1 or <−1 and a *p* value <0.05, according to the MAGeCK analysis, were considered to be significantly depleted or enriched for CRISPR screens.

## siRNA transfection

SMARTpool siRNAs targeting mouse Nfe2l2, Keap1, and Aldh3a1 were obtained from Dharmacon (ON TARGETplus; Horizon Discovery). A non-targeting siRNA (Qiagen) was used as a control. Cells were transfected 24 h after seeding with 20 nM siRNA using Lipofectamine RNAiMAX (Thermo Fisher Scientific) following the manufacturer's instructions. Cells were harvested, and downstream assays were conducted 72 h post-transfection.

## Mass spectrometry sample preparation

KP, KPL, or KPP cells were treated with DMSO or KI696 (1 µM) for 24 h. Cells were harvested and washed with PBS by centrifugation. Further, Lysis buffer (2% SDS, 50 mM Tris, pH 8.5, 10 mM TCEP, 40 mM CAA, supplemented with protease inhibitor cocktail) was added to the pellets and cells were homogenized by sonication in ice and boiling at 95 °C for 10 min. Proteins were precipitated using methanol-chloroform and resuspended in 8 M urea and 50 mM Tris, pH 8.5. The Bradford assay was used to determine the final protein concentration in the lysate. Fifty micrograms of proteins were digested with 1:50 w/w LysC (Wako Chemicals, cleaves at the carboxylic side of lysine residue) and 1:100 w/w trypsin (Promega, Sequencing-grade) overnight at 37 °C after dilution to a final urea concentration of 1 M using 50 mM Tris, pH 8.5. Digested peptides were then acidified (pH 2–3) using trifluoroacetic acid (TFA) and purified using C18 SepPak columns (Waters). Desalted peptides were dried and resuspended in TMT-labeling buffer (200 mM EPPS, pH 8.2, 20% acetonitrile). Ten micrograms of peptides per condition were subjected to TMT labeling with a 1:2.5 peptide TMT ratio (w/w) for 1 h at room temperature. The labeling reaction was quenched by the addition of hydroxylamine to a final concentration of 0.5% and incubation at room temperature for 15 min. Successful TMT labeling was verified by mixing equimolar ratios of peptides and subjecting the mix to single-shot LC-MS/MS analysis. For high pH reversed phase fractionation on a Dionex analytical HPLC, 50 µg of pooled and purified TMT labeled samples were resuspended in 10 mM ammonium-bicarbonate (ABC), 5%ACN, and separated on a 250 mm long C18 column (Aeris Peptide XB-C18, 4.6 mm ID,

2.6 µm particle size; Phenomenex) using a multistep gradient from 100% Solvent A (5% ACN, 10 mM ABC in water) to 60% Solvent B (90% ACN, 10 mM ABC in water) over 70 min. Eluting peptides were collected every 45 s into a total of 96 fractions, which were cross-concatenated into 24 fractions and dried in a vacuum concentrator and resuspended in 3% ACN, 0.1% TFA for LC-MS analysis.

## Mass spectrometry data acquisition (proteome)

Tryptic peptides were analyzed on an Orbitrap Lumos coupled to an easy nLC1200 (Thermo Fisher Scientific) using a 35 cm long, 75 µm ID fused-silica column packed in-house with 1.9 µm C18 particles (Reprosil pur, Dr. Maisch), and kept at 50 °C using an integrated column oven (Sonation). HPLC solvents consisted of 0.1% Formic acid in water (Buffer A) and 0.1% Formic acid, 80% Acetonitrile in water (Buffer B). Assuming equal amounts in each fraction, 400 ng of peptides were eluted by a non-linear gradient from 7 to 40% B over 90 min, followed by a step-wise increase to 90%B in 6 min, which was held for another 9 min. A synchronous precursor selection (SPS) multi-notch MS3 method was used to minimize ratio compression as previously described (PMID: 24927332). Full-scan MS spectra (350–1400 m/z) were acquired at a resolution of 120,000 at m/z 200, a maximum injection time of 100 ms, and an AGC target value of $4 \times 10^5$. The most intense precursors with charge state between 2 and 6 were selected for fragmentation ("Top Speed" with a cycle time of 1.5 s) and isolated with a quadrupole isolation window of 0.7 Th. MS2 scans were performed in the Ion trap (Turbo) using a maximum injection time of 50 ms, AGC target value of $1.5 \times 10^4$ and fragmented using CID with a normalized collision energy (NCE) of 35%. SPS-MS3 scans for quantification were performed on the ten most intense MS2 fragment ions with an isolation window of 0.7 Th (MS) and 2 m/z (MS2). Ions were fragmented using HCD with an NCE of 65% (TMTclassic)/ 50% (TMTpro) and analysed in the Orbitrap with a resolution of 50,000 at m/z 200, scan range of 100–500 m/z, AGC target value of $1.5 \times 10^5$ and a maximum injection time of 86 ms. Repeated sequencing of already acquired precursors was limited by setting a dynamic exclusion of 60 s and 7 ppm, and advanced peak determination was deactivated. All spectra were acquired in centroid mode.

## Mass spectrometry data analysis (proteome)

MS raw data were analyzed using FragPipe v21.1, with MSFragger v.4.0 (PMID: 28394336) and Philosopher v.5.1.0 (PMID: 32669682). The built-in workflows "TMT10-MS3" and "TMT16-MS3" were used with a precursor mass tolerance of 20 ppm and fragment mass tolerance of 20 ppm. The human proteome database used by FP (ID: UP000005640, 09/03/2024) comprised 20,468 reviewed sequences only and their corresponding decoys, including common contaminant proteins. Identifications were filtered to obtain false discovery rates (FDR) below 1% for both peptide spectrum matches (minimum peptide length of 7) and proteins using a target-decoy strategy. For all searches, carbamidomethylated cysteine was set as a fixed modification and oxidation of methionine and N-terminal protein acetylation as variable modifications with allowing up to 3 modifications per peptide. Strict trypsin cleavage was set as the protein digestion rule. Label-

free quantification was performed using IonQuant v.1.10.27 (PMID: 33813065). Data were further processed using FragPipe Analyst (PMID: 38496650). Subsequently, the data were plotted in R using custom scripts.

## RNA-Seq

RNA was extracted from the samples following the RNAzol® RT extraction protocol (Sigma-Aldrich). Before library generation, the integrity of extracted RNA was validated using RNA ScreenTape on TapeStation 4150 (Agilent). Sequencing libraries were generated using the QuantSeq FWD mRNA Seq V2 Library Prep Kit with UDI (Lexogen) following the manufacturer's instructions. Final libraries were inspected using High Sensitivity D1000 ScreenTape on TapeStation 4150 (Agilent). The libraries were mixed in equimolar ratio (15 pM each) and sequenced on the Illumina NextSeq 2000 using NextSeq 1000/2000 P2 Reagents (100 Cycles) v3 (Illumina) using the following parameters: read1—75, index1— 12, index2—12. The original BCL files were demultiplexed and converted to FASTQ using Illumina Dragen v4.2.7. FASTQ files were expected using FastQC v0.12.0 and processed before alignment, first by using Trimmomatic v0.40 with the following parameters: ILLUMINACLIP: TruSeq3-SE.fa:2:30:10, HEAD-CROP:12, followed by Cutadapt v5.1 with the following parameters: -a "polyA=A{20}" -a "QUALITY = G{20}" -m 20 -O 20. GENCODE annotation was used to map reads to the Maus genome GRCm39.114 (GCA_000001635.9) with STAR v2.7.11b and the following parameters: --outSAMtype BAM SortedByCoordinate --outFilterMultimapNmax 10 --alignEndsType EndToEnd. To count the reads into exons, HTSeq-count v1.99.2 was used, with the default parameters. The count tables were used as input for differential gene expression analysis with DESeq2 v1.48.1. Genes with an absolute log2 fold change|log2FC| >1 were considered significant at adj. $p < 0.05$.

## NADH/NAD$^+$ assay

Cellular NAD$^+$ and NADH levels were quantified using the NAD/NADH-Glo™ Assay (Promega) following the manufacturer's protocol. Briefly, to each well of a 96-well plate, 50 µl PBS was added, and cell lysis was performed by adding 50 µl base solution containing 1% DTAB. Each sample was split into two fractions: one treated with 0.4 N HCl to measure NAD$^+$ and the other left as base-treated to measure NADH. Samples were incubated at 60 °C for 15 min, equilibrated at room temperature, and neutralized with Trizma® base. An equal volume of NAD/NADH-Glo™ Detection Reagent was then added to each well, mixed, and incubated for 30–60 min at room temperature. Luminescence was recorded using a plate luminometer. Luminescence from acid-treated wells reflected NAD$^+$ levels, whereas base-treated wells reflected NADH levels.

## NADPH/NADP$^-$ assay

Cellular NADP$^+$ and NADPH were quantified using the NADP/NADPH-Glo™ Assay (Promega) according to the manufacturer's instructions. To each well of a white 96-well luminometer plate, 50 µl PBS was added, and cell lysis was performed by adding 50 µl base solution containing 1% DTAB. Each sample was split into two fractions: one was transferred (50 µl) to an empty well and treated with 25 µl 0.4 N HCl (acid-treated, for NADP$^+$), while the original wells remained base-treated (for NADPH). Plates were covered and incubated at 60 °C for 15 min, then equilibrated at room temperature for 10 min. Acid-treated wells were neutralized with 25 µl 0.5 M Trizma® base; base-treated wells received 50 µl HCl/Trizma® solution. An equal volume of NADP/NADPH-Glo™ Detection Reagent (e.g., 100 µl) was added to each well, plates were gently mixed, and incubated for 30–60 min at room temperature. Luminescence was recorded on a plate luminometer. Signal from acid-treated wells reflects NADP$^+$, whereas signal from base-treated wells reflects NADPH.

## ROS measurement

Reactive oxygen species were quantified using the ROS-Glo™ H$_2$O$_2$ Assay (Promega) according to the manufacturer's instructions. Cells were seeded in white opaque 384-well plates and treated with KI696 or DMSO for 42 h, with a final volume of 40 µl. Following treatment, 10 µl of H$_2$O$_2$ Substrate Solution was added to each well and incubated for 6 h at 37 °C. Subsequently, 50 µl of ROS-Glo™ Detection Solution was added, the plates were incubated for 20 min at room temperature, and luminescence was measured using a plate luminometer.

## Immunoblotting of total cell lysates

Cells were collected from the culture medium and washed with PBS before lysis in buffer containing 50 mM Tris-HCl (pH 7.5), 150 mM NaCl, 10 mM NaF, 1 mM EGTA, 5 mM MgCl$_2$, 1% Triton X-100, and protease and phosphatase inhibitors. Insoluble material was removed by centrifugation at 20,000×g for 10 min at 4 °C, and protein concentration was determined using the Bradford reagent (Sigma-Aldrich). Proteins were denatured in SDS sample buffer at 95 °C for 10 min, separated by SDS-PAGE, and transferred to PVDF membranes (Millipore). Membranes were blocked in PBS containing 0.1% Tween 20 (PBST) with 5% BSA or 5% nonfat milk for 30 min, followed by overnight incubation at 4 °C with primary antibodies. After washing with PBST, membranes were incubated with secondary antibodies in blocking solution, washed three times with PBST, and signals were detected using the iBright 1500 (Thermo Fisher) or ChemiDoc MP Imaging System (Bio-Rad).

## Extracellular flux analysis

We used Seahorse XF96 extracellular flux analyzer (Agilent) to analyze oxygen consumption rates (OCR) of the cell lines. Cells were plated in Seahorse 96-well cell culture plates at $3 \times 10^4$ cells/well 1 day before the assay and equilibrated for 1 h in Seahorse DMEM medium (Agilent) supplemented with 25 mM glucose and 2 mM glutamine. After recording basal OCR, sequential injections of 2.5 µM oligomycin (Sigma-Aldrich), 1 µM carbonyl cyanide 3-chlorophenylhydrazone (CCCP, Sigma-Aldrich), 1 µg/ml anti-mycin (Sigma-Aldrich), and 1 µM rotenone (Sigma-Aldrich) were performed as indicated. Maximal respiration refers to OCR after CCCP injection. Shown is a representative OCR trace from three biological replicates.

## Cell viability assays

In order to assess cell viability, human lung cancer cells ($n = 2500$) were seeded into 96-well plates (Thermo Scientific Nunclon™ Delta Surface) and cultivated for 24 h at 37 °C, 5% $CO_2$ and 95% humidity. Then, cells were treated with Bardoxolone (0–1 μM), Omaveloxolone (0–10 μM), or DMSO as a control for 24 h at 37 °C, 5% $CO_2$ and 95% humidity. Afterwards, cells were stained with PI (Propidium iodide 25 μg/ml) and Hoechst (1 μg/ml) for 15 min at room temperature. Plates were washed with PBS, and cells were fixed with PFA (4% in PBS) for 7 min at room temperature. Subsequently, the staining was imaged using a high content screening platform and analysed with HCS Navigator ™ (Thermo Fisher; version 6.6.4 (build 8616)). Murine lung tumor cells were seeded in 96-well plates following sgRNA and/or siRNA treatment. After 24 h, cell viability was assessed using the CellTiter-Glo Luminescent Cell Viability Assay (Promega, G7570) according to the manufacturer's instructions. Viability values were normalized to the control sgRNA condition.

## Immunofluorescence (High content)

Cells ($n = 2500$) were seeded into 96-well plates (Thermo Scientific Nunclon™ Delta Surface) and cultivated for 24 h at 37 °C, 5% $CO_2$ and 95% humidity. Afterwards, cells were treated with Bardoxolone (0–0.33 μM), Omaveloxolone (0–3.33 μM) or DMSO as control for 24 h at 37 °C, 5% $CO_2$ and 95% humidity. Afterwards, cells were washed with PBS and cells fixed with PFA (4% in PBS) for 7 min at room temperature. The samples were permeabilized with PBS 0.4% Triton X-100 for 4 min, washed with PBS and blocked for 1 h at room temperature with 5% BSA in PBS. To stain, antibodies were primary labeled with the FlexAble 2.0 CoraLite® Plus Antibody Labeling Kit for Rabbit IgG according to manufactures instructions. The samples were stained overnight at 4 °C with the antibodies, respectively. Cells were washed with PBS and subsequently, the staining was imaged using a high content screening platform and analysed with HCS Navigator ™ (Thermo Fisher; version 6.6.4 (build 8616)). Data were analysed and visualized in RStudio (version 2023.09.1 build 494 based on the R-version 4.3.1)

## Genetic interaction models

Genetic interactions were scored according to an additive model, scoring the difference between observed and expected phenotypes of double mutants induced by CRISPR knockout. The phenotype of individual and double mutants was measured as the log2 fold change of gRNA abundance between samples. The expected phenotype was calculated as the sum of the individual phenotypes. The difference between observation and expectations was calculated as the delta of the two values, dLFC = observed − expected.

### Statistics

Genes with an LFC >1 or <−1 and a $p$ value <0.05, according to the MAGeCK analysis, were considered to be significantly depleted or enriched for CRISPR screens. Significance was calculated either using paired Student's $t$-test, one-way ANOVA and Mann–Whitney test.

### Analysis of human publicly available datasets

Kaplan–Meier curves for NSCLC overall and free progression human survival rates were estimated with the online tool KM plotter (https://kmplot.com) based on GEO, EGA, and TCGA datasets. Overall survival $n = 2167$ samples and free progression survival $n = 1252$ samples. By using the KM plotter online tool, patients were split into "Auto select best cutoff" in high or low gene expression groups. In "Auto select best cutoff", to avoid missing correlations due to the use of a specific cutoff, all available cutoff values between the lower and upper quartiles of expression are used for the selected gene, and false discovery rate (FDR) using the Benjamini–Hochberg method is computed to correct for multiple hypothesis testing. The cutoff value with the highest significance (lowest FDR) is determined. In case of multiple cutoff values with identical significance, the cutoff with the highest hazard (HR) rate is selected for the final analysis. Hazard ratios (HR) with 95% confidence intervals and log-rank $p$ values of the Kaplan–Meier curves were calculated with the KM plotter. The next probe ID (Gene symbol) were used for the survival curves: 202417_at (KEAP1), 201746_at (TP53), 222461_s_at (HERC2), 207783_x_at (HUWE1), 201146_at (NFE2L2), 215636_at (UBR4), 225363_at (PTEN), and 41657_at (STK11).

Human lung cancer genomic alterations were obtained in cBioPortal (Gao et al, 2013). The following studies were used for the different analyses: lung adenocarcinoma (LUAD-TCGA Firehose Legacy; $n = 586$) and lung squamous cell carcinoma (LUSC-TCGA Firehose Legacy; $n = 511$).

Graphic comparing gene expression between normal and tumor human samples were generated using the online tool GEPIA2 (Tang et al, 2017). As indicated on the GEPIA2 website, the differential analysis was based on TCGA tumors versus TCGA normal and/or GTEX, whereas the expression data were log2 $(TPM + 1)$ − transformed, and the log2FC was defined as median (tumor) − median (normal). LUAD GEPIA2 dataset: $n$Tumor = 483 and $n$Normal = 59 and LUSC GEPIA2 dataset: $n$Tumor = 486 and $n$Normal = 50. Boxplots were generated using the GEPIA2 website. The $p$ values were calculated with a one-way ANOVA comparing tumors with normal samples.

Boxplots comparing relative protein abundance (TMT log2-ratio) in normal and tumor samples were generated using the online tool Cancer Proteogenomic Data Analysis Site based on the National Cancer Institute's Clinical Proteomic Tumor Analysis Consortium (CPTAC) and National Cancer Institute's International Cancer Proteogenome Consortium (ICPC) datasets (Wang et al, 2023). The statistical $p$-values were automatically calculated by the online tool as indicated on the website of the online tool (https://cprosite.ccr.cancer.gov/#/).

Protein correlation of lung cancer cell lines were performed using the online tool data explorer from the Dependency Map (Depmap) portal, as indicated on the website (https://depmap.org/portal/).

Gene expression analysis using KEAP1/NRF2 wt and mutant LUAD and LUSC human samples were performed with the data obtained from Arolt et al (Arolt et al, 2023). Gene ontology analysis were performed with the online tool ShinyGO 0.80 (http://bioinformatics.sdstate.edu/go/) and volcano plots using the online tool VolcaNoseR (https://huygens.science.uva.nl/VolcaNoseR/).

## sgRNA design

sgRNA for 3Cs libraries were selected using Azimuth 2.0 (GPP sgRNA Designer, PMID: 26780180). sgRNAs for mouse in vivo experiments were designed using the CRISPRtool https://chopchop.cbu.uib.no/).

## AAV production

To produce AAVs, $5x\,10^6$ HEK293T-cells were seeded in 15 cm cell culture dishes and cultivated for 24 h or until a confluence of ~60–70% was achieved. Cells were transfected with the pRepCap (pDJ), the cis-plasmid (pAAV), and the pAdDeltaF6 in a 1:1:2 molar ratio. To harvest the AAV, cells and supernatant were collected after 96 h and transferred into a 50 ml conical tube. At first, NaCl was added (f.c. 0.5 M) and slowly mixed for 1 h at 4 °C. Next, chloroform was added (f.c. 10%) and slowly mixed for 30 min at 4 °C. Eventually, the suspension was centrifuged at $2000\times g$ for 30 min at 4 °C. The water phase was transferred into a new conical tube, and PEG8000 was added (f.c. 10%) and mixed well. The AAV was precipitated overnight at 4 °C. After the centrifugation at $2000\times g$ for 20 min at 4 °C the pellet was resuspended in AAV-resuspension buffer (PBS + 0.001% pluronic F68 + 200 mM NaCl) (~100 µl/15 cm dish used) and protease inhibitor and DNase/RNase were added (f.c. 1x). After incubation for 2 h at 37 °C, chloroform was added (1:1 ratio) and mixture was centrifuged at $12,000\times g$ for 5 min at 4 °C. The chloroform step was repeated, and the water phase was collected, followed by titration and packaging control or stored at −80 °C.

## Licenses

All in vivo experiments were approved by the Regierung Unterfranken and the ethics committee under the license numbers 2532-2-362, 2532-2-367, 2532-2-374, and 2532-2-1003. The mouse strains used for this publication are listed. B6J.129(Cg)-*Gt(ROSA) 26Sor*^tm1.1(CAG-cas9*,-EGFP)Fezh^/J

## Animal welfare

All animals are supervised daily, and animal health is monitored with a sentinel mouse. Furthermore, a disease screening was conducted every three months. Animals are housed in standard cages in pathogen-free facilities on a 12-h light/dark cycle with ad libitum access to food and water. The FELASA2014 guidelines were followed for animal maintenance.

## AAV-induced lung tumors

Lung tumors were induced as described previously (PMID: 32128997). Adult mice were anesthetized with Isoflurane in a chamber with a constant flow of 3% Isoflurane and intratracheally intubated with 60 µl AAV virus ($1 \times 10^{11}$ PFU/ml). For endotracheal instillation, a gauge 24 catheter was used, and the AAVs were pipetted to the top of the catheter. During normal animal breathing, the virus was distally expanded and delivered into the lungs. Viruses were quantified using the AAV titration by qPCR protocol from Addgene. As a control, some animals were intratracheally intubated with 60 µl AAV-resuspension buffer without AAVs. Animals were sacrificed after 12 weeks by cervical dislocation, and lungs were fixed using 5% NBF. If animals showed severe symptoms or lost more than 10% of their weight, they were taken out at that point.

## Immunohistochemistry

Paraffin-embedded sections of murine samples were cut into 2–4 µm sections with a microtome (Leica). Before staining, slides were de-paraffinized and rehydrated using the following protocol: 3 × 5 min in xylene, 2 × 3 min in EtOH (100%), 2 × 3 min in EtOH (95%), 2 × 3 min in EtOH (70%), 2 min in EtOH (50%), and 3 min in $H_2O$. After de-paraffinization and rehydration, antigen retrieval was performed with 10 mM sodium citrate buffer (pH 6.0) in a microwave oven at 800, 650, and 360 W for 5 min, respectively. The samples were permeabilized with TBS 0.1% Tween 20 for 10 min and washed with TBS and blocked for 1 h at room temperature with 10% goat-serum, 1.5% BSA in TBS. The respective primary antibodies were diluted in blocking solution and incubated overnight at 4 °C. Next, the endogenous Peroxidase was blocked with TBS containing 3% $H_2O_2$ for 10 min. Slides were developed with the SuperBoost™ HRP coupled secondary antibodies and with SignalStain® DAB Substrate Kit (Cell Signaling Technology) and counterstained with Hematoxylin (Sigma H3136). Slides were scanned in 40x resolution using a Olympus VS100 slide scanner and analysed using QuPath (version 0.5.0).

## Patient-derived tissue microarrays (TMA)

Paraffin molds were cast using an Arraymold Kit (IHC World, Kit D, IW-115, core diameter 2 mm, 36 cores). Human samples were cut and stained using hematoxylin and eosin and digitized using an Olympus VS120 slide scanner. Tumor and non-transformed tissue were identified and manually "punched" and transferred from the tissue block to the tissue array. Upon completion, 3-µm-thick sections were cut using a microtome and processed as described below.

## Histopathology and human NSCLC TMA

For IHC and H&E, slides were de-paraffinized and rehydrated as follows: IHC slides were subjected to epitope retrieval and blocked in 3% BSA at RT for 1 h. The antibody manufacturer's instructions were followed for all antibodies. In general, primary antibodies (diluted in 1% BSA) were incubated overnight at 4 °C, followed by three washes with PBS and subsequent incubation with the DAB secondary antibody for 1 h at RT. Then, slides were washed twice with 1xPBS for 5 min and stained with the DAB staining solution in 1xPBS. Upon DAB staining, slides were counterstained with hematoxylin. Slides were mounted using 200 µl of Mowiol® 40–88 covered by a glass coverslip. IHC slides were recorded using an Olympus VS120 and analyzed using QuPath software, PRISM and ImageJ. Antibodies used are listed in the consumables section.

# Data availability

Proteomics data generated in this study have been deposited in the PRIDE repository with the dataset identifier PXD067472. The data

were publicly available at https://www.ebi.ac.uk/pride/archive/projects/PXD067472. RNA-seq data have been deposited in the National Center for Biotechnology Information (NCBI) Gene Expression Omnibus (GEO) repository, with the dataset identifier GSE306334 [https://www.ncbi.nlm.nih.gov/geo/query/acc.cgi?acc=GSE306334].

The source data of this paper are collected in the following database record: biostudies:S-SCDT-10_1038-S44318-026-00737-9.

## Peer review information

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

## Acknowledgements

We thank all members of the ID laboratory for their support and constructive discussions. We also acknowledge Antonia Hofmann and Kristin Jordan for technical assistance, and Giulio Giliani for critical input on the manuscript. The research leading to these results was funded by the Deutsche Forschungsgemeinschaft (DFG, German Research Foundation)—TRR 387/1—514894665, DFG GRK2243, DIP DI931/18-1, and DKH grant 70114554. We thank all members of the Quantitative Proteomics Unit at IBC2 (Goethe University, Frankfurt), in particular, Martin Adrian-Allgood for sample preparation and measurements, Kristina Wagner for producing LC columns and David Krause for help in (bio)informatics. We thank the Deutsche Forschungsgemeinschaft (German Research Foundation, DFG) for funding the LC-MS system (easy nLC1200, Orbitrap Fusion LUMOS) used in this study (FuGG Project-ID: 403765277). Figures 1A–D, 4G, 6G and EV1E were created in Biorender.

## Author contributions

**Varun Jayeshkumar Shah**: Conceptualization; Data curation; Formal analysis; Supervision; Validation; Investigation; Visualization; Writing—original draft; Project administration; Writing—review and editing. **Oliver Hartmann**: Data curation; Formal analysis; Validation; Investigation; Visualization; Methodology; Writing—review and editing. **Martin Wegner**: Data curation; Formal analysis; Visualization. **Cristian Prieto-Garcia**: Data curation; Formal analysis; Visualization; Methodology; Writing—review and editing. **Rubina Kazi**: Data curation; Formal analysis; Investigation; Visualization. **Viktoria Von Heyl zu Herrnsheim**: Data curation; Formal analysis; Validation; Investigation; Visualization. **Amin Wanli**: Formal analysis; Investigation. **Igor Mačinković**: Data curation; Formal analysis; Investigation; Visualization. **Bianka Bohnacker**: Formal analysis; Investigation. **Koraljka Husnjak**: Writing—review and editing. **Dmitry Namgaladze**: Data curation; Formal analysis; Investigation; Visualization. **Mathias Rosenfeldt**: Resources. **Manuel Kaulich**: Data curation; Supervision; Visualization. **Markus E Diefenbacher**: Conceptualization; Resources; Formal analysis; Supervision; Funding acquisition; Visualization; Methodology; Project administration; Writing—review and editing. **Ivan Dikic**:

Conceptualization; Resources; Supervision; Funding acquisition; Writing—original draft; Project administration; Writing—review and editing.

Source data underlying figure panels in this paper may have individual authorship assigned. Where available, figure panel/source data authorship is listed in the following database record: biostudies:S-SCDT-10_1038-S44318-026-00737-9.

## Funding

## Disclosure and competing interests statement

Ivan Dikic is a member of the Advisory Editorial Board of The EMBO Journal. This has no bearing on the editorial consideration of this article for publication. The remaining authors declare no competing interests.

# Expanded View Figures

**Figure EV1.   Generation of UPS-centric single CRISPR/Cas9 library.**

(A) Lorenz curve displaying the cumulative fraction of represented NGS reads versus the gRNAs ranked by abundance of each library revealed a uniform distribution of gRNA sequences. Area under the curve values (AUC) confirm the uniform gRNA distribution of these libraries. (B) The E3-DUB gRNA single screens are highly reproducible, as visualized by scatter plots comparing biological replicates (Exp#1/Exp#2/Exp#3) of normalized gRNA read counts at the 14-day time point in single CRISPR/Cas9 screens. (C, D) Sequencing depth of library and 14-day time point for gRNA (C) and gene level (D) plotted. Sample minimas are defined as the lowest single black dot, medians as green lines, bounds of boxes and whiskers as 25% of the sample population. Sample size (n) is three and indicated as the number associated with cell line IDs (e.g., KP1, KP2, and KP3). (E) Schematics of single and multiplex CRISPR/Cas9 screens. For a single library- coverage 1000x, MOI of 0.5 and for a multiplex library- coverage 60x and MOI of 0.5.

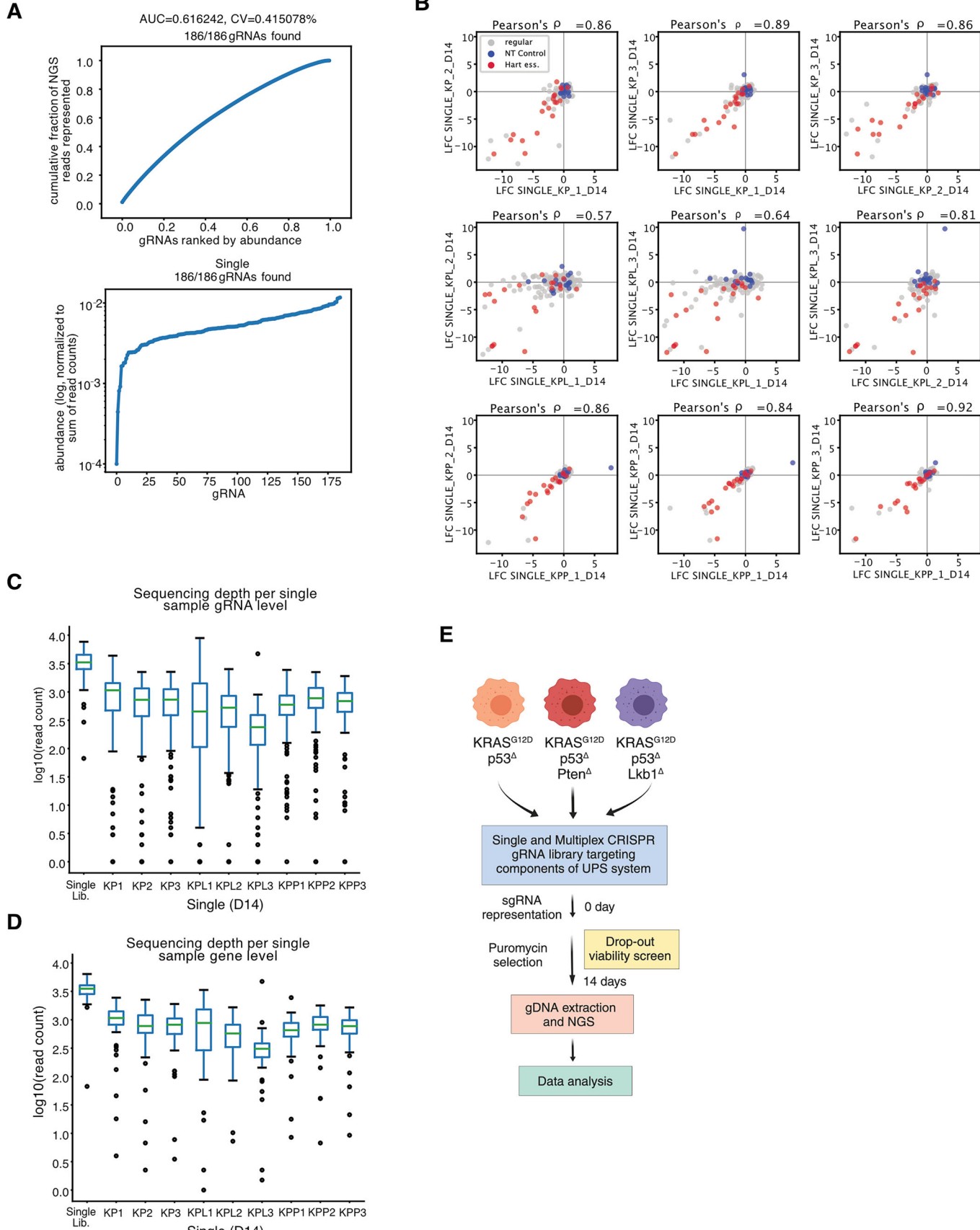

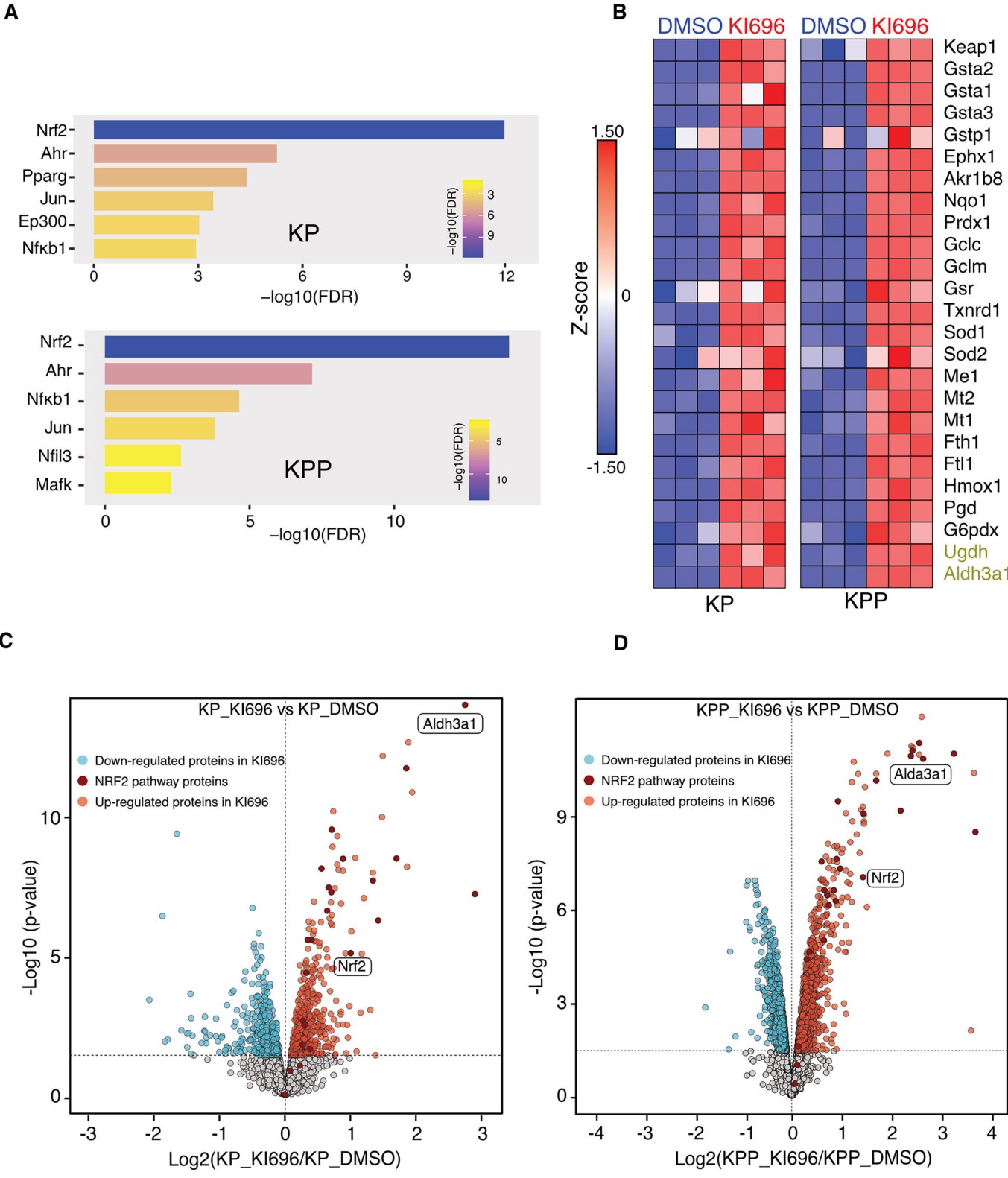

◀ **Figure EV2. Keap1 inhibition activates NRF2-driven pathways.**

(A) KP (top) and KPP (bottom) cells were treated with KI696 (1 μM) for 24 h, and changes in gene expression were assessed by RNA-seq. Results are presented as enriched transcription factor signatures. (B) Heat map showing gene expression (z-score) of NRF2 target genes in *DMSO* and KI696 ($n = 3$) treated KP (left) and KPP (right) cells. (C, D) Whole-cell proteome analysis by mass spectrometry. Volcano plot of proteomic changes in KP cells (C) or KPP cells (D) treated with KI696 (1 μM) versus DMSO. Significantly decreased (blue) and increased (red) proteins ($p < 0.05$) are shown; dark red highlights significantly increased NRF2 pathway proteins ($n = 3$). Statistical significance was assessed by two-sided moderated t-test as implemented in the limma package via FragPipe Analyst; $p$ values were adjusted using the Benjamini–Hochberg method.

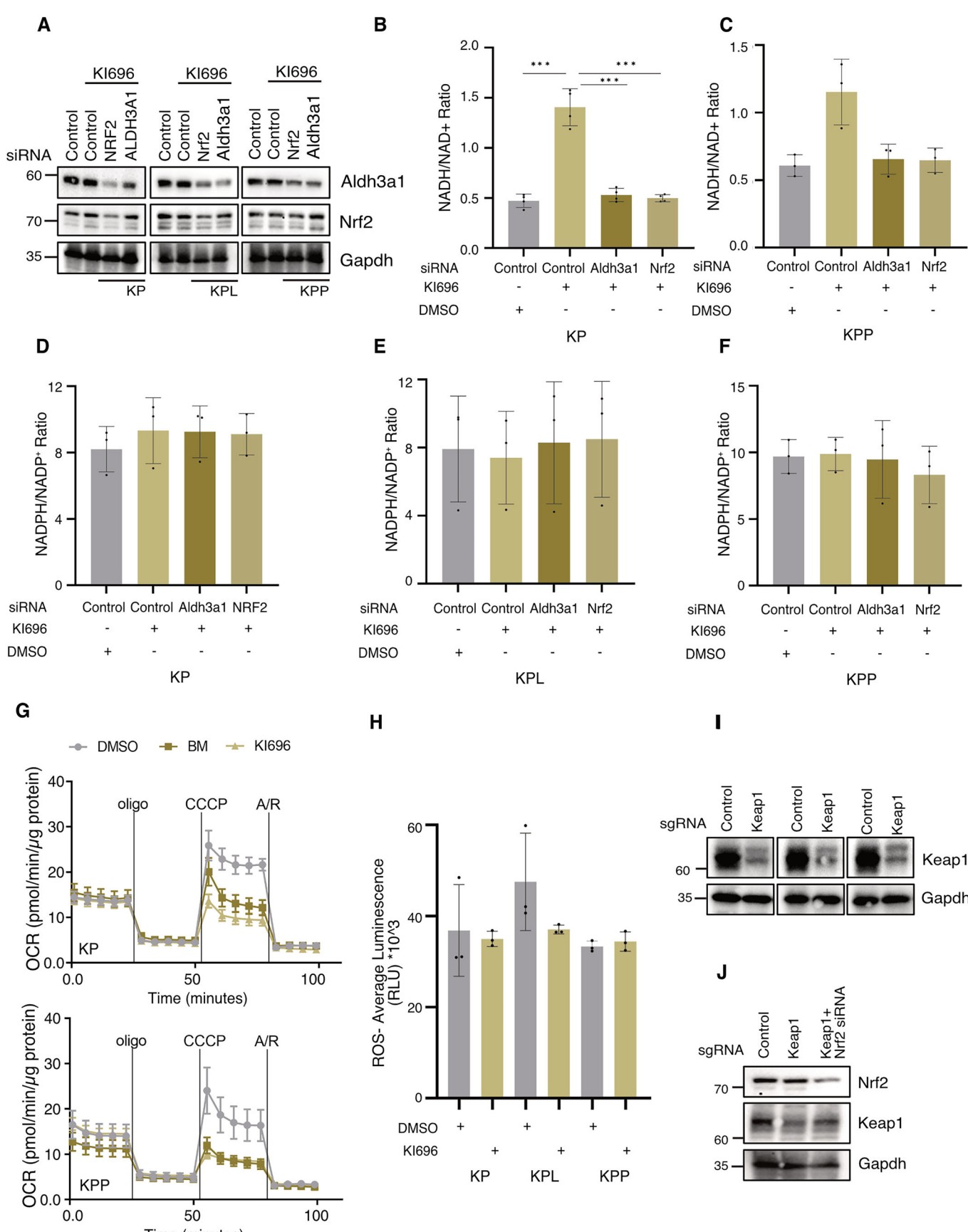

◀ **Figure EV3. Keap1 regulates redox stress and mitochondrial functions.**

(A) Immunoblots for Aldh3a1, NRF2, and GAPDH in KP, KPL, and KPP cells transfected with si-Control, si-NRF2, or si-Aldh3a1, in the presence or absence of KI696. (B) NADH/NAD$^+$ ratio in KP cells treated with control, Aldh3a1, or NRF2 siRNA, in the presence of DMSO or KI696. Data represent mean ± SD of three biological replicates. ***$P < 0.001$. Control vs control + KI696 $P = 0.0001$, Control + KI696 vs Aldh3a1 si + KI696 $P = 0.0001$, Control + KI696 vs NRF2 si + KI696 $P = 0.0001$. Statistical significance was calculated using one-way ANOVA. (C) NADH/NAD$^+$ ratio in KPP cells treated with control, Aldh3a1, or NRF2 siRNA, in the presence of DMSO or KI696. Data represent mean ± SD of three biological replicates. ns not significant ($P > 0.05$). Statistical significance was calculated using one-way ANOVA. (D–F) NADPH/NADP$^+$ ratio in KP (D), KPL (E), or KPP (F) cells treated with control, Aldh3a1, or NRF2 siRNA, in the presence of DMSO or KI696. Data represent mean ± SD of three biological replicates. ns not significant ($P > 0.05$); statistical significance determined by one-way ANOVA. (G) Oxygen consumption rate (OCR) of KP (top) and KPP (below) cells treated with DMSO, KI696, or Bardoxolone methyl (BM). Data represent mean ± SEM of five technical replicates. (H) ROS levels of KP, KPL and KPP cells treated with DMSO or KI696 for 48 h. ns not significant ($P > 0.05$). Statistical significance was calculated using a paired *t*-test. (I) Immunoblots for Keap1 and GAPDH in KP, KPL, and KPP cells transduced with Control or Keap1 sgRNA. (J) Immunoblots for NRF2, Keap1, and GAPDH in KPP cells transduced with Control sgRNA, Keap1 sgRNA, or Keap1 sgRNA + NRF2 siRNA.

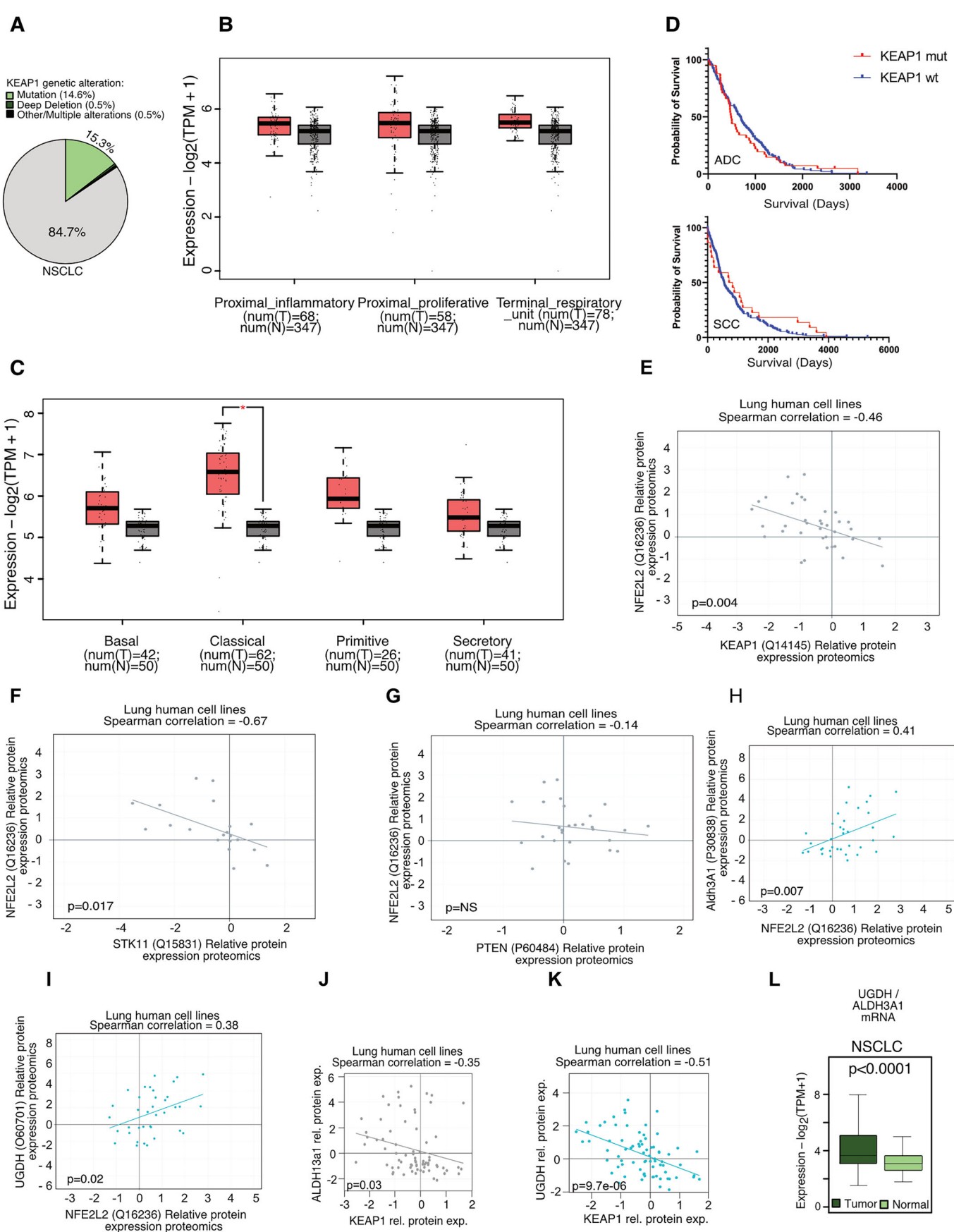

◀ **Figure EV4. Comparison of relative expression data.**

(A) Occurrence of mutations within Keap1 in NSCLC adenocarcinoma and squamous cell carcinoma. Data from cbiportal.org. (B, C) Expression of KEAP1 in non-transformed (N) and (B) LUAD (T) or (C) LUSC (T) samples, represented according to consensus LUAD and LUSC subtypes. Publicly available data were extracted from TCGA via the web tool GEPIA2 (http://gepia2.cancer-pku.cn/). Normal samples comprises TCGA normal and GETEx samples. Num indicates samples analysed. Significance was calculated with Dixon's Q-test. *P* value *$p < 0.05$. The box plot represents the lower (25%) and upper quartile (75%) with the median (50%) displayed. The whisker represents the minimum and maximum. (D) Patient survival data based on Keap1 status (wildtype or mutant) in ADC (top) (NS) and SCC (bottom) NS not significant ($P > 0.05$). (Source: Tcga, Xena variant. https://xena.ucsc.edu/). (E) Correlation of protein expression of NFE2L2 and Keap1 in lung cancer cell lines. (F) Correlation of protein expression of STK11 and Keap1 in NSCLC cancer cell lines. (G) Correlation of protein expression of NFE2L2 and PTEN in lung tumors. (H, I) Correlation of protein expression of NFE2L2 and Aldh3a1 (H) and UGDH (I) in lung tumor cancer cell lines. (J, K) Keap1 protein expression relative to Aldh3A1 (J) and UGDH (K), respectively. (L) UGDH and ALDH3A1 gene expression levels in lung cancer samples relative to adjacent non-transformed tissue. *P* value *$p < 0.0001$. The significance was calculated with a one-way ANOVA comparing tumors with normal samples. Data from gepia2.cancer-pku.cn.

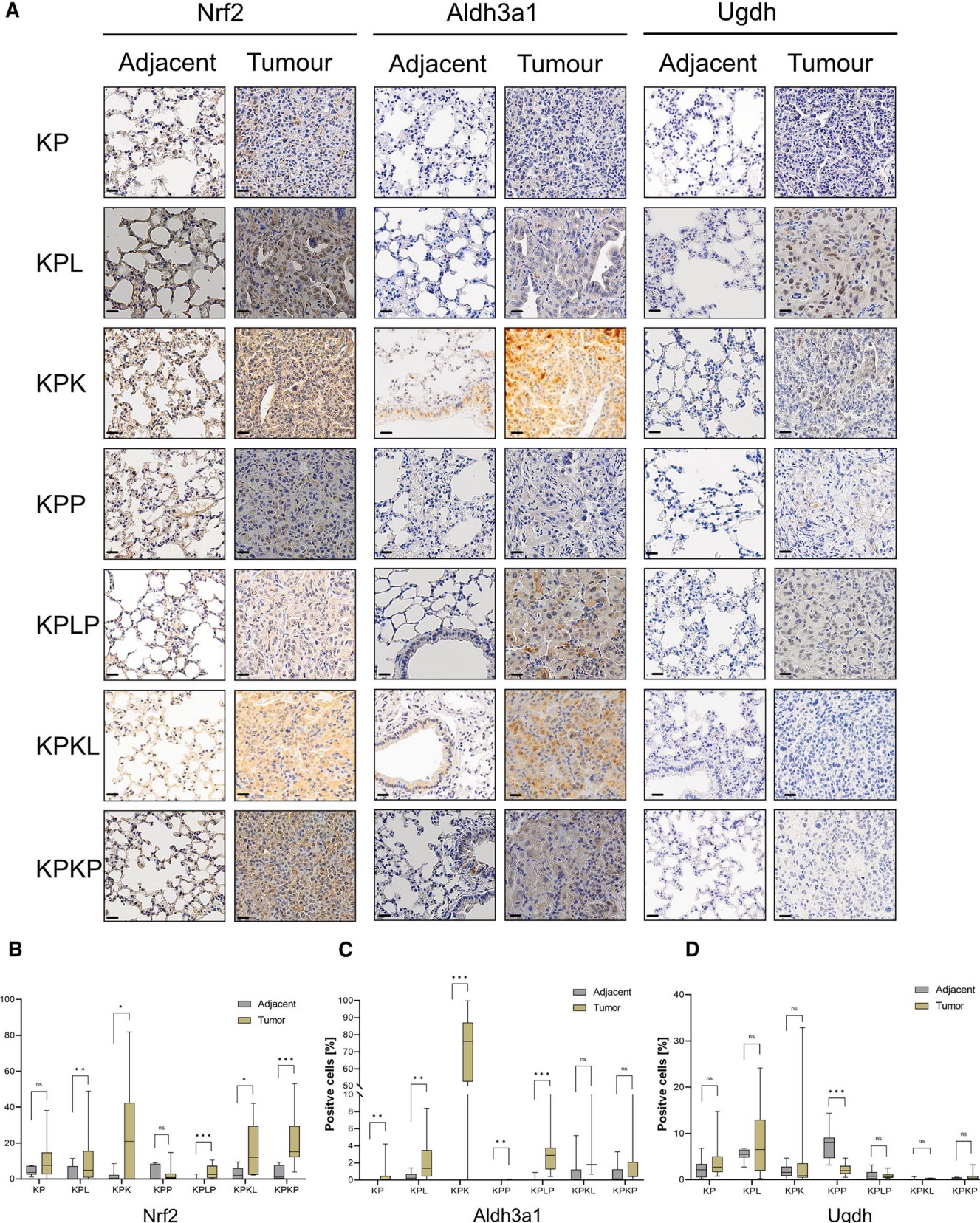

**Figure EV5.  Immunohistochemistry (IHC) analysis of reductive stress markers in tumors and adjacent lung tissue.**

(**A**) Representative IHC sections from mice of different genetic backgrounds, 12 weeks post-intubation. Staining for NRF2, ALDH3A1, and UGDH in adjacent and tumor tissue was performed to assess redox stress. Scale bar, 20 μm. (**B–D**) Quantification of positive cells was performed using QuPath (v0.6.0) from IHC staining shown in (**A**) for Nrf2 (**B**), Aldh3a1 (**C**), and Ugdh (**D**). Significance was calculated using the Mann−Whitney test. (Significance $p < 0.05$) ($n$[adjacent] >8; $n$[tumor] >8; except $n$[KPKL] = 3). (**B**) $p$(KP) = 0.1216; $p$(KPL) = 0.0019; $p$(KPK) = 0.0430; $p$(KPP) = 0.2002; $p$(KPLP) = 0.0002; $p$(KPKL) = 0.0348; $p$(KPKP) < 0.0001. (**C**) $p$(KP) = 0.0013; $p$(KPL) = 0.0081; $p$(KPK) < 0.0001.; $p$(KPP) = 0.0044; $p$(KPLP) < 0.0001; $p$(KPKL) = 0.0824; $p$(KPKP) = 0.1670. (**D**) $p$(KP) = 0.2257; $p$(KPL) = 0.7694; $p$(KPK) = 0.3115; $p$(KPP) < 0.0001.; $p$(KPLP) = 0.8071; $p$(KPKL) = 0.0070; $p$(KPKP) = 0.4555. The box plot represents the lower (25%) and upper quartile (75%) with the median (50%) displayed. The whisker represents the minimum and maximum.

