## [Peer Review File · The EMBO Journal]

Targeting Ubiquitin Signaling Vulnerabilities in KEAP1-Inactivated Lung Cancer

Varun Shah, Oliver Hartmann, Martin Wegner, Cristian Prieto-Garcia, Rubina Kazi, Viktoria Von Heyl zu Herrnsheim, Amin Wanli, Igor Mačinković, Bianka Bohnacker, Koraljka Husnjak, Dmitry Namgaladze, Mathias Rosenfeldt, Manuel Kaulich, Markus Diefenbacher, and Ivan Dikic

Corresponding author(s): Ivan Dikic (dikic@biochem2.uni-frankfurt.de) , Markus Diefenbacher (markus.diefenbacher@helmholtz-munich.de)

Review Timeline:

Submission Date:	22nd Nov 24
Editorial Decision:	15th Jan 25
Revision Received:	18th Sep 25
Editorial Decision:	28th Nov 25
Revision Received:	3rd Feb 26
Accepted:	11th Feb 26

Editor: Hartmut Vodermaier

Transaction Report:

Prof. Ivan Dikic
Institute of Biochemistry II, Faculty of Medicine, Goethe University, Frankfurt, Germany
Institute of Biochemistry II
Theodor-Stern-Kai 7
Frankfurt 60590
Germany

15th Jan 2025

Re: EMBOJ-2024-119715
Targeting Ubiquitin Signaling vulnerabilities in KEAP1-Inactivated Lung Cancer

Dear Ivan,

Thank you for submitting your manuscript on KEAP1 roles in lung cancer for our consideration. I am sorry for the delay in getting back to you with a response, which was mainly due to limited reviewer availability at the end of the year, and generous deadline extensions needed them. We have now finally received the below-copied reports from four referees with expertise in KEAP1/NRF2 pathways, lung cancer, and ubiquitination/cancer links, respectively. As you will see, all referees acknowledge the potential interest and novelty of your findings, but also note that the underlying mechanisms have not been established as conclusively as proposed, and that certain aspects of the work would require strengthened or complementary evidence. Having considered the various comments in detail, I feel that the study would warrant further consideration for EMBO Journal publication, pending adequate revisions of key points raised by the reviewers. Especially reviewers 3 and 4 raise issues regarding the cancer models/human cancer modalities and oncogenic drivers/suppressors, as well as confirmation of their respective manipulation in tumor cells/tissues. Furthermore, there are concerns regarding the partly descriptive nature of the study and stronger evidence for causative connections, with referees 1, 3 and 4 noting several aspects on which deeper insights could be obtained. While I would not expect you to follow up on each of these suggestions, I would strongly encourage you to contact me with a tentative point-by-point response and revision plan already during the early stages of your revision work, so that we could discuss how key issues raised in the reports might best be resolved; this would be helpful in light of our policy to consider only a single round of major revision. If needed, we would also be open to extension of the regular three-months revision period, during which our 'scooping protection' would of course remain valid.

Further information on preparing, formatting and uploading a revised manuscript can be found below and in our Guide to Authors. Thank you again for the opportunity to consider this work for The EMBO Journal, and I look forward to hearing from you in due time.

With kind regards,

Hartmut

9) To facilitate reproducibility and cross-laboratory adoption of methodologies, please structure the Materials & Methods section as outlined in our guide to authors, including a completed Reagents and Tools Table that can be downloaded from our author guidelines as well (<https://www.embopress.org/page/journal/14602075/authorguide#structuredmethods>).

10) Digital image enhancement is acceptable practice, as long as it accurately represents the original data and conforms to community standards. If a figure has been subjected to significant electronic manipulation, this must be clearly noted in the figure legend and/or the 'Materials and Methods' section. The editors reserve the right to request original versions of figures and the original images that were used to assemble the figure. Finally, we generally encourage uploading of numerical as well as gel/blot image source data; for details see: embopress.org/page/journal/14602075/authorguide#sourcedata

At EMBO Press, we ask authors to provide source data for the main manuscript figures. Our source data coordinator will contact you to discuss which figure panels we would need source data for and will also provide you with helpful tips on how to upload and organize the files.

In the interest of ensuring the conceptual advance provided by the work, we recommend submitting a revision within 3 months (15th Apr 2025). Please discuss the revision progress ahead of this time with the editor if you require more time to complete the revisions. Use the link below to submit your revision:

Link Not Available

Referee #1:

This study is based on previous knowledge that genetic alterations in components of the UPS, including E3 ubiquitin ligases (E3s) and deubiquitinating enzymes (DUBs), which facilitate the maintenance of proteostasis, are common in cancer, facilitating tumor progression, but also creating potential therapeutic vulnerabilities. Using mouse lung cancer models, specifically KRAS-driven non-small cell lung cancers (NSCLC) with co-occurring deletions in TP53 (KrasG12D105 Tp53^{-/-}; KP), TP53 and LKB1 (KrasG12D Tp53^{-/-}Lkb1^{-/-}; KPL), or TP53 and PTEN (KrasG12D Tp53^{-/-} Pten^{-/-}; KPP), the authors have performed combinatorial CRISPR/Cas9 dropout viability screens, targeting members of the UPS. Surprisingly, they identified Keap1 as a therapeutic vulnerability, and several druggable E3s and DUBs as Keap1-dependent co-vulnerabilities, such as the E3s Herc2, Ubr4 30 and Huwe1, the depletion of which suppressed the development of Keap1-inactivated tumors. This was surprising, because loss-of-function mutations in Keap1, leading to constitutive activation of transcription factor Nrf2, occur frequently in aggressive NSCLC, conferring resistance to therapies, including chemotherapy, radiation therapy and immunotherapy.

To investigate the underlying mechanisms, the authors examined the expression of ALDH3A1 (aldehyde dehydrogenase) and UGDH (UDP-glucose-6-dehydrogenase), which have been previously shown to regulate the NADH/NAD⁺ ratio in Keap1-dependent cells (Weiss-Sadan et al, 2023) and found increased levels of ALDH3A1 upon Keap1 deletion in KPP, KPL and KPLP mice. In parallel, ALDH3A1 and UGDH positively correlate with Nrf2 protein levels and negatively with Keap1 protein levels in lung human cell lines, and lung cancer patients with high Nrf2, UGDH and ALDH3A1 expression have significantly better overall survival.

The findings illustrate the complex role of the Keap1/Nrf2 system in lung cancer, and highlight the need for very careful consideration in the design of future clinical trials targeting Keap1/Nrf2.

Major points:

The study is comprehensive and robust. It employs advanced methodologies and relevant in vivo mouse models. The conclusions are generally well supported by the data presented, except for the mechanistic details. My recommendation is that the authors exert caution when stating the underlying mechanism(s), which they attribute to reductive stress and mitochondrial dysfunction.

1. The observations with regard to ALDH3A1 and UGDH overexpression is correlative, and causation has not been established. Moreover, no markers of reductive stress, such as the NADH/NAD⁺, NADPH/NADP⁺, GSH/GSSG ratios, or mitochondrial function, have been examined. Does Nrf2 depletion in their cell culture models abolish reductive stress and restore cell proliferation? This is important, because Keap1 has other binding partners, in addition to Nrf2. What is the effect of ALDH3A1 and/or UGDH depletion, with and without functional Nrf2? Would pharmacological inactivation of Keap1 have a similar effect to that of genetic deletion?
2. Is the reductive stress limited to the tumors, or is it also evident in the surrounding normal lung tissue?
3. It is also unclear whether reductive stress affects the function of the E3 ligases and DUBs (e.g. Huwe1 and Ubr4) that were identified as Keap1-dependent genetic co-vulnerabilities. Establishing whether or not there might be a link is a crucial point as it relates to the most novel findings of the study.

Minor points:

1. Has the anti-Nrf2 antibody been validated? Many of the commercially available anti-Nrf2 antibodies are unspecific and require rigorous validation.
2. 'FP' is used to abbreviate 'free progression' as well as 'first progression'.
3. Line 253: 'Figure 1F-G' should be 'Figure 2F-G'.

Referee #2:

Summary:

Shah and colleagues, through the integration of single-target and multiplex CRISPR screening, uncovered the oncogenic role of KEAP1 in murine models of non-small cell lung cancer (NSCLC). They subsequently identified components of the UPS and specific E3 ubiquitin ligases (such as HUWE1 and HERC2) and DUBs (such as USP28) as dependencies in KEAP1 mutant tumors which they validated in vivo. These findings provide new avenues for NSCLC treatment, especially those resistant to immunotherapy and chemotherapy.

In summary, I believe this is an elegant study and an exciting concept. However, some of the conclusions drawn by the authors should be supported with stronger evidence for acceptance at EMBO.

Comments:

1. Please provide immunoblots or sequencing results to confirm efficiency of knockdown for select targets.
- 2, On lines 252-253, "In agreement with previous studies, Keap1 depletion in KPP and KPL cohorts resulted in a significant upregulation of Nrf2 abundance (Figure 1F-G)". What is currently described does not correspond to Figure 1F-G. Figure 1F-G depicts "the drop-out viability screen in KPP (F) and KPL (G) cell lines".
- 4 , In Figure 4b, Please indicate which region of the original image was enlarged
- 5 , Please provide references cited in lines 359 to 360, "Interestingly, UBR4 has been recently reported as a regulator of mitochondrial homeostasis".
- 6 , "Kaplan-Meier curves showed that lung cancer patients with high UBR4, HERC2 and HUWE1 expression with concurrent high TP53 and KEAP1 levels had significantly better first progression (FP) and overall survival (OS) (Figure 4E-G)".

The description of this sentence is confusing. Based on the results shown in Figure 4E-G, it seems that high levels of UBR4/TP53/KEAP1, HERC2/TP53/KEAP1, and HUWE1/TP53/KEAP1 correspond with worse survival

7. Some charts (such as Kaplan-Meier survival curve and volcano map of screening results) lack sufficient annotation. Please clarify the statistical parameters used

8. The models of reductive stress and ubiquitin signaling pathways in Figures 2 and 4 are helpful, but adding annotations would enable readers to understand them more quickly.

Referee #3:

In this manuscript by Shah et al., the authors present data where they identify KEAP1, via a limited CRISPR screen targeting components of the UPS system, as a proto-oncogene in the context of Kras-G12D mutation with Trp53-Lkb1 (KPL), Trp53-Pten (KPP), and Trp53-Lkb1-Pten (KPLP) tumor suppressor loss and a neutral tumorigenic modifier with Kras-G12D and Trp53 loss. A second CRISPR screen of paired sgRNAs of UPS system identified Usp28, Ubr4, Herc2, Pdz3, and Huwe1 as synthetically lethal to murine Kras-G12D;Trp53^{-/-}, Keap1^{-/-} tumors

That Keap1 acts as a proto-oncogene is surprising and novel. The identification of E3 ligases as KEAP1 co-vulnerability suggests that these proteins may be therapeutic targets in KEAP1 mutant human NSCLC. However, there are some significant concerns regarding the studies presented here.

1. Overall, the manuscript is descriptive with little mechanistic insight as to how E3 ubiquitin ligases are synthetically lethal with KEAP1 loss in the KP context. Also there are no mechanistic insights as to how KEAP1 may be a proto-oncogene in KPL and KPP contexts.

2. In the discussion, the manuscript makes a general claim that E3 ubiquitin ligases may be therapeutic targets in KEAP1 mutant cancer. However, the data presented is limited to the KP context alone and therefore, such broad claims cannot be made. There are no data presented for the synthetic lethality of E3 ubiquitin ligases in the context of KPKL or KPKP. Furthermore, all of the models in this study have p53 loss. Thus, the conclusions here cannot be generalized to those lung cancers that do not contain p53 mutations. Also, only Kras is used as an oncogenic driver here. Other oncogenic drivers such as mutant EGFR, ALK-fusions, ROS-fusions, etc have not been explored. The different oncogenic drivers may change the dependency of KEAP1 loss on E3 ubiquitin ligases. Thus, it's unclear how much the synthetic lethality of E3 ubiquitin ligases with KEAP1 loss can be generalized.

3. KEAP1 seems to be a proto-oncogene in the KPLK and KPKP murine context. However, human KPLK and KLK LUAD occur with KLK tumors occurring almost as frequently as KPL tumors. If the studies in this manuscript accurately model human KPL tumors, how can human KPLK tumors exist? Also, is KEAP1 a proto-oncogene in the KL context? A discussion of these discrepancies is needed at least.

4. It's unclear why KEAP1 loss in KP context is neutral regarding tumorigenesis but is a proto-oncogene in KPL and KPK contexts. There are no data nor any discussion as to why this might be.

5. The H&E and IHC images in Fig. 2-4 are poor quality and very difficult to assess. Fig. 2E, one cannot make out any tumors to assess whether loss of KEAP1 truly decreases tumor burden. In Fig 2F-I, the IHC of the target protein is poor. KEAP1 seems to increase in KPKL section when it should be lost, NRF2 should be nuclear but that's not clear, ALDH3 should be cytoplasmic but seems more nuclear. Similar issues are found in Fig. 3 for IHC and Fig. 4 for H&E images. This is critical as these data are the foundation for the manuscript's conclusions.

6. The in vivo mouse models are complex CRISPR models requiring replacement of wt KRAS with a KRAS-G12D and indels leading to loss of multiple tumor suppressors. As the IHC images are the only data for loss of KEAP1 of in vivo tumors, other means are needed to demonstrate KEAP1 loss. Furthermore, since multiple tumor suppressors are attempted to be knocked out, concurrent loss of the other tumor suppressors should be shown (i.e. p53, Lkb1, Pten) in the same cells that Keap1 is lost for the in vivo tumors or early transformed cells. This is critical to demonstrate as the conclusions of the manuscript depend on the concurrent loss of these suppressors with Keap1 in the same tumor cell. For the KPK, KPKL, and KPKPL cell lines, Western blot demonstrating loss of the tumor suppressors and KEAP1 should be shown.

7. The human survival curves are sometimes all lung cancer patients, sometimes NSCLC patients, and sometimes broken up into LUAD and LUSC patients. This creates confusion and it seems the manuscript seems to conflate LUAD, LUSC, NSCLC and lung cancer (which also would include SCLC) together. LUAD and LUSC are very different diseases with distinct genetics and biology. Perhaps a gene or protein may be significant in both but this is not how the manuscript present the data. It would probably be best if the human data focused on LUAD since the murine in vivo models and cell lines for KP and KPP are LUAD.

As noted in their previous paper (Hartmann et al., 2021), the KPL tumors were also all LUAD.

8. The CRISPR KPK model here shows that loss of KEAP1 is neutral in terms of tumorigenesis in the KP context whereas the CRISPR induced loss of KEAP1 in the KP GEMM demonstrated significant increase in tumor burden. A discussion should be added as to why such different outcomes are seen.

9. In Fig EV7 B-D, expression of protein levels comparing tumor vs normal is not illustrative since EV7 A shows elevation in the context of KEAP1 loss. Panels B-D should be comparing tumors with KEAP1 loss vs KEAP1 wt.

10. In Fig. 4E-G, unclear what the significance of survival curves of patients with high UBR4, HERC2, and HUWE1 with concurrent high TP53 and KEAP1 in the context of this manuscript. The data provided thus far is elevated Ubr4, Herc2, and Huwe1 in context of p53 and KEAP loss, which would mean low mRNA levels of TP53 and KEAP1. Also, the text re: these panels are incorrect. The text notes that patients with high UBR4, HERC2, and HUWE1 with concurrent high TP53 and KEAP1 have better FP and OS where the data shows that these patients have worse survival. Also, only data for OS are shown in Fig. 4.

11. MINOR: In Fig. 3F, the legend is incomplete as labels for orange and sky blue/teal dots are missing.

Referee #4:

This study focuses on NSCLC murine models (KP, KPL, and KPP) to investigate tumor dependencies. Through bioinformatics analysis of public human lung tumor datasets and a CRISPR drop-out viability screen across the three models, the study identifies the E3 ligase Keap1 as a critical dependency. Further investigation reveals that NRF2 and its targets, ALDH3 and UGDH, may contribute to reductive stress in Keap1-depleted tumors. Additional multiplex CRISPR screens uncover combinatorial vulnerabilities involving other E3 ligases and deubiquitinases (DUBs) alongside Keap1. The use of robust NSCLC murine models underscores the in vivo relevance of these findings. The study provides valuable insights into Keap1-altered NSCLC and identifies novel co-vulnerabilities that offer new therapeutic directions. While the data quality is high and the study is intriguing, there are conceptual and experimental questions that need to be addressed before publication.

Fig 2F: Unlike KPL, Keap1 levels in KPP appear less abundant, yet the drop-out screen consistently identifies Keap1 as a dependency in both models. Could the differential dependency on Keap1 be influenced by its expression levels, or are other genetic alterations dictating this dependency?

Is Keap1 downregulation-induced tumor growth reduction due to hyperactivation of oncogene (such as NRF2) induced cellular senescence or cell stress/death? In this case, Keap1 still functions as a tumor suppressor. Could NRF2 depletion partially rescue this phenotype?

Given the availability of Keap1 inhibitors, it would be valuable to investigate whether the regulation of Keap1 substrates or its binding proteins contributes to the observed Keap1-dependent phenotypes.

While ALDH3 and UGDH transcription/protein levels are monitored as markers of NRF2 activity, their role in regulating cellular ROS levels suggests that direct measurement of ROS levels would be necessary to confirm this activity.

Fig. 4: Does the depletion of Herc2, Ubr4, Pdzrn3, or Usp28 affect NRF2 or Keap1 levels in KP or KPK models? Do these E3 ligases or DUBs influence the same signaling pathways as Keap1/NRF2, or do they operate through distinct mechanisms that create co-vulnerabilities?

The study notes that Keap1 expression is downregulated in NSCLC tumors compared to adjacent normal tissues, despite Keap1 being a dependency for tumor growth. Can the authors provide explanations or speculations for why tumors reduce Keap1 expression, given its essential role?

Point by Point response to reviewers-

Referee #1:

This study is based on previous knowledge that genetic alterations in components of the UPS, including E3 ubiquitin ligases (E3s) and deubiquitinating enzymes (DUBs), which facilitate the maintenance of proteostasis, are common in cancer, facilitating tumor progression, but also creating potential therapeutic vulnerabilities. Using mouse lung cancer models, specifically KRAS-driven non-small cell lung cancers (NSCLC) with co-occurring deletions in TP53 (KrasG12D105 Tp53^{-/-}; KP), TP53 and LKB1 (KrasG12D Tp53^{-/-}-Lkb1^{-/-}; KPL), or TP53 and PTEN (KrasG12D Tp53^{-/-}-Pten^{-/-}; KPP), the authors have performed combinatorial CRISPR/Cas9 dropout viability screens, targeting members of the UPS. Surprisingly, they identified Keap1 as a therapeutic vulnerability, and several druggable E3s and DUBs as Keap1-dependent co-vulnerabilities, such as the E3s Herc2, Ubr4 30 and Huwe1, the depletion of which suppressed the development of Keap1-inactivated tumors. This was surprising, because loss-of-function mutations in Keap1, leading to constitutive activation of transcription factor Nrf2, occur frequently in aggressive NSCLC, conferring resistance to therapies, including chemotherapy, radiation therapy and immunotherapy.

To investigate the underlying mechanisms, the authors examined the expression of ALDH3A1 (aldehyde dehydrogenase) and UGDH (UDP-glucose-6-dehydrogenase), which have been previously shown to regulate the NADH/NAD⁺ ratio in Keap1-dependent cells (Weiss-Sadan et al, 2023) and found increased levels of ALDH3A1 upon Keap1 deletion in KPP, KPL and KPLP mice. In parallel, ALDH3A1 and UGDH positively correlate with Nrf2 protein levels and negatively with Keap1 protein levels in lung human cell lines, and lung cancer patients with high Nrf2, UGDH and ALDH3A1 expression have significantly better overall survival.

The findings illustrate the complex role of the Keap1/Nrf2 system in lung cancer, and highlight the need for very careful consideration in the design of future clinical trials targeting Keap1/Nrf2.

Major points:

The study is comprehensive and robust. It employs advanced methodologies and relevant *in vivo* mouse models. The conclusions are generally well supported by the data presented, except for the mechanistic details. My recommendation is that the authors exert caution when stating the underlying mechanism(s), which they attribute to reductive stress and mitochondrial dysfunction.

We thank the reviewer for the thoughtful and constructive feedback on our manuscript. In line with your suggestion, we have revised the text to present our mechanistic conclusions with greater clarity. In the revised manuscript, we provide experimental evidence linking Keap1 function to reductive stress. Specifically, our findings show that Keap1 inhibition leads to NRF2 activation, which induces Aldh3a1 expression, resulting in elevated NADH/NAD⁺ ratios and the onset of reductive stress.

The observations with regard to ALDH3A1 and UGDH overexpression is correlative, and causation has not been established. Moreover, no markers of reductive stress, such as the NADH/NAD⁺, NADPH/NADP⁺, GSH/GSSG ratios, or mitochondrial function, have been examined. Does Nrf2 depletion in their cell culture models abolish reductive stress and restore cell proliferation? This is important, because Keap1 has other binding partners, in addition to Nrf2. What is the effect of ALDH3A1 and/or UGDH depletion, with and without functional Nrf2? Would pharmacological inactivation of Keap1 have a similar effect to that of genetic deletion?

We thank the reviewer for this constructive comment. In the revised manuscript, we now provide mechanistic evidence linking Keap1 inactivation to reductive stress. Using both pharmacological inhibitors (KI696, Bardoxolone methyl, Omaveloxolone) and CRISPR-mediated Keap1 deletion, we observed Nrf2 activation, transcriptomic and proteomic upregulation of Aldh3a1 and Ugdh (Figure 2A-C, EV2A-D), and a consistent increase in the NADH/NAD⁺ ratio in all three cell lines and NADPH/NADP⁺ ratio in KPL and KPP. Knockdown of Nrf2 or Aldh3a1 restored NADH/NAD⁺ levels, confirming that the Keap1–Nrf2–Aldh3a1 axis drives reductive stress (Figure 2D-F, EV3A-F). Keap1 inhibition also impaired mitochondrial respiratory capacity but did not alter ROS levels in basal condition (Figure 2G, EV3G-H). Importantly, co-depletion of Nrf2 with Keap1 partially rescued cell viability, indicating both Nrf2-dependent and -independent pathways downstream of Keap1 play a role in regulating cell proliferation (Figure 2H and I, EV3I and J). Finally, pharmacological Keap1 inhibition resulted in increased nuclear NRF2 and cytosolic ALDH3A1 levels leading to reduced proliferation in multiple human lung cancer lines (H2170, A549, CALU1, H1299) (Figure AF2-4). Further, tumor microarray (TMA) analysis of lung cancer patient samples revealed an inverse correlation between KEAP1 and NRF2/ALDH3A1/UGDH in lung cancer samples. Tumors with low KEAP1 showed high NRF2, ALDH3A1, and UGDH, whereas high KEAP1 tumors displayed low levels (Fig. 4A–C). Together, we establish a mechanistic link between KEAP1 inactivation and reductive stress. Our findings demonstrate that Keap1 inhibition leads to NRF2 activation, which subsequently induces ALDH3A1 expression, resulting in elevated NADH/NAD⁺ ratios, leading to the onset of reductive stress and impaired cell proliferation.

2. Is the reductive stress limited to the tumors, or is it also evident in the surrounding normal lung tissue?

We assessed Nrf2, Aldh3a1, and Ugdh by IHC in tumors and adjacent lung tissue across genetic combinations (Figure EV5A–D). These markers were consistently very low in surrounding normal tissue compared to tumors. In summary, reductive stress signaling is restricted to tumors and does not extend to surrounding normal lung tissue.

3. It is also unclear whether reductive stress affects the function of the E3 ligases and DUBs (e.g. Huwe1 and Ubr4) that were identified as Keap1-dependent genetic co-vulnerabilities. Establishing whether or not there might be a link is a crucial point as it relates to the most novel findings of the study.

We agree that the potential impact of reductive stress on E3 ligases and DUBs such as Huwe1 and Ubr4 is an important and intriguing hypothesis. In our study, Keap1 modulation, which it induced reductive stress, did not lead to significant changes in the RNA or protein levels of these ligases or their known targets. This argues against a direct mechanistic link or compensatory effect. Instead, the observed Keap1-dependent genetic co-vulnerabilities may reflect secondary synthetic interactions arising from independent pathways.

Prior studies show that Huwe1 and Ubr4 can influence mitochondrial function through mechanisms such as mitophagy, proteasomal degradation of mitochondrial precursors, and ATM regulated processes related to oxidative stress but not necessarily driven by reductive stress (PMID: 40081620, PMID: 40531870, PMID: 38297121, PMID: 30217973). Given their broad cellular roles, it is plausible that vulnerabilities emerge in this context without direct modulation of their expression or activity.

In summary, while the hypothesis remains compelling, our current data do not support a direct connection between reductive stress and E3 ligase/DUB function at the RNA, protein, or inferred activity levels.

Additionally, analysis of publicly available datasets show that the identified ubiquitin modifiers are involved in a plethora of cancer related pathways. Hence, the co-depletion of these factors, especially in a Keap1 null context, could induce additional stress, resulting in the significant reduction in cell viability. We have presented these data below for reviewers only (Reviewers figure 1A-C).

Biological function of:

- PDZRN3: Loss of control over pathways associated with basal cell carcinoma, Hippo, or WNT signaling could impair cellular fitness.*
- UBR4: Impairment of homologous recombination, together with RAS, MAPK, and PI3K signaling, would impose significant stress on KRAS-driven NSCLC cells.*
- HERC2: it is an attractive target given its role in ferroptosis, homologous recombination, proteostasis, and splicing—pathways tumor cells heavily rely on under stress.*

Minor points: -

- 1. Has the anti-Nrf2 antibody been validated? Many of the commercially available anti-Nrf2 antibodies are unspecific and require rigorous validation.**

NRF2 antibody was verified by siRNA treatment (Reviewers figure 1D). Further, antibody was verified by cell treatment to ensure that the antibody binds to the antigen stated. Detection of altered subcellular localization of the target protein by cell treatment demonstrates antibody specificity. Immunofluorescence analysis using Nrf2 Polyclonal Antibody (Product # PA5-27882), shows nuclear translocation of NRF2 upon TBHQ treatment.

Cell treatment validation info.

<https://www.thermofisher.com/antibody/product/Nrf2-Antibody-Polyclonal/PA5-27882>

2. 'FP' is used to abbreviate 'free progression' as well as 'first progression'.

We acknowledge that 'FP' was previously used to denote both free progression and first progression. We apologize for this inconsistency. In the revised manuscript, we have standardized the terminology to ensure clarity and precision. For clarification, the term first progression survival curve is commonly used interchangeably with progression-free survival (PFS) curve. This curve represents the duration of time during which patients with a given disease, such as cancer, remain free from disease progression or death following initiation of treatment. It reflects the proportion of patients who have not experienced progression over time. While both expressions are sometimes used in the literature, to maintain terminological consistency within the manuscript, we have adopted the term first progression throughout.

3. Line 253: 'Figure 1F-G' should be 'Figure 2F-G'.

We apologize for the error in the figure citation. The text has been corrected to read: "In agreement with previous studies, Keap1 depletion in KPP and KPL cohorts resulted in a significant upregulation of Nrf2 abundance (Figure 3C) in the revised manuscript.

Referee #2:

Summary:

Shah and colleagues, through the integration of single-target and multiplex CRISPR screening, uncovered the oncogenic role of KEAP1 in murine models of non-small cell lung cancer (NSCLC). They subsequently identified components of the UPS and specific E3 ubiquitin ligases (such as HUWE1 and HERC2) and DUBs (such as USP28) as dependencies in KEAP1 mutant tumors which they validated in vivo. These findings provide new avenues for NSCLC treatment, especially those resistant to immunotherapy and chemotherapy.

In summary, I believe this is an elegant study and an exciting concept. However, some of the conclusions drawn by the authors should be supported with stronger evidence for acceptance at EMBO.

We thank the reviewer for the positive remarks on our study and for recognizing the novelty and potential impact of our concept. We appreciate the constructive feedback. In the revised manuscript, we have incorporated additional experimental data and clarifications to strengthen the mechanistic basis and address the points raised, ensuring that our conclusions are better supported by the evidence presented.

Comments:

1. Please provide immunoblots or sequencing results to confirm efficiency of knockdown for select targets.

The tumors were previously characterized, and sequencing data from our 2021 publication confirm concurrent loss of the other tumor suppressors. For clarity, we have now cited the 2021 publication (PMID: 33738287, 32128997, 35477555) in the revised manuscript to direct readers to the detailed characterization data (Reviewers Figure 2).

2. On lines 252-253," In agreement with previous studies, Keap1 depletion in KPP and KPL cohorts resulted in a significant upregulation of Nrf2 abundance (Figure 1F-G)". What is currently described does not correspond to Figure 1F-G. Figure 1F-G depicts "the drop-out viability screen in KPP (F) and KPL (G) cell lines".

We apologize for the error in the figure citation. The text has been corrected to read: In agreement with previous studies, Keap1 depletion in KPP and KPL cohorts resulted in a significant upregulation of Nrf2 abundance (Figure 3C) in the revised manuscript.

4. In Figure 4b, Please indicate which region of the original image was enlarged

We have now clearly indicated the region of the original image that was enlarged in revised figure 6A now.

5. Please provide references cited in lines 359 to 360, "Interestingly, UBR4 has been recently reported as a regulator of mitochondrial homeostasis".

We apologize for the oversight in omitting the citation. The referenced work has now been incorporated in the revised manuscript (PMID: 38297121).

6. "Kaplan-Meier curves showed that lung cancer patients with high UBR4, HERC2 and HUWE1 expression with concurrent high TP53 and KEAP1 levels had significantly better first progression (FP) and overall survival (OS) (Figure 4E-G)". The description of this sentence is confusing. Based on the results shown in Figure 4E-G, it seems that high levels of UBR4/TP53/KEAP1, HERC2/TP53/KEAP1, and HUWE1/TP53/KEAP1 correspond with worse survival.

We apologize for the error in the original description. The sentence has been corrected to: Kaplan-Meier curves showed that lung cancer patients with low UBR4, HERC2, and HUWE1 expression, along with concurrent low TP53 and KEAP1 levels, had significantly better overall survival (OS) (Figure 6D-F). Additionally, we have now included the previously missing First Progression (FP) data in Figure AF8F-H.

7. Some charts (such as Kaplan-Meier survival curve and volcano map of screening results) lack sufficient annotation. Please clarify the statistical parameters used

We appreciate the reviewer's feedback and have expanded the Methods section to detail all statistical parameters used. For most Kaplan-Meier survival curves, statistical analyses were performed using the KMplot.com platform, a widely used and validated resource cited in numerous high-impact publications (PMID: 32128997; 33956155). We have now clearly stated the statistical approaches applied for both the Kaplan-Meier analyses and the volcano plots in the revised

manuscript. We have also indicated the identity of specific probe used for KMplot.com platform facilitating the reproducibility of our results.

8. The models of reductive stress and ubiquitin signaling pathways in Figures 2 and 4 are helpful, but adding annotations would enable readers to understand them more quickly.

We have added annotations to the legends of Figures 4 and 6 in the revised manuscript to enhance clarity and facilitate quicker understanding for readers.

Referee #3:

In this manuscript by Shah et al., the authors present data where they identify KEAP1, via a limited CRISPR screen targeting components of the UPS system, as a proto-oncogene in the context of Kras-G12D mutation with Trp53-Lkb1 (KPL), Trp53-Pten (KPP), and Trp53-Lkb1-Pten (KPLP) tumor suppressor loss and a neutral tumorigenic modifier with Kras-G12D and Trp53 loss. A second CRISPR screen of paired sgRNAs of UPS system identified Usp28, Ubr4, Herc2, Pdzm3, and Huwe1 as synthetically lethal to murine Kras-G12D;Trp53^{-/-}, Keap1^{-/-} tumors

That Keap1 acts as a proto-oncogene is surprising and novel. The identification of E3 ligases as KEAP1 co-vulnerability suggests that these proteins may be therapeutic targets in KEAP1 mutant human NSCLC. However, there are some significant concerns regarding the studies presented here.

1. Overall, the manuscript is descriptive with little mechanistic insight as to how E3 ubiquitin ligases are synthetically lethal with KEAP1 loss in the KP context. Also there are no mechanistic insights as to how KEAP1 may be a proto-oncogene in KPL and KPP contexts.

During the revision of this manuscript, we have made a focused effort to further elucidate the mechanism of KEAP1 dependency in NSCLC.

While Keap1 loss itself is neutral in KP, it leads to increased levels of Nrf2, Aldh3a1, and Ugdh, indicating the induction of reductive stress. The observation that Usp28, Huwe1, Pdzn3, Ubr4, and Herc2 are synthetic lethal with Keap1 loss can be explained by at least two possibilities: a) these UPS members function as 'buffers' to reduce cellular stress and are required to maintain proliferation, or b) their depletion generates co-vulnerabilities that tumor cells cannot tolerate. Further, we have analyzed publicly available datasets and we refer the reviewer to our response to Reviewer 1, Comment 3.

Additionally, it is worth noting that crucial substrates, particularly proto-oncogenes, are tightly regulated by the ubiquitin system. A prime example is the WNT effector molecule β -catenin. Beyond its regulation by the E3 ligases β -TRCP (I/II), additional enzymes fine-tune its abundance, localization, and function. Notably, the E3 ligases RNF4, JADE1, UBR5, HUWE1, and FBXW7 contribute to this regulation. This example illustrates the complexity of the ubiquitin system and shows that, at least for

some proteins, inbuilt redundancies may exist that could represent exploitable vulnerabilities.

Our revised manuscript provides evidence that Keap1 functions as a proto-oncogene by supporting tumor cell proliferation and stress adaptation. Acute deletion or pharmacological inhibition of Keap1 impaired cell fitness, activated Nrf2, and induced redox stress, accompanied by upregulation of Jun and Nfkb1, reflecting elevated stress responses consistent with prior findings (Weiss-Sadan et al., 2023).

Beyond Nrf2 regulation, Keap1 controls redox homeostasis and ferroptosis, supporting rapidly dividing tumors under metabolic and replicative stress. Together, these results support a model in which Keap1 preserves tumor fitness, consistent with a proto-oncogenic role in Kras-driven NSCLC.

2. In the discussion, the manuscript makes a general claim that E3 ubiquitin ligases may be therapeutic targets in KEAP1 mutant cancer. However, the data presented is limited to the KP context alone and therefore, such broad claims cannot be made. There are no data presented for the synthetic lethality of E3 ubiquitin ligases in the context of KPKL or KPKP. Furthermore, all of the models in this study have p53 loss. Thus, the conclusions here cannot be generalized to those lung cancers that do not contain p53 mutations. Also, only Kras is used as an oncogenic driver here. Other oncogenic drivers such as mutant EGFR, ALK-fusions, ROS-fusions, etc have not been explored. The different oncogenic drivers may change the dependency of KEAP1 loss on E3 ubiquitin ligases. Thus, it's unclear how much the synthetic lethality of E3 ubiquitin ligases with KEAP1 loss can be generalized.

The role of KEAP1 loss in lung tumorigenesis appears highly context dependent. Consistent with our study, work reported by the group of Rocio Sotillo demonstrated that KEAP1 loss can be either tumor burden neutral or might partially suppress tumor formation in a KRAS-independent, TP53-wildtype EML-ALK4 model (doi.org/10.1101/2024.08.26.609730). Further evidence that KEAP1 loss is context-dependent arose from the work of the Katerina Politis group. In a Tp53fl/fl:Tet-On-EGFR^{L858R} mouse model of NSCLC they observed that CRISPR-mediated loss of KEAP1 was tumor burden neutral and that CRISPR mediated Keap1 loss in a genetic KP model reduced tumor burden (Foggetti et al, 2021). Taken together, these observations and the in vivo effects in our somatically engineered murine NSCLC models demonstrate that Keap1 biology in NSCLC is far more complex and functionally versatile than previously recognized.

As noted, in our study, experimental evidence was obtained only in the KP context, as the KPKL and KPKP mouse models did not develop tumors in vivo. We agree that, based on our current data, the findings are applicable specifically to models with p53 deletion and KRAS mutation.

In line with the reviewers' suggestion, we have revised the discussion to clearly state that our conclusions are limited to KRAS-mutant, p53-deficient backgrounds, and cannot be generalized to all KEAP1-mutant cancers without further experimental validation. Future studies will be important to determine whether the synthetic

lethality of E3 ligases with KEAP1 loss extend to tumors driven by other oncogenic drivers and whether these vulnerabilities can be therapeutically exploited.

3. KEAP1 seems to be a proto-oncogene in the KPLK and KPKP murine context. However, human KPLK and KLK LUAD occur with KLK tumors occurring almost as frequently as KPL tumors. If the studies in this manuscript accurately model human KPL tumors, how can human KPLK tumors exist? Also, is KEAP1 a proto-oncogene in the KL context? A discussion of these discrepancies is needed at least.

We thank the reviewer for highlighting KEAP1's role across mutational contexts and the differences between murine models and human LUAD. To explore this, we interrogated publicly available datasets, particularly TCGA. These analyses confirm co-occurrence of KEAP1 mutations with KRAS and LKB1/STK11, while highlighting mutual exclusivity with TP53 mutations or PTEN loss (reviewers figure 3). Thus, our models partially reflect the mutational patterns observed in patients. As most NSCLCs harbor TP53 mutations, we focused our analyses on this cohort.

The inability of KPLK and KPKP models to develop tumors in vivo may reflect biological differences in tumor initiation and progression, including sequential mutagenesis timing, selective pressures, or buffering effects from co-mutations. Additionally, it remains possible that TP53 loss acts as a key stratifier for KEAP1 function. Such factors may help explain the existence of human KPLK tumors despite their apparent absence in the murine models used here. We have included discussion of these possibilities in the revised manuscript to acknowledge the limitations of our models and to highlight directions for future investigation.

4. It's unclear why KEAP1 loss in KP context is neutral regarding tumorigenesis but is a proto-oncogene in KPL and KPK contexts. There are no data nor any discussion as to why this might be.

We appreciate the reviewer's insightful question regarding the differential roles of KEAP1 loss across genetic contexts. In the KP model, KEAP1 loss has been shown to accelerate lung tumorigenesis, primarily via activation of the KEAP1/NRF2 antioxidant stress response pathway. In contrast, the KPL and KPP models display a more complex metabolic landscape. Loss of LKB1 drives metabolic reprogramming that supports accelerated growth (Bourouh et al., 2022), and KRAS/PTEN mutations have been reported to transcriptionally upregulate NRF2 (Rojo et al., 2014; Tao et al., 2014). In these contexts, further KEAP1 depletion may exacerbate metabolic stress, suggesting a buffering role in maintaining ROS and redox balance. This could explain why KEAP1 appears proto-oncogenic in KPL and KPK tumors but neutral in KP tumors.

Importantly, our findings are consistent with independent observations from Rocio Sotillo and Katerina Politis group as described earlier in comment 2. These results, obtained in a KRAS-independent and TP53-wildtype NSCLC background, reinforce the conclusion that the impact of KEAP1 loss is highly context dependent.

5. The H&E and IHC images in Fig. 2-4 are poor quality and very difficult to assess. Fig. 2E, one cannot make out any tumors to assess whether loss of KEAP1 truly decreases tumor burden. In Fig 2F-I, the IHC of the target protein is poor. KEAP1 seems to increase in KPKL section when it should be lost, NRF2 should be nuclear but that's not clear, ALDH3 should be cytoplasmic but seems more nuclear. Similar issues are found in Fig. 3 for IHC and Fig. 4 for H&E images. This is critical as these data are the foundation for the manuscript's conclusions.

As suggested by the reviewer, we have improved H&E and IHC images and incorporated them in the revised manuscript in Figure 4 and 6.

In the situation of KPKL only 3 tumours were established 12 weeks post intratracheal installation of the respective AAV virus to induce tumour formation. All these tumours still expressed Keap1, suggesting that in the combinatorial loss of TP53, Lkb1/Stk11 in combination with Kras, Keap1 plays an important role in tumour initiation and its loss is poorly tolerated. This could explain the presence of Keap1-espacer cells, still expressing the E3-ligase. Additionally, as seen in the analysis of patient data provided for comment 3, are co-depletion events of KEAP1-STK11 and TP53 rare events in a human NSCLC cohort. This observation would be reflected in the in vivo SEMM data presented in this manuscript.

We refer the reviewer to our response to reviewer 1, minor comment 1 for NRF2 AB validation and localisation. ALDH3A1 Antibody validation has been provided in (reviewers figure 4). Validation and IHC-staining was done and provided by [humanproteinatlas](https://www.proteinatlas.org/ENSG00000108602-ALDH3A1/summary/antibody) (<https://www.proteinatlas.org/ENSG00000108602-ALDH3A1/summary/antibody>). Localization was taken from [genecards.org](https://www.genecards.org/cgi-bin/carddisp.pl?gene=ALDH3A1#:~:text=ALDH3A1%20(Aldehyde%20Dehydrogenase%203%20Family,and%20Nuclear%20receptors%20meta%2Dpathway.)) ([https://www.genecards.org/cgi-bin/carddisp.pl?gene=ALDH3A1#:~:text=ALDH3A1%20\(Aldehyde%20Dehydrogenase%203%20Family,and%20Nuclear%20receptors%20meta%2Dpathway.\)](https://www.genecards.org/cgi-bin/carddisp.pl?gene=ALDH3A1#:~:text=ALDH3A1%20(Aldehyde%20Dehydrogenase%203%20Family,and%20Nuclear%20receptors%20meta%2Dpathway.)))

6. The in vivo mouse models are complex CRISPR models requiring replacement of wt KRAS with a KRAS-G12D and indels leading to loss of multiple tumor suppressors. As the IHC images are the only data for loss of KEAP1 of in vivo tumors, other means are needed to demonstrate KEAP1 loss. Furthermore, since multiple tumor suppressors are attempted to be knocked out, concurrent loss of the other tumor suppressors should be shown (i.e. p53, Lkb1, Pten) in the same cells that Keap1 is lost for the in vivo tumors or early transformed cells. This is critical to demonstrate as the conclusions of the manuscript depend on the concurrent loss of these suppressors with Keap1 in the same tumor cell. For the KPK, KPKL, and KPKPL cell lines, Western blot demonstrating loss of the tumor suppressors and KEAP1 should be shown.

The tumors were previously characterized, and sequencing data from our 2021 publication confirm concurrent loss of the other tumor suppressors. For clarity, we have now cited the 2021 publication (PMID: 33738287, 32128997, 35477555) in the revised manuscript to direct readers to the detailed characterization data (reviewers' figure 2 and 5).

7. The human survival curves are sometimes all lung cancer patients, sometimes NSCLC patients, and sometimes broken up into LUAD and LUSC patients. This

creates confusion and it seems the manuscript seems to conflate LUAD, LUSC, NSCLC and lung cancer (which also would include SCLC) together. LUAD and LUSC are very different diseases with distinct genetics and biology. Perhaps a gene or protein may be significant in both but this is not how the manuscript present the data. It would probably be best if the human data focused on LUAD since the murine in vivo models and cell lines for KP and KPP are LUAD. As noted in their previous paper (Hartmann et al., 2021), the KPL tumors were also all LUAD.

We apologies for the confusion in nomenclature. Our study focuses exclusively on NSCLC (excluding SCLC). All human analysis curves labelled “lung cancer” were performed on a cohort of NSCLC samples (as example: for Kaplan-Meier curves a combined cohort of 1,308 LUAD and 931 LUSC samples was used). While LUAD and LUSC have distinct genetics and biology, our findings indicate that the dependencies described here are driven primarily by redox status and/or KEAP1–NRF2 pathway activity rather than histological subtype. Both LUAD and LUSC samples in our cohort generally exhibited high reductive stress as stated in the manuscript:

“As previously reported [18, 32, 33], comparing transcriptomic signatures in wild-type and KEAP1 (inactivation)/NFE2L2 (activation) mutant LUAD (Figure 4D) and LUSC (Figure EV10A) patient samples revealed upregulation of reductive stress markers”.

We did not restrict our analyses to LUAD alone because our data show that the observed vulnerabilities are present in both LUAD and LUSC and correlate with reductive stress/KEAP1–NRF2 activity rather than histology. Limiting the analysis to LUAD would therefore omit relevant findings and reduce the generalizability of our conclusions.

Given the absence of significant differences between LUAD and LUSC in our key analyses, we presented the combined data to avoid redundancy. When introducing novel findings—such as the synthetic lethal candidates UBR4 and HUWE1—we analysed both subtypes separately (e.g., Figure EV13) and again observed no histology-specific effects.

We also wish to clarify that in our prior work (Hartmann et al., 2021; Prieto-Garcia et al., 2020), KP tumors were predominantly LUAD with occasional transdifferentiation to LUSC, while KPL and KPP models produced both LUAD and LUSC tumors. Please find the section of the original article below:

*While mutations of *Apc* and *Keap1* led to the development of ADC, targeting of *Stk11/Lkb1* and *Pten* also led to the development of SCC tumors (*Krt5*⁺, *P63*⁺, *TTF1*⁺) (Figure 4D; Prieto-Garcia et al., 2020).*

*Furthermore, we observed that the activation of the *Nrf2* pathway in primary murine tumours depleted for *STK11/LKB1* and *PTEN*. Please find the relevant section of the paper below:*

Targeting of Stk11/Lkb1 significantly upregulated tumor burden and tumor cell proliferation. These tumors also showed elevated protein abundance of Nrf2/Nfe2l2 (Figure 4E). Loss of Stk11/Lkb1 impacts tumor cell metabolism and increases ROS production, while elevated abundance of Nrf2 establishes ROS homeostasis, enabling tumor cells to grow under stress conditions (Ma, 2013; Faubert et al., 2014; Kaufman et al., 2014; Galan-Cobo et al., 2019). Lastly, loss of Pten strongly synergized with mutations in Trp53 and KRas. Tumors devoid of Pten accumulated Nrf2/Nfe2l2 as well, which is in line with published data (Figure 4E; Rojo et al., 2014; Best et al., 2018).

This explains our use of three independent CRISPR screening systems: KP and KPP LUAD cell lines, and KPL LUSC cell lines. Across all models, vulnerabilities correlated more strongly with reductive stress and KEAP1–NRF2 status than with histology.

In summary, our analyses and validations demonstrate that vulnerabilities such as UBR4 and HUWE1 are linked to reductive stress and KEAP1–NRF2 pathway activity rather than to a specific histological subtype. We have clarified this point in the revised manuscript to prevent further confusion.

8. The CRISPR KPK model here shows that loss of KEAP1 is neutral in terms of tumorigenesis in the KP context whereas the CRISPR induced loss of KEAP1 in the KP GEMM demonstrated significant increase in tumor burden. A discussion should be added as to why such different outcomes are seen.

The observation that loss of Keap1 is neutral in a KRAS–TP53 background aligns with our analysis of publicly available patient data, where KEAP1 loss did not significantly affect survival over time (Figure EV4D). Furthermore, we observed that KEAP1 loss is mutually exclusive with TP53 and PTEN co-mutations, a finding validated in our murine model fully relying on somatic engineering of patient-relevant mutations.

In similar lines, inactivation of Keap1 in EGFR-driven lung tumors (EGFR; p53; Cas9) or KRAS-driven lung tumors (KrasG12D; p53; Cas9) was neutral and did not significantly alter tumor growth, consistent with our findings (PMID: 33707235).

In contrast, recent studies using GEMM models of LUAD combined with lentiviral delivery of Cre, SpCas9, and sgRNA targeting KEAP1 (Thales Papagiannakopoulos et al., 2017, 2023, 2024) reported increased tumor burden. Unlike these hybrid models, our murine model relies entirely on somatic engineering of epithelial cells, including deletion of tumor suppressors and precise editing of endogenous KRAS to KRAS^{G12D}. While the overall frequency of editing events is lower compared to GEMMs, the tumor niche more closely resembles the patient environment, with all surrounding cells genetically wild type. Notably, in our model every epithelial cell is initially wild type for KRAS, whereas GEMM cells are heterozygous.

We previously compared both base genetic models (KP-GEMM vs. KP-CRISPR) and, while they showed a high degree of similarity, we observed differences—particularly in growth factor signaling—that may contribute to the discrepancies in tumorigenesis (PMID 33738287). Additionally, although sgRNA sequences differed

between studies, the use of two independent Cas9 systems in vivo (SpCas9 and SaCas9) and SpCas9 in cellulo—where we observed consistent phenotypes—supports the robustness of the biological consequences of KEAP1 loss in NSCLC.

Minor point:

Lastly, several studies report that lentiviral integration occurs preferentially in intronic regions of gene-rich chromosomes (PMID: 21386821), which could influence overall cellular fitness and contribute to differences observed in hybrid GEMM models.

9. In Fig EV7 B-D, expression of protein levels comparing tumor vs normal is not illustrative since EV7 A shows elevation in the context of KEAP1 loss. Panels B-D should be comparing tumors with KEAP1 loss vs KEAP1 wt.

The goal of this analysis is to show that UBR4, HERC2, and HUWE1 could be suitable targets in tumors with high reductive stress, independent of tumor histology, as these proteins are generally highly expressed in cancer compared to normal tissue.

In our previous analysis of gene expression in human tumors with KEAP1 mutations, we did not detect notable differences in the expression of these targets. This aligns with our proteomic and RNA-seq data, which indicate that KEAP1 inhibition does not significantly alter their expression.

We therefore conclude that their elevated expression in cancer is likely driven by factors unrelated to KEAP1 status, making them potential vulnerabilities in tumors experiencing increased reductive stress, regardless of histological subtype.

10. In Fig. 4E-G, unclear what the significance of survival curves of patients with high UBR4, HERC2, and HUWE1 with concurrent high TP53 and KEAP1 in the context of this manuscript. The data provided thus far is elevated Ubr4, Herc2, and Huwe1 in context of p53 and KEAP loss, which would mean low mRNA levels of TP53 and KEAP1. Also, the text re: these panels are incorrect. The text notes that patients with high UBR4, HERC2, and HUWE1 with concurrent high TP53 and KEAP1 have better FP and OS where the data shows that these patients have worse survival. Also, only data for OS are shown in Fig. 4.

We apologize for the error in the text. We have corrected the statement to: “Kaplan–Meier curves showed that lung cancer patients with low UBR4, HERC2, and HUWE1 expression, concurrent with low TP53 and KEAP1 levels, had significantly better first progression (FP) and overall survival (OS) (Figure 5D–F).” The manuscript figures and text have been updated accordingly in the revised version.

Additionally, we have now included the previously missing First Progression (FP) data in Figure AF8F–H.

11. MINOR: In Fig. 3F, the legend is incomplete as labels for orange and sky blue/teal dots are missing.

We apologize for the oversight. The figure legend for Fig. 3F has been updated to include labels for the orange and sky blue/teal dots in the revised manuscript.

Referee #4:

This study focuses on NSCLC murine models (KP, KPL, and KPP) to investigate tumor dependencies. Through bioinformatics analysis of public human lung tumor datasets and a CRISPR drop-out viability screen across the three models, the study identifies the E3 ligase Keap1 as a critical dependency. Further investigation reveals that NRF2 and its targets, ALDH3 and UGDH, may contribute to reductive stress in Keap1-depleted tumors. Additional multiplex CRISPR screens uncover combinatorial vulnerabilities involving other E3 ligases and deubiquitinases (DUBs) alongside Keap1. The use of robust NSCLC murine models underscores the in vivo relevance of these findings. The study provides valuable insights into Keap1-altered NSCLC and identifies novel co-vulnerabilities that offer new therapeutic directions. While the data quality is high and the study is intriguing, there are conceptual and experimental questions that need to be addressed before publication.

Thank you for your positive feedback on the data quality and the intriguing nature of our study. We appreciate your thoughtful input and will address the conceptual and experimental questions in the revised manuscript.

1. Fig 2F: Unlike KPL, Keap1 levels in KPP appear less abundant, yet the drop-out screen consistently identifies Keap1 as a dependency in both models. Could the differential dependency on Keap1 be influenced by its expression levels, or are other genetic alterations dictating this dependency?

We appreciate the reviewer's insightful question regarding the differential roles of KEAP1 loss across genetic contexts. While functional outcomes cannot be determined solely based on Keap1 expression levels across cell lines, it is plausible that dependency on Keap1 is influenced by additional genetic alterations.

The KPL and KPP models exhibit a more complex metabolic landscape due to the loss or inactivation of tumor suppressors LKB1 and PTEN, respectively. Genetic loss of LKB1 promotes metabolic reprogramming that supports increased cancer cell growth and division (Bourouh et al., 2022), while PTEN mutations have been reported to transcriptionally upregulate NRF2 (Rojo et al., 2014; Tao et al., 2014). Given that KPL and KPP tumors are already metabolically rewired, KEAP1 depletion may exacerbate metabolic stress and increase vulnerabilities linked to reductive stress. The influence of Keap1 expression levels in this context may be important but remains beyond the scope of the current study. These findings underscore the context-dependent nature of KEAP1's role, with additional co-mutations modulating tumor biology and influencing the consequences of KEAP1 loss.

2. Is Keap1 downregulation-induced tumor growth reduction due to hyperactivation of oncogene (such as NRF2) induced cellular senescence or cell stress/death? In this case, Keap1 still functions as a tumor suppressor. Could NRF2 depletion partially rescue this phenotype?

We examined the effect of Keap1 depletion using CRISPR/Cas9 gene editing. Consistent with CRISPR screen data, genetic inactivation of Keap1 led to a significant reduction in cell viability (Figure 2H, EV3I). Interestingly, co-depletion of

NRF2 along with Keap1 partially restored cell viability, suggesting that NRF2 activation contributes to the proliferative defect, but that NRF2-independent pathways downstream of Keap1 also play a role in regulating cell proliferation (Figure 2I, EV3J).

To rule out the involvement of senescence in Keap1 depletion–induced growth reduction, we performed β -galactosidase staining. However, Bardoxolone methyl treatment did not induce senescence, suggesting the involvement of alternative cell death pathways (Reviewers Figure 6A). Analysis of RNA-Seq and proteomics data revealed upregulation of several ferroptosis markers at both the RNA and protein levels (such as Hmox1, Slc3a2, Slc7a11), suggesting induction of ferroptosis upon Keap1 inhibition (Reviewers Figure 6B).

To further strengthen our findings, we employed clinically relevant Keap1 inhibitors—Bardoxolone methyl and Omaveloxolone—to assess their impact on ferroptosis across four lung human cancer cell lines (H2170 (KEAP1 truncated at aa336)), A549 (KEAP1 G333C mutant), H1299, and CALU1 (both KEAP1 wt)). Immunofluorescence-based quantification revealed increased NRF2 expression upon treatment with these inhibitors. In addition, we observed significant upregulation of ferroptosis-associated markers, including Hmox1, Acsl4, and Slc7a11, indicating activation of ferroptosis as a cell death pathway following Keap1 inhibition (Reviewer Figure 7 and 8).

3. Given the availability of Keap1 inhibitors, it would be valuable to investigate whether the regulation of Keap1 substrates or its binding proteins contributes to the observed Keap1-dependent phenotypes.

To address the potential impact of KEAP1 substrates on KEAP1-dependent phenotypes, we performed whole-cell proteomic analysis in the presence of the KEAP1 inhibitor KI696 (Reviewers Figure 9A). This analysis revealed upregulation of various KEAP1 substrates upon KI696 treatment. Based on our latest data (Figure 2 and EV2-5, AF2-5), we conclude that the induction of reductive stress is primarily dependent on NRF2 activation. However, NRF2 and other KEAP1 substrates or interactors likely contribute to the proliferation defect, although identifying these additional factors remains beyond the scope of the current study.

4. While ALDH3 and UGDH transcription/protein levels are monitored as markers of NRF2 activity, their role in regulating cellular ROS levels suggests that direct measurement of ROS levels would be necessary to confirm this activity.

We assessed whether KEAP1 inhibition affected reactive oxygen species (ROS) levels. Interestingly, KEAP1 inhibition did not cause any measurable change in cellular ROS levels under basal conditions in KP, KPL, and KPP cell lines (Figure EV3H). We believe that the absence of increased ROS, together with elevated expression of NRF2 and its targets, shifts the cellular redox balance toward reductive stress.

5. Fig. 4: Does the depletion of Herc2, Ubr4, Pdzn3, or Usp28 affect NRF2 or Keap1 levels in KP or KPK models? Do these E3 ligases or DUBs influence the same signaling pathways as Keap1/NRF2, or do they operate through distinct mechanisms that create co-vulnerabilities?

Crispr constructs targeting Herc2, Huwe1, Pdzn3, Ubr4 and Usp28, with or without a second guide RNA targeting Keap1, were cloned into a plasmid backbone containing RFP as a marker. Successful cloning was confirmed by sanger sequencing. These constructs were transiently transfected into KP, KPL, and KPP cell lines and were also used to produce adeno-associated virus (AAV) particles for subsequent cell infection. Transient transfection did not yield detectable RFP-positive cells. Thus, we switched to AAV infections. However, less than 5% of the population showed RFP-positive cells,

Since transient transfection did not yield sufficient cells and given the rapid decline in transduced cells we consider to use fluorescent activated cell sorting (FACS) to sort and enrich for transduced cells. Unfortunately, we were not able to acquire sufficient cell numbers positive for RFP signal for effective sorting. Attempts to isolate and expand single RFP-positive clones were unsuccessful, as no RFP expression was observed after clonal picking.

To allow for a more stringent selection we recloned the single and combinatorial sgRNA constructs into a plasmid containing a puromycin resistance cassette. Following transient transfection or AAV-mediated infection of KP, KPL and KPP cells, puromycin selection was applied. Selection yielded individual clones with very low occurrence (<5 clones per 10cm plate), immunoblot analysis against the identified co-dependency proteins to KEAP1 showed no evidence of target protein knockdown (due to gene deletion) in the cultivated clones. We therefore hypothesize that the majority of transduced cells may have been lost, while non-infected cells acquired puromycin resistance, leading to survival without effective gene knockdown. This indicates a high dependency on the aforementioned factors for cell survival in the used genotypes.

Analyzing publicly available data sets we checked for co-occurring alterations of Keap1 with the members of the UPS; Herc2, Huwe1, Pdzn3, Ubr4 and Usp28 (Reviewer Figure 9B). Neither in humans LUAC nor in LUSC we have significant co-occurrence of alterations in the aforementioned genes. This further hints to a co-dependency in the development or maintenance of NSCLC.

Regarding USP28, we explore its loss of function in the human SCC line A431. Here the depletion of the DUB upregulated ALDH3A1, downregulated SLC7A11, but had no significant effect on either KEAP1 or NRF2 (Reviewer figure 10a). This could argue towards the induction of metabolic stress, which could be governed, in part, by KEAP1-NRF2. Indeed, we report that USP28, via SREBP2, contributes to the control of metabolic stress, especially in SCC (<https://doi.org/10.1038/s41418-023-01173-6>). This observation further suggests that the identified factors induce co-vulnerabilities in cells.

Further, we have analyzed publicly available datasets and we refer the reviewer to our response to Reviewer 1, Comment 3.

This is further supported by publicly available data (DEPMAP). Here combinatorial loss of KEAP1 with PDZRN3, UBR4, HERC2 or HUWE1 in 128 human NSCLC negatively affect cell line growth (Reviewer figure 10b).

6. The study notes that Keap1 expression is downregulated in NSCLC tumors compared to adjacent normal tissues, despite Keap1 being a dependency for tumor growth. Can the authors provide explanations or speculations for why tumors reduce Keap1 expression, given its essential role?

We thank the reviewer for this insightful comment. Upon reanalysis of publicly available datasets, we found that, contrary to the initial statement, KEAP1 expression is in fact elevated in both adenocarcinoma (ADC) and squamous cell carcinoma (SCC) according to TCGA data. We have included these results in the revised figure EV4B and C.

Furthermore, analysis of a publicly available spatial transcriptomic dataset confirms the increase of KEAP1 expression in tumor cells. For this, we reanalyzed a dataset deposited by 10x Genomics using the Visium platform (<https://www.10xgenomics.com/datasets?configure%5BhitsPerPage%5D=50&configure%5BmaxValuesPerFacet%5D=1000&query=lung>). The H&E overview image is shown alongside a K-means 3-cluster separation, based on comparable gene expression profiles of the individually highlighted RNA capture areas. The accompanying heatmap depicts global gene expression patterns of the identified clusters—cancer (here a SCC), stromal, alveolar type II (AT2), and trachea/bronchioles. Volcano plots derived from these regions highlight representative marker genes for the three major tissues analyzed (cancer: TP63, EGFR, SLC7A11, EPCAM; AT2: SFTPC, SFTPA; bronchiole: SCGB1A1) (reviewer figure 11A and B).

Next, we sorted enriched genes in the tumor area versus normal tissue for ubiquitin-modifying enzymes, including E1, E2, E3, and DUBs, and represented the data as a volcano plot. Among the highlighted genes, such as UCHL1, PRAME, and RNF168—which were elevated in tumor cells—we also identified KEAP1 as significantly upregulated, consistent with the TCGA results presented above (reviewer figure 12A). Furthermore, UMAP visualization of this dataset confirmed the elevated abundance of KEAP1 and NFE2L2/NRF2 specifically in cancer cells, compared with AT2, stromal, or bronchiolar cells (reviewer figure 12B and C).

We also analyzed a publicly available single-cell RNA sequencing dataset from a time-resolved murine NSCLC cohort of KRAS or KRAS/TP53 tumors (PMID: 32707077). In this dataset, KEAP1 expression was upregulated in a time-dependent manner, particularly in KP tumors, peaking between 12 and 20 weeks post-infection. Similarly, UGDH, ALDH3A1, and several identified E3 ligases and DUBs also showed increased expression over time. Notably, UBR4 was elevated early in TP53 wild-type tumors and remained elevated in KP tumors (reviewer figure 13).

Taken together, these findings support the conclusion that tumors—particularly KRAS-driven NSCLC—upregulate KEAP1 and likely require it to balance ROS and redox stress.

Reviewers Figure 1

Figure for reviewers removed

Reviewers Figure 2

A

Representative immunohistochemical staining against encoding sgRNA targets Keap1, Lkb1 and Pten in KP, KP-Keap1, KP-Lkb1 or KP-Pten. Quantification of Keap1 positive to negative tumors in mice infected with AAV encoding sgRNA targeting Keap1. Quantification of Lkb1/Stk11 positive to negative tumors in mice infected with AAV encoding sgRNA targeting Lkb1/Stk11. Quantification of Pten positive to negative tumors in mice infected with AAV encoding sgRNA targeting Pten.

B

Sanger sequencing of representative tumor explants 12 weeks post intratracheal instillation of AAV containing sgRNA to delete Trp53, as well as encoding sgRNA and repair template for CRISPR mediated genome editing to KRasG12D(KPCRISPR). Sequencing primers bind 500bp proximal and distal to sgRNA sequences. Representative DNA gel electrophoresis image of PCR of genomic DNA of Trp53 and KRas locus, compared to KP GEMM DNA after recombination.

C

Reviewers Figure 3

The analysis tested 10 pairs between the 5 tracks in the OncoPrint.

Mutual exclusivity Co-occurrence Significant only

Columns

A	B	Neither	A Not B	B Not A	Both	Log2 Odds Ratio	p-Value	q-Value	Tendency
KEAP1	STK11	900	126	74	44	2.086	<0.001	<0.001	Co-occurrence
KRAS	STK11	830	196	55	63	2.278	<0.001	<0.001	Co-occurrence
KRAS	KEAP1	764	210	121	49	0.559	0.047	0.052	Co-occurrence
TP53	PTEN	354	688	14	88	1.693	<0.001	<0.001	Co-occurrence

Mutual exclusivity Co-occurrence Significant only

Columns

A	B	Neither	A Not B	B Not A	Both	Log2 Odds Ratio	p-Value	q-Value	Tendency
KEAP1	PTEN	882	160	92	10	-0.739	0.146	0.146	Mutual exclusivity
KRAS	TP53	226	142	659	117	-1.823	<0.001	<0.001	Mutual exclusivity
KRAS	PTEN	787	255	98	4	-2.989	<0.001	<0.001	Mutual exclusivity
STK11	PTEN	926	116	100	2	-2.647	0.002	0.002	Mutual exclusivity
TP53	STK11	300	726	68	50	-1.719	<0.001	<0.001	Mutual exclusivity
TP53	KEAP1	301	673	67	103	-0.540	0.033	0.041	Mutual exclusivity

Reviewers Figure 4

ALDH3A1 localizations

ALDH3A1 [ENSP00000411821]

Aldehyde dehydrogenase, dimeric NADP-prefering; ALDHs play a major role in the detoxification of alcohol-derived acetaldehyde (Probable). They are involved in the metabolism of corticosteroids, biogenic amines, neurotransmitters, and lipid peroxidation (Probable). Oxidizes medium and long chain aldehydes into non-toxic fatty acids. Preferentially oxidizes aromatic aldehyde substrates. Comprises about 50 percent of corneal epithelial soluble proteins (By similarity). May play a role in preventing corneal damage caused by ultraviolet light (By similarity).

Synonyms: ALDH3A1, ALDH3A1p, hALDH3A1, A8MYB8, C9JKT2 ...

Linkouts: STRING Pharos UniProt

Knowledge

Name	Source	Evidence	Confidence
Plasma membrane	HPA	IDA	★★★★☆
Extracellular space	UniProtKB	HDA	★★★★☆
Endoplasmic reticulum	LIFEdb	IDA	★★★★☆
Cytosol	HPA	IDA	★★★★☆
Cytosol	HPA	IDA	★★★★☆
Cytosol	HPA	IDA	★★★★☆
Plasma membrane	HPA	IDA	★★★★☆
Plasma membrane	HPA	IDA	★★★★☆
Cytoplasm	GO_Central	IBA	★★★★☆

Lung cancer

HPA051150

Male, age 61
Lung (T-28000)
Squamous cell carcinoma,
NOS (M-80703)
Patient id: 2354

Tumor cells

Staining: **Medium**
Intensity: **Strong**
Quantity: **<25%**
Location: **Cytoplasmic/
membranous
nuclear**

GENERAL INFORMATION ¹	
Gene name ¹	ALDH3A1
Gene description ¹	Aldehyde dehydrogenase 3 family member A1
Protein class ¹	Enzymes Metabolic proteins
Predicted location ¹	Intracellular
Number of transcripts ¹	10
HUMAN PROTEIN ATLAS INFORMATION ¹	
Main location ¹	Localized to the Plasma membrane (approved), Cytosol (enhanced) View proteome in REACTOME
Additional location ¹	In addition localized to the Nucleoplasm (approved), Vesicles (approved) View proteome in REACTOME
Single-cell variation ¹	Single-cell variation in protein expression observed.
Cell cycle dependency ¹	Variation in protein and transcript expression do not correlate to the cell cycle.
Reliability score ¹	Supported
Antibodies ¹	HPA051150, HPA063783

SHOW MORE

Reviewers Figure 5

To analyse the concurrent loss of LKB1/Stk11 and PTEN with Keap1 we performed IHC staining against LKB1/Stk11 and PTEN in the tumour bearing animals. As a control KPL or KPP was used respectively. In Reviewer Figure 15 exemplary high magnification images of positive (upper) and negative (lower) tumors are shown for LKB1 (left) and PTEN (right). When targeting multiple tumour suppressors, we could not detect LKB1 positive tumours in KPKL or KPLP. Additionally, no PTEN positive tumours were detected in KPKP and KPLP.

Reviewers Figure 6

Figure for reviewers removed

Reviewers Figure 7

Figure for reviewers removed

Reviewers Figure 8

Figure for reviewers removed

Reviewers Figure 9

A

B

Lung Adenocarcinoma (TCGA, PanCancer Atlas n = 507)

Lung Squamous Cell Carcinoma (TCGA, PanCancer Atlas n = 469)

Reviewers Figure 10

Volcano plot shUSP28 highlighting Redox/ferroptosis genes

Reviewers Figure 11

Figure for reviewers removed

Reviewers Figure 12

Figure for reviewers removed

Reviewers Figure 13

Prof. Ivan Dikic
Institute of Biochemistry II, Faculty of Medicine, Goethe University, Frankfurt, Germany
Institute of Biochemistry II
Theodor-Stern-Kai 7
Frankfurt 60590
Germany

28th Nov 2025

Re: EMBOJ-2024-119715R
Targeting Ubiquitin Signaling Vulnerabilities in KEAP1-Inactivated Lung Cancer

Dear Ivan and Markus,

Thank you again for submitting your revised manuscript to The EMBO Journal. Three of the original referees have now reviewed it once more, and their comments are copied below. As you will see, all referees appreciate your revisions and are now in principle favorable of publication. Referees 1 and 3 nevertheless still retain a few specific concerns that should be clarified prior to acceptance. In this respect, any data you would have to dispel referee 1's worries regarding Nrf2 antibody detection would be very helpful. For referee 3, addressing the issues with image quality and statistical analyses would be important; while the remainder of the points could be addressed by textual/presentational changes and responded to in a final referee response letter.

Based on these reports, I decided to return the manuscript to you for an additional round of minor revision, in which I would invite you to incorporate the remaining suggestions, as well as the following formal/editorial issues:

- Please adjust the order of the manuscript sections, and also make sure to use the correct section headers: Title page with complete author information, Abstract, Keywords, Introduction, Results, Discussion, Methods, Data Availability, Acknowledgements, Disclosure and Competing Interests Statement, References, Main Figure Legends, Tables, Expanded Figure Legends.
- Please reduce the number of keywords on the abstract page to five (ideally choosing broad general terms).
- Please rename the Conflict of Interest section into "Disclosure and Competing Interests Statement", in accordance with our updated Guide to Authors (<https://www.embopress.org/competing-interests>), and amend it with the statement "Ivan Dikic is a member of the Advisory Editorial Board of The EMBO Journal. This has no bearing on the editorial consideration of this article for publication."
- As we are switching from a free-text author contribution statement towards a more formal statement based on Contributor Role Taxonomy (CRediT) terms, please remove the present Author Contribution section and instead specify each author's contribution(s) directly in the Author Information page of our submission system during upload of the final manuscript. See <https://casrai.org/credit/> for more information.
- Please carefully go through the reference list and make sure that each reference is complete with citation year, volume, and page/eLocator numbers - this information is currently missing for several of them. Also, please make sure to only include DOIs or "preprint" labels for studies that are only available on preprint servers, but not for those that already have a formal journal reference.
- In the Data Availability section, please remove the referee access information, and ensure that the data are becoming publicly available at this point. Furthermore, please include (a) direct link(s) to the database(s) where data have been deposited (suggested wording: "The [structural coordinates | microarray | mass spectrometry] data from this publication have been deposited to the [name of the database] database [URL] and assigned the identifier [accession | permalink | hashtag].").
- In the Figure Legends section, please label the EV Figures with "Figure EV1/2/3...", and make sure to also stick to this nomenclature when referencing these figures throughout the text.
- Figure AF1-AF8 should not be uploaded as individual Figure files, but compiled in a single "Appendix" PDF, with each Appendix Figure legend appearing directly underneath instead of in the main text. Please also change their naming and callouts to "Appendix Figure S1/2/3..." throughout the Appendix and the main text. Finally, please head the Appendix PDF with a title page stating "Appendix for " and a brief table of contents with the page numbers for the listed items.
- Please note that Source data files need to be saved according to the scheme: one figure per zipped folder, and then uploaded

as .zip files. E.g. all the Source data files for figure 1 need to be saved in a single folder and this needs to be zipped and then uploaded as "SD figure 1.zip" file. For the source data contained in Excel worksheets, please make sure to include Figure panel labels as appropriate.

- During routine pre-acceptance checks, our data editors have raised the following queries regarding figures, data, and legends, which I would ask you to address (ideally using the Track Changes option):

1. Please note that the legend for figure 4D is missing in the manuscript. This needs to be rectified
2. Please note that the exact p values are not provided in the legends of figures 4B, C; EV4 C, L; EV5 B-D
3. Please indicate the statistical test used for data analysis in the legends of figures 1E-G; 2C, EV2 C, D; EV4 C, L
4. Please note that the box plots need to be defined in terms of minima, maxima, centre, bounds of box and whiskers, and percentile in the legends of figures 3F, 6B, EV1 C, D; EV4 C
5. Please note that information related to n is missing in the legends of figures 3F, 4D, 6B, EV1 C, D; EV5 B-D
6. Please note that the error bars are not defined in the legends of figures EV5 B-D

- Finally, please provide suggestions for a short 'blurb' text prefacing and summing up the conceptual aspect of the study in two sentences (max. 250 characters), followed by 3-5 one-sentence 'bullet points' with brief factual statements of key results of the paper; they will form the basis of an editor-written 'Synopsis' accompanying the online version of the article. Please also upload a synopsis image, which can be used as a "visual title" for the synopsis section of your paper. The image should be in PNG or JPG format, and please make sure that it remains in the modest dimensions of (exactly) 550 pixels wide and 300-600 pixels high.

I am therefore returning the manuscript to you for a final round of revision, to allow you to make these changes and upload all modified files. Should you have any questions regarding the referee comments or this decision, please do not hesitate to contact me directly.

With kind regards,

Hartmut

*** PLEASE NOTE: All revised manuscript are subject to initial checks for completeness and adherence to our formatting guidelines. Revisions may be returned to the authors and delayed in their editorial re-evaluation if they fail to comply to the following requirements. As a first step please read our guidelines for revised submissions:
<https://link.springer.com/journal/44318/submission-guidelines#cms-Revised-submissions>

1) Every manuscript requires a Data Availability section (even if only stating that no deposited datasets are included). Primary datasets or computer code produced in the current study have to be deposited in appropriate public repositories prior to resubmission, and reviewer access details provided in case that public access is not yet allowed.

4) Each main and each Expanded View (EV) figure should be uploaded as individual production-quality files (preferably in .eps, .tif, .jpg formats). For suggestions on figure preparation/layout, please refer to our Figure Preparation Guidelines:
<https://media.springernature.com/original/springer-cms/rest/v1/content/27825798/data/v1>

6) Please complete our Author Checklist, and make sure that information entered into the checklist is also reflected in the manuscript; the checklist will be available to readers as part of the Review Process File.

8) Please note that supplementary information at EMBO Press has been superseded by the 'Expanded View' for inclusion of additional figures, tables, movies or datasets; with up to five EV Figures being typeset and directly accessible in the HTML version of the article.

9) To facilitate reproducibility and cross-laboratory adoption of methodologies, please structure the Materials & Methods section as outlined in our guide to authors, including a completed Reagents and Tools Table.

10) Digital image enhancement is acceptable practice, as long as it accurately represents the original data and conforms to community standards. If a figure has been subjected to significant electronic manipulation, this must be clearly noted in the figure legend and/or the 'Materials and Methods' section. The editors reserve the right to request original versions of figures and the original images that were used to assemble the figure. Finally, we generally encourage uploading of numerical as well as gel/blot image source data.

In the interest of ensuring the conceptual advance provided by the work, we recommend submitting a revision within 3 months (26th Feb 2026). Please discuss the revision progress ahead of this time with the editor if you require more time to complete the revisions. Use the link below to submit your revision:

Link Not Available

Referee #1:

The authors have addressed many of my previous comments appropriately.

I am still questioning whether the Nrf2 antibody that they are using is detecting Nrf2, because in their western blots the detected band migrates at the expected molecular weight, which Nrf2 never does; for example, see:

<https://pmc.ncbi.nlm.nih.gov/articles/PMC3503463/>

This can be seen even in the supplier validation link that they provide (<https://www.thermofisher.com/antibody/product/Nrf2-Antibody-Polyclonal/PA5-27882>), where it can be seen that, depending on the electrophoresis conditions, Nrf2 does not migrate at the expected molecular weight.

Also, the authors correctly state that TP53 mutations are common in lung cancer, but it is not appropriate to justify the relevance of their mouse model based on this fact, because their mouse model is p53-null, and p53 has multiple isoforms, and mutant(s) p53 can lead to different outcomes than the absence of p53 (e.g. see: <https://pmc.ncbi.nlm.nih.gov/articles/PMC9090915/>).

Referee #3:

The authors have significantly improved the manuscript with experiments and alterations to the manuscript that have addressed many of my previous concerns. There are still some concerns which are noted below. However, once these concerns are addressed, I believe this manuscript is suitable for publication.

Concerns:

1. The Discussion section still discusses E3 ligases as a general vulnerability for KEAP1 mutant lung cancers, particularly in the first several paragraphs of the Discussion. Genetic context is discussed much later but with much less emphasis. However, I believe this is not accurate. All of the data in this manuscript for E3 ligases as vulnerabilities for mutant KEAP1 is in the context of p53 loss. There are no data with KEAP1 loss alone, nor KEAP1 loss with other tumor suppressor loss such as LKB1/STK11 or PTEN without p53 loss. I believe this is an important point. Human NSCLC with concomitant loss of LKB1 and KEAP1 are highly aggressive and have lower responses to therapy (Shen et al., J. of Precision Oncol, 2019; Wolhieter et al., Cell Reports, 2020; Galan-Cobo et al., Cancer Cell 2025). In contrast, in this manuscript, loss of KEAP1 in KPL mice leads to significant loss of tumor formation and burden. Interestingly, in Shen et al. (J. of Precision Oncol, 2019), NSCLC with mutations in KRAS, KEAP1, STK11, and TP53 and mutations in KEAP1, STK11, and TP53 regardless of oncogene drivers are also in the highest risk category in stark contrast to the mouse results here. Thus, the context of vulnerabilities of KEAP1 mutants to E3 ligase

inhibition should be more strongly emphasized. Also, the differences that are seen in the mouse models here and human lung cancers in the context of LKB1 loss (as noted above) should also be discussed.

2. It's unclear how to interpret the data in Fig AF2 and AF3 are helpful.

a. H2170 and A549, with genetically altered KEAP1, presumably have dysfunctional KEAP1. If so, why are the effects of bardoxolonein and omaveloxolone, as KEAP1 inhibitors, on NF2 nuclear localization and ALDH3A1 cytoplasmic localization the same as H1299 and CALU1 with wt KEAP1? I'm not sure if NF2 nuclear localization and ALDH3A1 cytoplasmic localization are really good read outs of KEAP1 function?

b. The effects of bardoxolonein Fig AF2 and AF3 seem similar to the effects of KI696 although the text notes bardoxolonein's effects were significant but KI696's effects were not. Statistical evaluation is needed in all of the graphs.

c. There is no explanation as to how bardoxolonein and omaveloxolone, as KEAP1 inhibitors, induce cell death in KEAP1 mutant cell line, A549, when presumably KEAP1 is dysfunctional. Could this be off-target effects of BM and omaveloxolone? Also, why doesn't KI696 have any effects on cell growth while bardoxolonein and omaveloxolone do? Genetic knock out or knock down of KEAP1 in A549 cells would be helpful here.

d. It should also be noted that A549 cells are mutant for both LKB1 and KEAP1 but wild type for TP53, which fits outside of the context of the KPKL mouse model which showed significant inhibition of tumor formation.

3. The IHC stains in Fig. 3 and 5 are still relatively poor even with magnified images. What are presumably positive stains are very light and difficult to distinguish from the experimental controls.

4. MINOR: Most of the survival curves in the manuscript are labeled "Lung Cancer" but should be labeled "NSCLC" to be more accurate. As noted previously, "lung cancer" includes both SCLC and NSCLC.

Referee #4:

The authors have addressed my previously raised concerns. I support the publication of it present version.

Referee #1:

The authors have addressed many of my previous comments appropriately.

We thank the reviewer for their positive assessment and appreciate the acknowledgment of our revisions.

1. I am still questioning whether the Nrf2 antibody that they are using is detecting Nrf2, because in their western blots the detected band migrates at the expected molecular weight, which Nrf2 never does; for example, see:

<https://pmc.ncbi.nlm.nih.gov/articles/PMC3503463/>

This can be seen even in the supplier validation link that they provide

(<https://www.thermofisher.com/antibody/product/Nrf2-Antibody-Polyclonal/PA5-27882>), where it can be seen that, depending on the electrophoresis conditions, Nrf2 does not migrate at the expected molecular weight.

We appreciate the reviewer's careful attention to this point. We apologize for the earlier oversight in annotating the molecular weight of the NRF2 band in our western blot. Upon re-examining the original gel files, we confirm that the NRF2 band in our samples migrates at approximately ~80 kDa (Reviewer's Figure 1), which is consistent with the known unusual migration pattern of NRF2 and matches the supplier's validation data. We have now revised the figure to correctly annotate the molecular weight markers for all proteins and to provide a clearer and more accurate presentation.

Reviewers Figure 1

2. Also, the authors correctly state that TP53 mutations are common in lung cancer, but it is not appropriate to justify the relevance of their mouse model based on this fact, because their mouse model is p53-null, and p53 has multiple isoforms, and mutant(s) p53 can lead to different outcomes than the absence of p53 (e.g. see: <https://pmc.ncbi.nlm.nih.gov/articles/PMC9090915/>).

We fully agree that all data presented in this manuscript identifying E3 ligases as vulnerabilities in KEAP1-mutant settings are specifically derived from models with concomitant TP53 loss, in combination with KRAS^{G12D}. While this represents a widely used genetic setting and reflects a frequent co-alteration in human NSCLC (PMID: 19561589; 28967920; 33738287), we acknowledge that TP53-null models are not equivalent to tumors harboring mutant TP53, as distinct TP53 mutations and isoforms can exert gain-of-function or dominant-negative effects that are not captured by p53 loss alone. Accordingly, we have revised the Discussion to clearly state that our conclusions are restricted to a TP53-null genetic context.

Referee #3:

The authors have significantly improved the manuscript with experiments and alterations to the manuscript that have addressed many of my previous concerns. There are still some concerns which are noted below. However, once these concerns are addressed, I believe this manuscript is suitable for publication.

We thank the reviewer for their positive assessment and appreciate the acknowledgement of the substantial revisions made to the manuscript. We have carefully considered the additional concerns raised and provide detailed responses below. We hope that our clarifications and revisions satisfactorily address all remaining points.

Concerns:

1. The Discussion section still discusses E3 ligases as a general vulnerability for KEAP1 mutant lung cancers, particularly in the first several paragraphs of the Discussion. Genetic context is discussed much later but with much less emphasis. However, I believe this is not accurate. All of the data in this manuscript for E3 ligases as vulnerabilities for mutant KEAP1 is in the context of p53 loss. There are no data with KEAP1 loss alone, nor KEAP1 loss with other tumor suppressor loss such as LKB1/STK11 or PTEN without p53 loss. I believe this is an important point. Human NSCLC with concomitant loss of LKB1 and KEAP1 are highly aggressive and have lower responses to therapy (Shen et al., J. of Precision Oncol, 2019; Wolhietter et al., Cell Reports, 2020; Galan-Cobo et al., Cancer Cell 2025). In contrast, in this manuscript, loss of KEAP1 in KPL mice leads to significant loss of tumor formation and burden. Interestingly, in Shen et al. (J. of Precision Oncol, 2019), NSCLC with mutations in KRAS, KEAP1, STK11, and TP53 and mutations in KEAP1, STK11, and TP53 regardless of oncogene drivers are also in the highest risk category in stark contrast to the mouse results here. Thus, the context of vulnerabilities of KEAP1 mutants to E3 ligase inhibition should be more strongly emphasized. Also, the differences that are seen in the mouse models here and human lung cancers in the context of LKB1 loss (as noted above) should also be discussed.

As suggested by the reviewer, we have revised the Discussion to introduce genetic context earlier and more prominently. We fully agree that all data presented in this manuscript identifying E3 ligases as vulnerabilities in KEAP1-mutant settings are specifically derived from models with concomitant TP53 loss, which is a widely used genetic setting and reflects a frequent co-alteration in human NSCLC (PMID: 19561589; 28967920; 33738287). We have now clarified this point explicitly to avoid overgeneralization of KEAP1-dependent vulnerabilities.

We also acknowledge the reviewer's observation regarding the discrepancy between our murine models and human NSCLC, particularly in the context of KEAP1 and LKB1/STK11 co-loss. As noted in our previous revision (Revision Round 1, Reviewer #3, Comment 3), we have expanded the Discussion to address these differences directly.

The inability of KPLK and KPKP models to develop tumors *in vivo* may reflect biological differences in tumor initiation and progression, including sequential mutagenesis timing, selective pressures, or buffering effects from co-mutations. Additionally, it remains possible that TP53 loss acts as a key stratifier for KEAP1 function. Such factors may help explain the existence of human KPLK tumors despite their apparent absence in the murine models used here. We have incorporated these considerations into the revised Discussion to clearly acknowledge the context-dependent nature of KEAP1-associated vulnerabilities, the limitations of the current models, and the need for future studies examining alternative genetic backgrounds.

2. It's unclear how to interpret the data in Fig AF2 and AF3 are helpful.

a. H2170 and A549, with genetically altered KEAP1, presumably have dysfunctional KEAP1. If so, why are the effects of bardoxolonein and omeveloxolone, as KEAP1 inhibitors, on NRF2 nuclear localization and ALDH3A1 cytoplasmic localization the same as H1299 and CALU1 with wt KEAP1? I'm not sure if NRF2 nuclear localization and ALDH3A1 cytoplasmic localization are really good read outs of KEAP1 function?

We appreciate the reviewer's concern regarding the KEAP1-mutant cell lines A549 and NCI-H2170. In line with the reviewer's suggestion, we have amended the figures and corresponding text and have excluded these data from the revised manuscript.

That said, it is worth noting that structural and functional analyses of KEAP1 somatic mutations derived from lung cancer patients have shown that several KEAP1 mutations do not completely abolish NRF2 binding or E3 ligase function (PMID:16507366). Thus, residual KEAP1 regulatory capacity may persist in certain mutant contexts. Consistent with this, KEAP1 mutations exhibit marked functional heterogeneity, including wild-type-like passenger events, super-binder mutants, and hypomorphic variants that retain NRF2 binding but fail to promote its degradation. Collectively, these findings indicate that KEAP1 mutations do not uniformly cause complete loss of function but instead lead to diverse regulatory effects on NRF2 stability and activity (PMID:24322982).

Beyond KEAP1 status, NSCLC cell lines differ substantially in histological subtype, TP53 and KRAS alterations, basal NRF2 activity, metabolic state, redox buffering capacity, and differentiation status, all of which can independently influence NRF2 signaling and drug responsiveness (PMID:17020408; 16507366; 23272301; 31406302).

The responses observed in A549 and NCI-H2170 cells are therefore unlikely to reflect KEAP1-dependent regulation. Instead, they are more likely attributable to the broad, electrophilic nature of bardoxolone methyl and omeveloxolone, which can influence NRF2 signaling independently of KEAP1 status. However, this interpretation remains speculative and would require further mechanistic validation, which is beyond the scope of the present study.

b. The effects of bardoxolonein Fig AF2 and AF3 seem similar to the effects of KI696 although the text notes bardoxolonein's effects were significant but KI696's effects were not. Statistical evaluation is needed in all of the graphs.

We thank the reviewer for pointing this out and apologize for the earlier oversight in not clearly indicating the statistical evaluation in Fig. AF2 and AF3. We have now incorporated appropriate statistical analyses in all graphs, and the figure legends have been updated accordingly.

Upon treatment with bardoxolone methyl and omeveloxolone, we observed significant alterations in the abundance of KEAP1 and NRF2, as well as ALDH3A1 (Fig. AF2). Furthermore, treatment with these two KEAP1–NRF2 inhibitors resulted in significant growth retardation and increased cell death (Fig. AF3).

c. There is no explanation as to how bardoxolonein and omeveloxolone, as KEAP1 inhibitors, induce cell death in KEAP1 mutant cell line, A549, when presumably KEAP1 is dysfunctional.

Could this be off-target effects of BM and omeveloxolone? Also, why doesn't KI696 have any effects on cell growth while bardoxolonein and omeveloxolone do? Genetic knock out or knock down of KEAP1 in A549 cells would be helpful here.

Bardoxolone methyl (BM) and omeveloxolone are electrophilic agents that primarily exert their activity through covalent modification of specific cysteine residues on KEAP1, most notably Cys151. However, due to their highly electrophilic nature, these compounds also react with multiple thiol-containing proteins beyond KEAP1, resulting in broad off-target effects. For example, both bardoxolone methyl and omeveloxolone have been reported to inhibit NF-κB signaling. In addition, bardoxolone methyl can abrogate ferroptosis and induce apoptosis and autophagy in cancer cells, while omeveloxolone activates NRF2 and suppresses NF-κB signaling. In contrast, KI696 is a selective inhibitor of the KEAP1–NRF2 interaction, which may explain its comparatively minor effects on cell growth. It is therefore plausible that the combined perturbation of multiple signaling pathways, including NRF2, NF-κB, and other redox-sensitive pathways, contributes to the selective cell death observed upon treatment with bardoxolone methyl and omeveloxolone, whereas KI696 shows limited effects. Considering that we have amended the figures and corresponding text and have excluded KI696 data from the revised manuscript.

Notably, our mass spectrometry and transcriptomic analyses of lung cancer cell lines treated with KI696 (Figure 2 and EV2) indicate upregulation of NF-κB signaling, which may induce proteostatic stress. However, the precise reasons why KI696 does not significantly affect cell growth, in contrast to bardoxolone methyl and omeveloxolone, remain unclear. A detailed mechanistic characterization of these compounds, is beyond the scope of the present manuscript.

d. It should also be noted that A549 cells are mutant for both LKB1 and KEAP1 but wild type for TP53, which fits outside of the context of the KPKL mouse model which showed significant inhibition of tumor formation.

We appreciate the reviewer's comment and have therefore re-analyzed publicly available mutational data from lung cancer patients using the TCGA cohort, with cBioPortal employed as the visualization platform. Specifically, we assessed the co-occurrence of genetic alterations in KEAP1, KRAS, TP53, STK11, and PTEN. Consistent with our earlier observations, the simultaneous occurrence of TP53^{mut}:STK11^{mut}:KRAS^{mut} and KEAP1^{mut} remains very low (0,6%).

It is possible that loss of TP53 alters the tolerance of tumour cells towards concurrent loss of KEAP1 and STK11, thereby acting as a strong negative selective pressure. Given that KEAP1 was reported to be involved in DNA damage and that loss of TP53 can result in genomic instability, such mutational combinations may be counter-selected in both human tumors and mouse models. This interpretation is supported by the data presented in this manuscript, the additional analyses provided in this rebuttal, and recent findings reported by Sotillo and colleagues (PMID: 40986428).

As mentioned in response to earlier comments, we have now excluded data from A549 and NCI-H2170 cells from the revised manuscript to improve clarity and ensure appropriate alignment with the genetic context of the in vivo model.

3. The IHC stains in Fig. 3 and 5 are still relatively poor even with magnified images. What are presumably positive stains are very light and difficult to distinguish from the experimental controls.

We carefully re-evaluated the immunohistochemical stainings and optimized the exposure settings. Where necessary, the stainings were repeated. We hope that the revised figures now allow clearer visualization of the positive signals and better distinguish them from the corresponding controls.

4. MINOR: Most of the survival curves in the manuscript are labeled "Lung Cancer" but should be labeled "NSCLC" to be more accurate. As noted previously, "lung cancer" includes both SCLC and NSCLC.

We appreciate the reviewer's comment. In line with the suggestion, we have revised all survival curve labels to "NSCLC" to accurately reflect the specific cancer subtype analyzed.

Referee #4:

The authors have addressed my previously raised concerns. I support the publication of it present version.

We sincerely thank the reviewer for their thoughtful assessment and greatly appreciate their support for the publication of the revised manuscript.

Prof. Ivan Dikic
Institute of Biochemistry II, Faculty of Medicine, Goethe University, Frankfurt, Germany
Institute of Biochemistry II
Theodor-Stern-Kai 7
Frankfurt 60590
Germany

11th Feb 2026

Re: EMBOJ-2024-119715R1
Targeting Ubiquitin Signaling Vulnerabilities in KEAP1-Inactivated Lung Cancer

Dear Ivan,

Thank you for submitting your final revised manuscript for our consideration. I am pleased to inform you that we have now accepted it for publication in The EMBO Journal.

You may qualify for financial assistance for your publication charges - either via a Springer Nature fully open access agreement or an EMBO initiative. Check your eligibility: <https://link.springer.com/journal/44318/how-to-publish-with-us>

With kind regards,

Hartmut

Please note that it is The EMBO Journal policy for the transcript of the editorial process (containing referee reports and your response letters) to be published as an online supplement to each paper. If you should prefer removal of any referee-only figures included in the point-by-point response(s), e.g. because they may still be used for future publication or because they have been reproduced from published work by others, please do let us know immediately via response email.

More information is available here: <https://link.springer.com/partners/embo-press/editorial-policies#Peer%20review>